# Sharp Gaussian approximations for Decentralized Federated Learning

**Soham Bonnerjee**
sohambonnerjee@uchicago.edu

**Sayar Karmakar**
sayarkarmakar@ufl.edu

**Wei Biao Wu**
wbwu@uchicago.edu

## Abstract

Federated Learning has gained traction in privacy-sensitive collaborative environments, with local SGD emerging as a key optimization method in decentralized settings. While its convergence properties are well-studied, asymptotic statistical guarantees beyond convergence remain limited. In this paper, we present two generalized Gaussian approximation results for local SGD and explore their implications. First, we prove a Berry-Esseen theorem for the final local SGD iterates, enabling valid multiplier bootstrap procedures. Second, motivated by robustness considerations, we introduce two distinct time-uniform Gaussian approximations for the entire trajectory of local SGD. The time-uniform approximations support Gaussian bootstrap-based tests for detecting adversarial attacks. Extensive simulations are provided to support our theoretical results.

## 1 Introduction

Federated Learning (FL), introduced by McMahan et al. [2017] as a decentralized model training paradigm while maintaining privacy, has seen rapid advancements driven by its applicability in domains such as next-word prediction on mobile devices, healthcare, and cross-silo collaborations among institutions. Subsequent works Kairouz et al. [2021], Li et al. [2020], Karimireddy et al. [2020], Wang et al. [2020b], Alistarh et al. [2017], Lin et al. [2018] have addressed key challenges around privacy and computational efficiency. Research has also extended to decentralized federated learning (DFL) Lalitha et al. [2019], Lian et al. [2017], He et al. [2019], Kim et al. [2020], Lian et al. [2017], Wang and Joshi [2021], Singh et al. [2023], which eliminates reliance on a central server by enabling peer-to-peer collaboration, thereby enhancing robustness, fairness, and resilience to adversarial threats. We refer to Gabrielli et al. [2023], Yuan et al. [2024] for a comprehensive survey of the literature. In this regard, Local SGD Stich [2019], Khaled et al. [2020], Woodworth et al. [2020b] has emerged as a widely adopted algorithm, allowing clients to perform multiple local updates before synchronizing, significantly reducing communication overhead.

While theoretical guarantees for convergence and speed in local SGD have been developed Haddad-pour et al. [2019], Woodworth et al. [2020a], Koloskova et al. [2020], a gap remains in understanding the statistical properties of fluctuations around the true parameter vector. This gap has practical implications: first, statistical guarantees on the final iterates are essential for inference; second, monitoring the entire trajectory is crucial for detecting adversarial behavior in high-stakes settings like traffic networks, autonomous systems, and financial platforms. For the first issue, emerging works on central limit theory Li et al. [2022], Gu and Chen [2024] provide initial insights, but estimating local covariance structures is numerically intensive. Multiplier bootstrap methods Fang et al. [2018], Fang [2019] offer computational relief, but require stronger results beyond the central limit theory. The second issue is even more challenging, as it demands control over the entire trajectory of local SGD, not just the last iterate. Classical inferential methods struggle with DFLs complex dependency structure, and a key open question is how to develop statistically valid, computationally efficient inference methods with minimal distributional assumptions and explicit error control.

39th Conference on Neural Information Processing Systems (NeurIPS 2025).

## 1.1 Main Contributions

In this article, we address this gap by proposing different refined Gaussian approximations that naturally lead to suitable bootstrap procedures. Our results go beyond central limit theory to establish sharper, step-by-step as well as uniform control over the DFL iterates $\{Y_t\}$. These results not only facilitate relevant bootstrap-based inference to produce asymptotically-valid confidence sets, but also enables us to perform statistical hypothesis tests to detect attacks, which are inaccessible otherwise. Our main contributions can be summarized as follows:

($\mathbf{1}$) In Section 2, we provide an explicit characterization of the Berry-Esseen error for the Polyak-Ruppert version of the local SGD algorithm (Algorithm 1) iterates. In particular, under standard regularity conditions on the client-level optimization problems as well as for a general class of connection graph of clients, we prove:

**Theorem 1.1** (Theorem 2.1, Informal). *For a decentralized federated learning set-up with $K$ clients, the Polyak-ruppert averaged iterates of the local SGD algorithm with $n$ iterations, and step size $\eta_t \asymp t^{-\beta}$, achieves*

$$d_{\texttt{Berry-Esseen}} \lesssim n^{1/2-\beta}\sqrt{K}.$$

Our result explicitly underpins the source of the assumption $K = o(n^{2\beta-1})$ used to derive central limit theory for the local SGD iterates Gu and Chen [2024]. Theorem 2.1 is accompanied by a corresponding Berry-Esseen theorem (Theorem 2.2) for final iterates of the DFL algorithm. Both these theorems involve a finite sample scaling (equivalently, scaling by a covariance matrix depending upon $n$, the number of iterations) of the local SGD iterates, leading to optimal error bounds. Our result is first such Berry-Esseen bounds for the local SGD updates.

($\mathbf{2}$) The finite sample scaling considered in Theorems 2.1 and 2.2 is usually not estimable. Shifting focus to an asymptotic, global scaling, our results uncover a novel computation-communication trade-off involving the Berry-Esseen result. In Theorem 2.3, we show that for $K = o(\sqrt{n})$, $\beta = 3/4$ represents an optimal choice of step-size; however, for $K \gtrsim \sqrt{n}$, for no $\beta \in (1/2, 1)$ does the Berry-Esseen bound converge towards zero. This observation is not merely an artifact of our proof, and the phase-transition are empirically validated through extensive simulations.

($\mathbf{3}$) A key motivation behind the local SGD algorithm is maintaining privacy. From this perspective, asymptotic inference on the final iterates is insufficient for detecting breach of privacy through some adversarial attack. Indeed, in Section 3, we discuss a general framework to detect a broad class of model poisoning in a distributed setting. Through an example in Section 3.1, we point out a class of maximal statistics which can be used to detect such attacks. Moreover, to perform inference on such statistics, we move beyond controlling simply the end-term iterates to a more general time-uniform Gaussian coupling of the entire local SGD process. Motivated from above, in Theorem 3.1, we establish a time-uniform Gaussian approximation.

**Theorem 1.2** (Theorem 3.1, Informal). *If the local SGD algorithm with $K$ clients runs $n$ iterations with step size $\eta_t \asymp t^{-\beta}$, then there exists a Gaussian process $Y_t^G = (I - \eta_t A)Y_{t-1}^G + \eta_t Z_t$ with $Z_t$ i.i.d. $N(0, \Gamma)$ for some matrix $\Gamma$ and $A$ being the Hessian of the problem, such that,*

$$\max_{1 \leq t \leq n} |\sum_{s=1}^{t}(Y_s - Y_s^G)| \approx o_{\mathbb{P}}(n^{1-\beta} + \frac{n^{1/p}}{\sqrt{K}}).$$

Here we assume $p \geq 2$ finite moments of the local noisy gradients. To facilitate bootstrap, we also provide an explicit characterization of $\Gamma$. To the best of our knowledge, these results constitute the first time-uniform Gaussian approximation results for stochastic approximation algorithms.

($\mathbf{4}$) In particular, Theorem 3.1 presents a Gaussian approximation (referred to as `Aggr-GA`) with a slightly sharper rate, but one requiring extensive synchronization during the bootstrap procedure. Recognizing that this may not be ideal from a privacy perspective, we further present a separate, client-level Gaussian approximation `Client-GA` in Theorem 3.2, which completely mimics the local SGD procedure. The approximation `Client-GA` is much more localized, leading to slight worsening of the approximation error but increased efficiency with regards to synchronization and computational cost. We argue and validate with simulations, that our Gaussian approximations are much sharper than that indicated by a standard, off-the-shelf functional central-limit theorem. In fact, our Gaussian approximations represent a version of the covariance-matching approximations introduced by Bonnerjee et al. [2024], however in a multivariate, non-stationary environment.

(**5**) Finally, in Section 4, we validate our theoretical findings with extensive numerical exercises. Our simulation results in Sections 4.1 and 4.2 not only indicate the sharpness of our theoretical results, but also project vividly the computation-communication trade-offs discussed in Remark 2.2. Moreover, the numerical results in Section 4.3 shows that the proposed Gaussian approximations `Aggr-GA` and `Client-GA` are significantly better than an off-the-shelf Brownian-motion based approximation, even in finite sample, complementing Theorems 3.1 and 3.2 well.

## 1.2 Notations

In this paper, we denote the set $\{1, \ldots, n\}$ by $[n]$. The $d$-dimensional Euclidean space is $\mathbb{R}^d$, with $\mathbb{R}^d_{>0}$ the positive orthant. For a vector $a \in \mathbb{R}^d$, $|a|$ denotes its Euclidean norm. The set of $m \times n$ real matrices is denoted by $\mathbb{R}^{m \times n}$, and correspondingly, for $M \in \mathbb{R}^{m \times n}$, $|M|_F$ denotes its Frobenius norm. For a random vector $X \in \mathbb{R}^d$, we denote $\|X\| := \sqrt{\mathbb{E}[|X|^2]}$. We also denote in-probability convergence, and stochastic boundedness by $o_{\mathbb{P}}$ and $O_{\mathbb{P}}$ respectively. We write $a_n \lesssim b_n$ if $a_n \leq Cb_n$ for some constant $C > 0$, and $a_n \asymp b_n$ if $C_1 b_n \leq a_n \leq C_2 b_n$ for some constants $C_1, C_2 > 0$.

## 1.3 Related Literature

In view of the plethora of classical literature for central limit theorems (CLT) on SGD and its different variants Ruppert [1988], Polyak and Juditsky [1992], Chen et al. [2020], it is rather surprising that this area has remained relatively untouched for local SGD or DFL. Li et al. [2022] establish a functional CLT for local SGD, but only when the number of clients is held fixed. More recently, Gu and Chen [2024] established a central limit theory for DFL while allowing an increasing number of clients. Non-asymptotic guarantees for SA algorithms exist in terms of MSE guarantees Nemirovski et al. [2009], Moulines and Bach [2011], Lan [2012], Mou et al. [2024]. Recently, Anastasiou et al. [2019] employed Stein's method to derive Gaussian approximation for a class of smooth functions of the SGD iterates. Later, Shao and Zhang [2022] obtains the first Berry-Esseen result for online SGD. Samsonov et al. [2024] extended the result to linear stochastic approximation algorithms and temporal difference learning, before being further improved by Wu et al. [2024], Sheshukova et al. [2025].

On the other hand, to the best of our knowledge, time-uniform 'entire-path' Gaussian approximation results have not appeared in the stochastic approximation literature. From classical time-series literature, such approximations are known as "Komlos-Major-Tusnady"(KMT) approximations, and have a long history Komlós et al. [1975], Sakhanenko [1984, 1989, 2006], Götze and Zaitsev [2008], Berkes et al. [2014], Karmakar and Wu [2020] and varied uses in change-point detection Wu and Zhao [2007], wavelet analysis Bonnerjee et al. [2024], simultaneous and time-uniform inference Liu and Wu [2010], Xie et al. [2020], Karmakar et al. [2022], Waudby-Smith et al. [2024]. However, this results require fast enough decay, and well-conditioned covariance structure, which are not usually available in even stochastic approximation algorithms with decaying step-size, let alone a general local SGD algorithm. Therefore, such results are not readily applicable in the current settings.

## 2 Berry-Esseen theory for local SGD

In this section we establish a general, Berry-Esseen type Gaussian approximation result in the decentralized federated learning setting. In order to rigorously state our results, it is imperative that we formally introduce the local stochastic gradient descent (SGD) algorithm and underline the key assumptions behind our theoretical results. This is done in Section 2.1. Finally, we present our first Gaussian approximation results in Section 2.2, and discuss the implications therein.

### 2.1 Preliminaries

Consider a typical decentralized heterogeneous federated learning setting with $K$ clients, each having access to a loss function $f_k : \mathbb{R}^d \times \mathbb{R}^{n_k} \to \mathbb{R}$, and a distribution $\mathcal{P}_k$ on $\mathbb{R}^{n_k}$ for $k \in [K]$. Here, $\mathcal{P}_k$ determines the distribution of the local noisy gradient for each client, realized by sampling $\xi^k \sim \mathcal{P}_k$. We allow for heterogeneity among the clients i.e. $\mathcal{P}_k$'s are allowed to be different. However, noise sampling (i.e. the $\xi^k$) is assumed to be independent from one client to the another. The corresponding risk or regret for the $k$-th client is denoted by $F_k(\theta) = \mathbb{E}_{\xi^k \sim \mathcal{P}_k} f_k(\theta, \xi^k)$. Con-

sider a pre-specified "importance" or weight schedule, given by $\{w_1, w_2, \ldots, w_K\} \in \mathbb{R}^K$, such that $\sum_{k=1}^K w_k = 1$. In an online federated learning setting, the weight schedule are typically known a-priori, usually informed by the level of heterogeneity for each client, and specified by the moderator of the decentralized system. The goal of DFL is to obtain

$$\theta_K^\star = \arg\min_\theta \sum_{k=1}^K w_k F_k(\theta) \in \mathbb{R}^d. \tag{2.1}$$

### 2.1.1 Communication

The client-level information is defined by loss functions $f_k$ and weights $w_k$. A key aspect of federated learning (FL) is preserving client privacy, often achieved via a synchronization step with parameter $\tau \in \mathbb{N}$. At each $\tau$-th step, the moderator aggregates client data and redistributes it following a policy. In decentralized SGD, averaging schemes Chaturapruek et al. [2015], Lian et al. [2017], Ivkin et al. [2019] or gossip-based methods Koloskova et al. [2020], Li et al. [2019], Qin et al. [2021], Wang and Joshi [2021] are common. In other words, a linear aggregation based on a fixed connection graph, is employed at the synchronization step. Following the notation of Gu and Chen [2024], we consider a connection network of the participating clients in the FL system, defined by an undirected graph $G = (V, E)$ where $V = \{v_k\}_{k=1}^K$ represents the set of clients and E specifies the edge set such that $(i, j) \in E$ if and only if clients $i$ and $j$ are connected. Let $\mathbf{C} = (c_{ij}) \in \mathbb{R}^{K \times K}$ be a symmetric connection matrix defined on $G = (V, E)$, where $c_{ij}$ is a nonnegative constant that specifies the contribution of the $j$ th data block to the estimation at node $i$. It is required that $c_{ij} > 0$ if and only if $(i, j) \in E$ and $\mathbf{C1} = \mathbf{1}$. Moreover, let $c_{i,i} > 0$.

Suppose $\mathbf{\Theta}_t = (\theta_t^1, \ldots, \theta_t^K) \in \mathbb{R}^{d \times K}$ denotes the local parameter updates of each client at the $t$-th step. Suppose the corresponding local gradient updates be summarized in the matrix $\mathbf{G}_t = K(w_1 \nabla f_1(\theta_{t-1}^1, \xi_t^1), \ldots, w_K \nabla f_K(\theta_{t-1}^K, \xi_t^K)) \in \mathbb{R}^{d \times K}$. Here, the initial points $\theta_0^k \in \mathbb{R}^d$ are arbitrarily initialized for $k \in [K]$, and have no bearing on the theoretical results. For the sake of completion, we also re-state the `local SGD` algorithm using the notations and the set-up established in the preceding sections 2.1 and 2.1.1.

---

**Algorithm 1** `local SGD`

---

**Input:** Initializations $\mathbf{\Theta}_0 = (\theta_0^1, \ldots, \theta_0^K) \in \mathbb{R}^{d \times K}$; Connection matrix $\mathbf{C}$; Synchronization parameter $\tau \in \mathbb{N}$; Loss functions $f_k(\cdot, \xi^k), \xi^k \sim \mathcal{P}_k, k \in [K]$, weights $\{w_k\}_{k=1}^K$, number of iterations $n$, step-size schedules $\{\eta_t\}_{t=1}^n$.

- Let $E_\tau = \{\tau, 2\tau, \ldots, L\tau\}$, where $L = \lfloor \frac{n}{\tau} \rfloor$.

- For $t = 1, \ldots, n$: $\quad \mathbf{\Theta}_t = (\mathbf{\Theta}_{t-1} - \eta_t \mathbf{G}_t) C_t, \quad C_t = \begin{cases} \mathbf{C}, & t \in E_\tau, \\ I_K, & \text{otherwise.} \end{cases}$ (2.2)

**Output:** $Y_n := K^{-1} \mathbf{\Theta}_n \mathbf{1} = K^{-1} \sum_{k=1}^K \theta_n^k$.

---

To simplify Algorithm 1, each client runs an SGD in parallel till every $\tau$-th step, when they must synchronize their updates in order to properly solve the optimization problem (2.1). Clearly, for $\tau = 1$, Algorithm 1 reduces to the vanilla SGD algorithm for (2.1), which hampers privacy as well as incurs great cost at each step, since typically, the number of clients $K$ increases with the number of iterations $n$. On the other hand, when $\tau > n$, there is no *synchronization*, and each client would solve their own local optimization problem $\arg\min_\theta F_k(\theta)$, defeating the benefits of sharing information. For the purpose of this paper, we assume $\tau$ to be fixed. Moreover, on a client level, we also assume that there exists constants $b_1, b_2 > 0$ such that for every $k \in [K]$, $b_1 \leq K w_k \leq b_2$.

### 2.2 Berry-Esseen theorems for client-averaged `local SGD` updates

Before we describe the Berry-Esseen theorems, it is important we briefly describe the conditions under which it hold. We assume the usual conditions of strong-convexity (Assumption A.1), and the stochastic Lipschitz-ness of the noisy gradients $\nabla f_k$ (Assumption A.2). Moreover, we also assume the continuous differentiability of $f_k$'s (Assumption A.3). Due to space constraints, the detailed description of these assumptions, alongside an extended discussion, is relegated to Appendix A. Here, we discuss an additional condition unique to the decentralized federated learning setting.

**Assumption 2.1.** *The connection matrix $\mathbf{C}$ satisfies $\mathbf{C1} = \mathbf{1}$ and $\mathbf{C}^\top = \mathbf{C}$. Moreover, if $\lambda_1 \geq \ldots \geq \lambda_K$ denote the ordered eigen-values of $\mathbf{C}$, then $\lambda_1 = 1$, and $\lambda_2 = \rho < 1$ for some $\rho \in (0, 1)$.*

This assumption also appears in Gu and Chen [2024]. Assumption 2.1 ensures that $\mathbf{C}$ is irreducible and the corresponding stationary distribution is unique; equivalently the underlying graph G is connected, ensuring an overall information sharing between each pair of clients through repeated synchronization steps. Mathematically, this can also be observed by noting that $\lim_{s\to\infty} \mathbf{C}^s = K^{-1}\mathbf{11}^\top$. Now, we present the first Gaussian approximation result concerning `local` SGD updates. Define the generalized Kolmogorov-Smirnov metric between two random variables $Y$ and $Z$ as

$$d_{\mathrm{C}}(Y, Z) := \sup_{\aleph \in \mathcal{B}(\mathbb{R}^d): A \text{ convex}} \left|\mathbb{P}(Y \in \aleph) - \mathbb{P}(Z \in \aleph)\right|. \tag{2.3}$$

Consider the `local` SGD output $Y_n$ from Algorithm 1. Our first theorem considers its corresponding Polyak-Ruppert averaged version

$$\bar{Y}_n := n^{-1}\sum_{t=1}^{n} Y_t = K^{-1}\sum_{k=1}^{K} n^{-1}\sum_{t=1}^{n} \theta_t^k, \tag{2.4}$$

and provides a Berry-Esseen theorem, proved in appendix Section B.1.

**Theorem 2.1.** *Define $\mathcal{A}_s^t := \prod_{j=s+1}^{t}(I - \eta_t A)$, $\mathcal{A}_t^t = I$, where $A := \nabla_2 F(\theta_K^\star)$ for $t \in [n]$. Further, for $s \in [n]$, define the random vectors*

$$u_s = \eta_s \sum_{k=1}^{K} w_k \left(\sum_{j=s}^{n} \mathcal{A}_s^j\right) g_k(\theta_K^\star, \xi_s^k), \text{ with } \Sigma_n := n^{-1}\sum_{s=1}^{n} \mathbb{E}[|u_s u_s^\top|], \ g_k(\theta, \xi^k) = \nabla F_k(\theta) - \nabla f_k(\theta, \xi^k).$$

*Let there exist a constant $C$ such that for $\xi^k \sim \mathcal{P}_k, k \in [K]$, it holds $\max_{k \in [K]} \mathbb{E}[|g_k(\theta_K^\star, \xi^k)|^2] \leq C$. Suppose that the step-size schedules of the clients satisfy that $\eta_t = \eta_0(t + k_0)^{-\beta}$ for some fixed $\eta_0, k_0 > 0$, and $\beta \in (1/2, 1)$. Then, under Assumptions 2.1, A.1 and A.2 and A.3 with $p = 4$, and $\bar{Y}_n$ as in (2.4), it holds that*

$$d_{\mathrm{C}}(\sqrt{n}(\bar{Y}_n - \theta_K^\star), Z) \lesssim \frac{1}{\sqrt{nK}} + n^{\frac{1}{2}-\beta}\sqrt{K} + \frac{n^{-\frac{\beta}{2}}}{\sqrt{K}}, \tag{2.5}$$

*where $\lesssim$ hides constants involving $d, \beta, \mu, L$ and $\rho$, and $Z \sim N(0, \Sigma_n)$.*

A slightly more general result, characterizing the effects of heterogeneity and synchronization, is presented in Corollary F.1 in the appendix. We present Theorem 2.1 here due to its enhanced amenability to interpretation, which we provide in subsequent remarks.

*Remark* 2.1. For a fixed $\beta \in (1/2, 1)$, the term $n^{1/2-\beta}\sqrt{K}$ dominates, requiring $K = o(T^{2\beta-1})$ for the central limit theory to hold for $\bar{Y}_n$. This condition, also noted in Theorem 3 of Gu and Chen [2024] without justification, is explicitly clarified by (2.5). As $\beta \to 1$, the rate in (2.5) becomes $\sqrt{K/n}$. The inclusion of the three terms highlights the influence of $K$, which is unique to federated systems. The $1/\sqrt{nK}$ term reflects the central limit theorem's convergence rate. The $n^{1/2-\beta}\sqrt{K}$ term captures the problem's difficulty, which increases with the number of clients running local SGD in parallel. Lastly, $n^{-\beta/2}K^{-1/2}$ represents the benefit of synchronization and information aggregation across clients. Even though this term is asymptotically dominated by $n^{1/2-\beta}\sqrt{K}$, this commands considerable finite sample effects as shown in Section 4.1.

Often, due to privacy reasons, clients might be unwilling to share $n^{-1}\sum_{i=1}^{n}\theta_i^k$ at time-point $n$, which makes the application of Theorem 2.1 impossible. In such cases, one can simply use a corresponding Berry-Esseen bound for the end-term iterates, which we provide in the following.

**Theorem 2.2.** *Under the assumptions of Theorem 2.1, it holds that*

$$d_{\mathrm{C}}(n^{\beta/2}(Y_n - \theta_K^\star), Z) \lesssim \frac{n^{-\beta/2}}{\sqrt{K}} + n^{\frac{1}{2}-\beta}\sqrt{K}, \tag{2.6}$$

*where $Z \sim N(0, \tilde{\Sigma}_n)$ with $\tilde{\Sigma}_n := n^\beta \sum_{s=1}^{n} \mathrm{Var}(\mathcal{A}_s^n \sum_{k=1}^{K} \eta_{s,k} w_k g_k(\theta_K^\star, \xi_s^k))$.*

Theorem 2.2 is proved in appendix Section B.2. When $K \asymp 1$, the rate (2.6) is consistent with the well-established asymptotic theory of SGD Chung [1954], Sacks [1958], Fabian [1968] for the end-term iterates. Hereafter, till the end of this section, we will continue to analyze $\bar{Y}_n$ further; exactly similar analysis also holds for $Y_n$, which we do not present separately to maintain continuity.

### 2.2.1 Estimating $\Sigma_n$

In Theorem 2.1, the `local` SGD updates are scaled by the matrix $\Sigma_n$, which is not usually known or estimable. This matrix originates as the covariance of the sum of independent vectors $u_s$, which acts as a linearized version of the updates $\bar{Y}_t = K^{-1} \sum_{k=1}^K \theta_t^k$. If $S = K \text{Var}(\sum_{k=1}^K w_k g_k(\theta_K^\star, \xi^k))$, then it can be shown that $K\Sigma_n \to \Sigma$ for $\Sigma = A^{-1}SA^{-\top}$ as $n \to \infty$. In general, we show the following theorem, proved in appendix Section C.

**Theorem 2.3.** *Under the assumptions of Theorem 2.1, it holds that*

$$|\Sigma_n - K^{-1}\Sigma|_F \lesssim K^{-1/2}n^{\beta-1}, \tag{2.7}$$

*and consequently, it holds that, with $Z' \sim N(0, K^{-1}\Sigma)$,*

$$d_{\mathrm{C}}(\sqrt{n}(\bar{Y}_n - \theta_K^\star), Z') \lesssim \sqrt{K}(n^{1/2-\beta} + n^{\beta-1}). \tag{2.8}$$

If $K = O(1)$, Theorem 2.3 reduces to Lemma 1 of Sheshukova et al. [2025].

*Remark* 2.2 (Computation-communication trade-off). Theorems 2.1 and 2.3 reveal a phase transition between classical central limit theory and the Berry-Esseen rate, which isn't clear from the condition $K = o(n^{2\beta-1})$ alone. This transition arises from a computation-communication trade-off Tsiatsis et al. [2005], Le Ny and Pappas [2013], Dieuleveut and Patel [2019], Ballotta et al. [2020], which, in our context, reflects a trade-off between the step-size parameter $\beta$ and the number of clients $K$. Specifically, if $K = o(n^c)$ for some $0 \le c \le 1/2$, the optimal $\beta_0 \in (1/2, 1)$ minimizing (2.8) is $\beta_0 = 3/4$. Conversely, if $K \gtrsim n^c$ for $c > 1/2$, no $\beta \in (1/2, 1)$ ensures that $\sqrt{K}(n^{1/2-\beta}+n^{\beta-1}) \to 0$. This implies that when $K \asymp n^c$ for some $c > 1/2$, the Kolmogorov error remains significant, regardless of the step-size, even though central limit theory still holds for $\beta \in (1/2 + c/2, 1)$. This phase transition highlights a new theoretical insight into the hardness of `local` SGD as $K$ increases.

An one pass estimation of $\Sigma$ is discussed in Gu and Chen [2024]. Additionally, in our appendix Section B.3, we point towards a new direction of multiplier bootstrap, leveraging our Berry-Esseen result, that does not require covariance estimation.

## 3 A time-uniform Gaussian coupling for the DFL updates

Section 2 quantifies the Gaussian approximation of the final `local` SGD updates $\bar{Y}_n$, with an error of order $\sqrt{n}$ in terms of iterations. However, maintaining privacy in a federated setting requires one to draw sharp inferences not only on the final output but on the entire `local` SGD trajectory, particularly for detecting model poisoning or adversarial attacks. From a theoretical standpoint, when the Assumption A.3 guarantees the existence of moments $p > 4$ (for example, when the data may be close to Gaussian), we should be able to derive sharper bounds on approximation errors, beyond the $\sqrt{n}$ result in Section 2. Since central-limit theory and Berry-Esseen estimates rely on fourth moments, we turn to classical strong approximation theory to exploit higher moments for precise bounds on the entire trajectory.

### 3.1 Motivation and Applications

A time-uniform Gaussian coupling for the entire `local` SGD updates has strong practical motivations, particularly for anomaly detection in "Internet-of-Vehicles" (IoV) Shalev-Shwartz et al. [2017], Ghimire and Rawat [2024], Zhu et al. [2024]. Assume that at some time point $t_0 \in [n]$, a subset of clients $\mathcal{K}_0 \subseteq [K]$ becomes malicious. This model poisoning can be mathematically described by a change in their local risk functions $F_k, k \in \mathcal{K}_0$, which affects the distribution of the `local` SGD updates $Y_t$. This perspective extends to other attacks, such as *LIE* (Little is Enough) or *MITM* (Man in the middle) Shen et al. [2016], Blanchard et al. [2017], Yin et al. [2018], Baruch et al. [2019], where an adversary injects noise or perturbs communication at time $t_0$, disrupting the distribution and trajectory of $Y_t$ for $t \ge t_0$ (Ding et al. [2024]). Methods offering explicit theoretical guarantees on precisely detecting attack initiation are rare; most of the literature concentrates around robustness guarantees (error bounds, convergence rates) assuming a certain adversarial profile or detection of malicious clients [Blanchard et al., 2017, Wang et al., 2020a, Qian et al., 2024], rather than provably devising poisoning alarm. Relatedly, Mapakshi et al. [2025] observed that attacks starting in later rounds can be more damaging compared to those present from the start.

Assume that at some time point $t_0 \in [n]$, a subset of clients $\mathcal{K}_0 \subseteq [K]$ becomes malicious. This model poisoning can be mathematically described by a change in their local risk functions $F_k, k \in \mathcal{K}_0$, which affects the distribution of the `local` SGD updates $Y_t$. To identify the time-point $t_0$ sequentially, we examine a CUSUM-type statistic $R_t := \max_{1 \leq s \leq t} s|\bar{Y}_s - \bar{Y}_t|$, widely used in change-point analysis. We expect $R_t$ to be large for $t > t_0$ if an attack has altered the mean behavior of the `local` SGD updates at $t_0$. The null distribution (i.e. when no attack takes place) of $R_t$ is usually mathematically intractable, hence posing a hindrance to performing valid inference. This necessitates a bootstrap procedure.

To identify the time-point $t_0$ sequentially, we examine a CUSUM-type statistic $R_t := \max_{1 \leq s \leq t} s|\bar{Y}_s - \bar{Y}_t|$, widely used in change-point analysis. We expect $R_t$ to be large for $t > t_0$ if an attack has altered the mean behavior of the `local` SGD updates at $t_0$.

Suppose there exists a Gaussian process $\mathcal{G}_t$ such that a time-uniform approximation holds:

$$\max_{1 \leq t \leq n} |t\bar{Y}_t - \mathcal{G}_t| = o_{\mathbb{P}}(\sqrt{n}). \tag{3.1}$$

Let $R_t^G = \max_{1 \leq s \leq t} |\mathcal{G}_s - \frac{s}{t}\mathcal{G}_t|$ Then it follows that,

$$
\begin{aligned}
n^{-1/2} \max_{1 \leq t \leq n} |R_t - R_t^G| &\leq n^{-1/2} \max_{1 \leq t \leq n} \max_{1 \leq s \leq t} |(s\bar{Y}_s - s\theta_K^\star - \mathcal{G}_s) - \frac{s}{t}(t\bar{Y}_t - t\theta_K^\star - \mathcal{G}_t)| \\
&\leq 2n^{-1/2} \max_{1 \leq t \leq n} |t\bar{Y}_t - t\theta_K^\star - \mathcal{G}_t| \\
&= o_{\mathbb{P}}(1). \tag{3.2}
\end{aligned}
$$

Equation (3.2) immediately suggests using Gaussian multiplier bootstrap leveraging $\mathcal{G}_t$ with precisely quantifiable approximation error. In particular, if $Q_{1-\alpha}(X)$ denotes the $(1-\alpha)$-th quantile of random variable $X$, then for a suitable positive sequence $\{a_n\}$,

$$\mathbb{P}(R_t > Q_{1-\alpha}(R_t^G) + a_n \text{ for some } t \in [n]) \leq \alpha + \mathbb{P}(\max_{1 \leq t \leq n} |R_t - R_t^G| > a_n) \to \alpha, \tag{3.3}$$

as long as $n^{-1/2}a_n \geq c$. We provide more details on these bootstrap algorithms in Appendix Section G. The two major questions that remain, are

- Does such a $\mathcal{G}_t$ exist? If yes, can we get a rate $\tau_{n,K}$ such that $\tau_{n,K} \ll \sqrt{n}$?
- Can we explicitly characterize its covariance structure, so as to enable bootstrap sampling?

The main results in Section 3.2 provide answers to both the questions above.

## 3.2 Optimal coupling for `local` SGD

The following theorem, proved in Section D.1, establishes a Gaussian approximation echoing (3.1).

**Theorem 3.1.** *For $W^k := g_k(\theta_K^\star, \xi^k)$, $\xi^k \sim \mathcal{P}_k$ independently for $k \in [K]$, let $\mathcal{V}_K = \mathrm{Var}(\sum_{k=1}^K w_k W_k)$. Suppose Assumption A.3 holds for a general $p \geq 2$. Then, under Assumptions A.1, A.2 and 2.1, (on a possibly richer probability space) there exists $Z_1, \ldots, Z_n \overset{i.i.d.}{\sim} N(0, K\mathcal{V}_K)$, such that with*

$$Y_{t,1}^G = (I - \eta_t A)Y_{t-1,1}^G + \eta_t Z_t K^{-1/2}, \ Y_{0,1}^G = \mathbf{0}, \tag{3.4}$$

*it holds that,*

$$\max_{1 \leq t \leq n} |\sum_{s=1}^t (Y_s - \theta_K^\star - Y_{s,1}^G)| = O_{\mathbb{P}}(n^{1-\beta}) + o_{\mathbb{P}}(n^{1/p}K^{-1/2}\log n). \tag{3.5}$$

We call the Gaussian approximation iterates (3.5) "Aggregated Gaussian approximation"(`Aggr-GA`). Note that `Aggr-GA` requires a complete sharing of the covariance structure to construct $\mathcal{V}_K$, which may affect privacy at inference-time. However, it turns out that one can further refine Theorem 3.1 to provide another Gaussian approximation results that exactly mimics the `local` SGD updates in their use of local structure along with periodic sharing. We call this latter approximation by `Client-GA`.

**Theorem 3.2.** *Under the assumptions of Theorem 3.1, on a possibly richer probability space, for each $k \in [K]$, there exist $Z_1^k, \ldots, Z_n^k \overset{i.i.d.}{\sim} N(0, \mathrm{Var}(W^k))$, such that with*

$$\tilde{\Theta}_t^G = ((I - \eta_t A)\tilde{\Theta}_{t-1}^G + \eta_t M_t)C_t, \ \tilde{\Theta}_0^G = (\mathbf{0}, \ldots, \mathbf{0}), \tag{3.6}$$

where $M_t := K(w_1 Z_t^1, \dots, w_K Z_t^K) \in \mathbb{R}^{d \times K}$, and $C_t$ as in (2.2), it holds that

$$\max_{1 \le t \le n} |\sum_{s=1}^{t} (Y_s - \theta_K^\star - Y_{s,2}^G)| = O_\mathbb{P}(n^{1-\beta} + (n/K)^{\frac{1}{4} + \frac{1}{2p}} (\log n)^{3/2}), \quad Y_{t,2}^G = K^{-1} \tilde{\Theta}_t^G \mathbf{1}. \quad (3.7)$$

Note that for $p = 2$, the rates of Theorems 3.1 and 3.2 coincide. Theorem 2.2 is proved in Section D.2. In both the results, $n^{1-\beta}$ reflects the fundamental error of a generic uniform Gaussian approximation for the `local SGD` updates $Y_n$, and as such, does not depend on $K$. The second error decreases with the number of clients, as an increasing number of clients enables `local SGD` updates to track a larger horizon, and the corresponding client-averaged $Y_t$ becomes more concentrated in their trajectory towards $\theta_K^\star$, leading to sharper approximations.

*Remark* 3.1 (Computational differences between `Aggr-GA` and `Client-GA`). At each iteration $t$, `Aggr-GA` has a computational complexity of $O(d^2)$, since it involves generating one random sample followed by a matrix-vector multiplication. In contrast, `Client-GA` has a complexity of $O(Kd^2)$ per iteration. Importantly, the structure of `Client-GA` naturally allows for parallel computation between synchronization steps, significantly reducing the computational burden while preserving periodic peer-to-peer communication.

*Remark* 3.2 (Difference with functional CLT). Li et al. [2022] proved a functional CLT for `local SGD` when the number of clients $K$ is fixed. Although such a result can theoretically be extended to the general setting considered here, nevertheless our approximations (3.5) and (3.7) are much sharper than a functional CLT approximation. As a toy example, consider the vanilla SGD setting, i.e. `local SGD` with $\tau = 1$, and suppose $K = 1$. Suppose $F(\theta) = (\theta - \mu)^2/2$, and $\nabla f(\theta, \xi) := \theta - \mu + \xi$. In this setting, both `Aggr-GA` and `Client-GA` collapse to the same Gaussian approximation

$$Y_t^G = (I - \eta_t A) Y_{t-1}^G + \eta_t Z_t, \; Z_t \sim N(\mathbf{0}, \text{Var}(\xi)), \; Y_0^G = \mathbf{0}. \quad (3.8)$$

Here $A = \nabla_2 F(\mu) = I$. On the other hand, the vanilla SGD iterates can also be seen as $Y_t - \mu = (I - \eta_t A)(Y_{t-1} - \mu) + \eta_t \xi_t$. Therefore, it can be seen that $Y_t - \mu$ and $Y_t^G$ have exactly the same covariance structure, i.e. $\text{Cov}(Y_s^G, Y_t^G) = \text{Cov}(Y_s, Y_t)$; on the other hand, even in such a simplified setting, an approximation by Brownian motion, such as that by functional CLT, captures the covariance structure of the iterates $\{Y_t - \mu\}_{t \ge 1}$ only in an asymptotic sense. The Gaussian approximation $Y_t^G$ in (3.8) is a particular example of covariance-matching approximations, introduced by Bonnerjee et al. [2024]. By extension, same intuition holds for `Aggr-GA` and `Client-GA` as well. However, at this point, we note that the covariance-matching approximations in Bonnerjee et al. [2024] were for short-range, univariate non-stationary process. On the other hand, in the `local SGD` setting, the polynomially decaying step-size introduces a non-stationarity that can possibly be long-range dependent. Moreover, our result allows for multivariate parameters in a direct generalization of these aforementioned, covariance-matching approximations. We empirically validate this in Section 4.

Note that $n^{1-\beta}$ indicates the fixed error for the `local SGD` updates with step-sizes $\eta_t \asymp t^{-\beta}$, and in order to completely underpin the effect of the assumption of additional moments $p > 2$, an optimal choice of step-size must be given so that $n^{1-\beta}$ becomes negligible compared to the second error term involving the moment $p$. This choice is indicated in the following proposition.

**Proposition 1.** *Grant the assumptions of Theorems 3.1 and 3.2, and consider the Gaussian approximations $Y_{s,1}^G$ and $Y_{s,2}^G$ defined therein. Suppose $K = o(n^c)$ for some $c \in (0, 1)$.*

*(i) If $c < 2/p$, then $\beta \ge 1 - 1/p + c/2$ ensures $\max_{1 \le t \le n} |\sum_{s=1}^{t} (Y_s - \theta_K^\star - Y_{s,1}^G)| = o_\mathbb{P}(n^{\frac{1}{p} - \frac{c}{2}} \log n)$.*

*(ii) For a general $c \in (0, 1)$, a choice of $\beta > 1 - (1-c)(\frac{1}{4} + \frac{1}{2p})$ ensures that $\max_{1 \le t \le n} |\sum_{s=1}^{t} (Y_s - \theta_K^\star - Y_{s,2}^G)| = o_\mathbb{P}(n^{(1-c)(\frac{1}{4} + \frac{1}{2p})} (\log n)^{3/2})$.*

Cases (i) and (ii) reveal a trade-off between `Aggr-GA` and `Client-GA`. While `Aggr-GA` requires information sharing at each step, yielding better approximation, it demands a stricter choice of $\beta$, since $1 - \frac{1}{p} + \frac{c}{2} > 1 - (1-c)(\frac{1}{4} + \frac{1}{2p})$ for all $p > 2, c > 0$. In contrast, `Client-GA`'s local operation supports $K = o(n)$ clients, aligning with Zhang et al. [2013], Gu and Chen [2023]. Both methods require known Hessians and local covariances, estimable efficiently via Gu and Chen [2024].

# 4    Simulation results

Here, we summarize the various empirical exercises to accompany our theory in Sections 2 and 3. In particular, in Section 4.1, we discuss the Berry-Esseen error $d_{\mathrm{C}}(\sqrt{n}(\bar{Y}_n - \theta_K^\star), Z)$ for $Z \sim N(0, \Sigma_n)$ with varying choices of the number of iterations $N$, number of clients $K$ and synchronization parameter $\tau$. In Section 4.2, we numerically explore the computation-communication trade-off discussed in Section 2.2. Finally, in Section 4.3, we explore the approximation error of `Aggr-GA` and `Client-GA` via Q-Q plots. Detailed explanations, and additional experiments, along with the model specifications, can be found in Appendix Section F. All codes are available in github.

## 4.1    Effect of $n$ and $K$ on the Berry-Esseen rate

As a proxy of $d_{\mathrm{C}}$, we consider $\tilde{d}_c = \sup_{x \in [0,c]} \left| \mathbb{P}(|\sqrt{n}\Sigma_n^{-1/2}(\bar{Y}_n - \theta_K^\star)| \leq x) - \mathbb{P}(|Z| \leq x) \right|$ where $Z \sim N(0, I)$ for a large enough $c > 0$. Figure 1 shows how $\tilde{d}_c$ varies with varying $n, K, \tau$ when the step-size is kept fixed at $\eta_t = 0.3t^{-0.75}$. In particular, $\tilde{d}_c$ decays with $N$ for fixed $K$, and increases with $K$ for fixed $n$. Additional simulations and further insights can be found in Appendix section F.1.

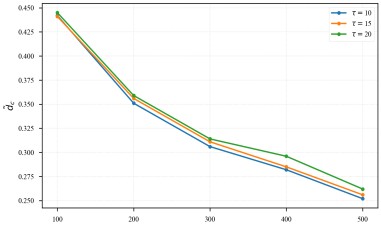 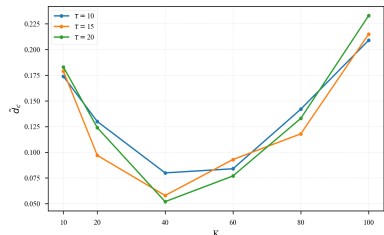

Figure 1: Plot of $\tilde{d}_c$ against $n$ and $K$ for $\gamma = 1$, and Settings 1(left), and 2(right).

## 4.2    Computation-communication trade-off

Here, we fix $n \in \{100, 200, 300, 400, 500\}$, and $K = \lfloor n^r \rfloor$ for $r \in \{0.2, 0.6\}$ and numerically investigate the computation-communication trade-off hinted at in Remark 2.2. We run the `local SGD` algorithm with $\tau = 5$, and $\eta_t = 0.5t^{-\beta}$, for $\beta \in \{0.85, 0.9, 0.95\}$. Clearly, $\tilde{d}_c$ decays with $n$ for $r = 0.2$, and increases with $n$ for $r = 0.6$, exemplifying our assertion about the computation-communication trade-off between $K$ and $\beta$. Appendix Section F.2 contains additional details.

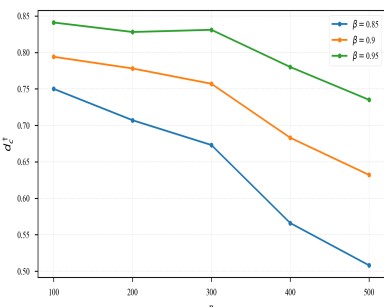 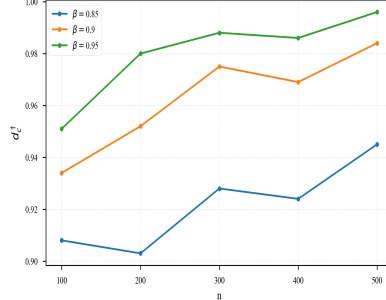

Figure 2: Plot of $d_c^\dagger$ against $(n, \beta)$ for $r = 0.2$ (left), and $r = 0.6$ (right). Here $\gamma = 0$.

## 4.3    Performance of the time-uniform Gaussian approximations

In this section, we fix $N = 500$, $\tau = 20$, and let $K \in \{10, 25, 50\}$, and compare the quantiles of the maximum partial sums of `local SGD` $U_n$, `Aggr-GA` $U_n^{\texttt{Aggr-GA}}$, `Client-GA` $U_n^{\texttt{Client-GA}}$ and approximation by Brownian motion: $U_n^{\texttt{f-CLT}}$ . Clearly, `Aggr-GA` seems to be performing the best, as suggested by Theorems 3.1 and 3.2. Furthermore, $U_n^{\texttt{f-CLT}}$ consistently has the worst approximation. Additional details can be found in appendix Section F.6.

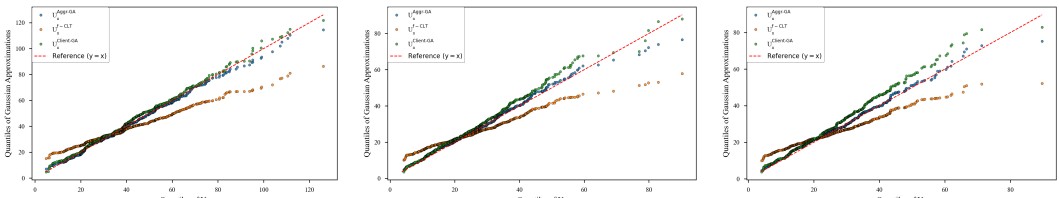

Figure 3: QQ-plots of $U_n^{\texttt{Aggr-GA}}$ (blue), $U_n^{\texttt{Client-GA}}$ (green) and $U_n^{\texttt{f-CLT}}$ (orange) against $U_n$ for $\gamma = 1$, $N = 500, \tau = 20$. Here $K = 10$(left), $K = 25$(middle), $K = 50$(right). Rest of the FRand-eff model specifications are as in Section F.1.1.

## 5  Conclusion

Sharper theoretical results beyond the central limit theorem is extremely crucial to perform valid and powerful statistical inference, yet such results have not previously appeared in the literature for `local` SGD and in general, decentralized federated learning. In this context, to the best of our knowledge, this is the first work deriving Berry-Esseen bounds as well as sharp time-uniform Gaussian approximations over the `local` SGD trajectory. These results enable the development of valid and powerful statistical inference methods, including bootstrap procedures Fang et al. [2018], Fang [2019], Zhong et al. [2023], which can be adapted to decentralized settings. The technical framework developed herein offers a pathway to sharper results in many other related settings including multi-agent systems and transfer learning Duan and Wang [2023], Pan et al. [2023], Knight and Duan [2023], Lin and Reimherr [2024]. It is also crucial to make explicit the effect of synchronization in the derived rates, which can reflect more trade-offs and constitute a suitable future work.

## 6  Acknowledgment

The authors would like to thank the reviewers, the area chair and the senior area chair for their constructive feedbacks that helped improve the paper significantly. SK and WBW thank NSF DMS grant 2124222 and NSF DMS grant 2311249 respectively for supporting their research.

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

# A  Technical assumptions and Some comments on Section 2

## A.1  Technical Assumptions

For the validity of our theoretical results, we require some mild regularity conditions on the loss functions as well as the noise level of each client. We remark, that the following assumptions have appeared extensively in the theoretical analysis of iterative convex optimization algorithms, and here, are merely adapted to the federated learning setting.

**Assumption A.1** (Strong convexity). *There exists $\mu > 0$ such that for each $k \in [K]$,*

$$\langle \nabla F_k(\theta) - \nabla F_k(\theta'), \theta - \theta' \rangle \geq \mu |\theta - \theta'|^2, \ \theta, \theta' \in \mathbb{R}^d.$$

Assumption of strong convexity is common in the analysis of SGD iterates, appearing in Ruppert [1988], Polyak and Juditsky [1992], Bottou et al. [2018], Chen et al. [2020], and as such, Assumption A.1 adapts this condition to the decentralized setting.

**Assumption A.2.** *(Stochastic Lipschitzness of noisy gradients) There exists $L > 0$ such that for each $k \in [K]$,*

$$\mathbb{E}_{\mathcal{P}_k}[|\nabla f_k(\theta, \xi^k) - f_k(\theta', \xi^k)|^2] \leq L|\theta - \theta'|^2, \ \theta, \theta' \in \mathbb{R}^d.$$

Assumption A.2 combines the $L$-smoothness condition on the risk functions $F_k$, with a stochastic Lipschitz condition on the gradient noise vectors $g_k(\theta, \xi^k) = \nabla F_k(\theta) - \nabla f_k(\theta, \xi^k)$.

**Assumption A.3.** *(Control on noisy gradients) The functions $f_k(\theta, \xi)$ is assumed to be continuously differentiable with respect to $\theta$ for any fixed $\xi$. Moreover, assume that $\max_{k \in [K]} \mathbb{E}[|g_k(\theta, \xi^k)|^p] < \infty$ for some $p \geq 2$.*

Assumption A.3 ensures that Newton-Leibnitz's integration rule holds and consequently, $\sum_{k=1}^K w_k g_k(\theta_t^k, \xi_t^k)$ constitutes a martingale difference sequence adapted to the filtration $\sigma(\Xi_s : s \leq t)$, where $\Xi_s = (\xi_s^1, \dots, \xi_s^K)$. Moreover, Assumptions A.2 and A.3 jointly imply that there exists a constant $L_Q$ such that for all $\theta \in \mathbb{R}^d$

$$\max_{k \in K} |\nabla F_k(\theta) - \nabla_2 F(\theta_K^\star)(\theta - \theta_K^\star)| \leq L_Q |\theta - \theta_K^\star|^2. \tag{A.1}$$

See Lemma 5 of Sheshukova et al. [2025]. The assumptions A.2 and A.3 are fairly ubiquitous in the stochastic optimization literature Zhu et al. [2023], Wei et al. [2023], Li et al. [2024]. In particular, assumption A.2 is much weaker than the corresponding Assumption $A2(p) - (ii)$ in Sheshukova et al. [2025].

## A.2  Is strong-convexity Assumption A.1 necessary?

It is important to note that Assumption A.1 fails to hold in certain M-estimation problems including logistic regression Bach [2010]. Gu and Chen [2024] addressed this issue by invoking a weaker local strong-convexity assumption, also known as the "local concordance" condition.

**Assumption A.4** (Local strong concordance). *There exists $\mu^\star > 0$ such that $\nabla_2 F(\theta_K^\star) \succeq \mu^\star$. Moreover, there exists a constant $C > 0$, and compact set $\Phi \subseteq \mathbb{R}^d$, such that for all $\theta_1, \theta_2 \in \Phi$, it holds that*

$$|\varphi'''(u)| \leq C |\theta_1 - \theta_2| \varphi''(u), \ where \ \varphi : u \mapsto F(\theta_1 + u(\theta_2 - \theta_1)), u \in \mathbb{R}.$$

In view of Assumption A.4, for theoretical validity of our results, one requires a projected `local` SGD updates $\boldsymbol{\Theta}_t = M_\Phi((\boldsymbol{\Theta}_{t-1} - \mathbf{G}_t)C_t)$ instead of (2.2), where $M_\Phi$ denotes the projection operator on the set compact $\Phi$. The key difference in the treatment of Assumption A.4 compared to that of Assumption A.1 lies in the analysis of the term $|\theta - \eta \nabla F(\theta)|^2$ for some small enough $\eta > 0$. In particular, a recurring theme of our proofs is to show that

$$|\theta - \theta_K^\star - \eta \nabla F(\theta)|^2 \leq (1 - \eta c)|\theta - \theta_K^\star|^2 \ \text{for some } c > 0, \theta \in \mathbb{R}^d. \tag{A.2}$$

We highlight the different arguments leading up-to (A.2), leveraging Assumptions A.1 and A.4 respectively.

### A.2.1 Proof of (A.2) via Assumptions A.1 and A.2

Note that

$$
\begin{aligned}
|\theta - \theta_K^\star - \eta \nabla F(\theta)|^2 =& |\theta|^2 - 2\eta(\theta - \theta_K^\star)^\top \nabla F(\theta) + \eta^2 |\nabla F(\theta)|^2 \\
\leq& (1 - 2\eta\mu + \eta^2 L^2)|\theta - \theta_K^\star|^2,
\end{aligned}
\tag{A.3}
$$

and hence, (A.2) is inferred by choosing $\eta$ to be small enough. In particular, since we work with decaying step size $\eta_t \propto t^{-\beta}$, it follows that $1 - 2\eta_t\mu + \eta_t^2 L^2 \leq 1 - \eta_t c$ for some $c > 0$ and all sufficiently large $t \in \mathbb{N}$.

### A.2.2 Proof of (A.2) via Assumptions A.4, A.2 and $|x| \leq R$

Fix $\theta \in \mathbb{R}^d$, and choose $\phi(u) = F(\theta_K^\star + u(\theta - \theta_K^\star))$, $u \in [0, 1]$. Note that $\phi''(0) \geq \mu^\star |\theta - \theta_K^\star|^2$. From Assumption A.4, one directly has

$$
\phi''(u) \geq \phi''(0) \exp(-C|\theta - \theta_K^\star|u),
$$

and therefore, recalling $|x| \geq R$

$$
\begin{aligned}
(\theta - \theta_K^\star)^\top \nabla F(\theta) =& \phi'(1) - \phi'(0) \\
\geq& \mu^\star |\theta - \theta_K^\star|^2 \int_0^1 \exp(-C|\theta - \theta_K^\star|u)\mathrm{d}u \\
=& \mu^\star |\theta - \theta_K^\star|^2 \frac{1 - \exp(-C|\theta - \theta_K^\star|)}{C|\theta - \theta_K^\star|} \\
\geq& \mu^\star C \exp(-R)|\theta - \theta_K^\star|^2,
\end{aligned}
\tag{A.4}
$$

which immediately can be applied to (A.3) to deduce (A.2).

In view of the analysis in Sections A.2.1 and A.2.2 coming to the same conclusion, for the sake of simplicity, our subsequent theoretical findings are stated and proved using Assumption A.1 only. We remark that an accompanying result invoking Assumption A.4 and the projected `local SGD` updates can easily be obtained via minor modifications of our arguments following Section A.2.2. For a more detailed discussion on the implications of Assumption A.4, we refer the interested readers to Assumption 3.4 and the associated remark in Gu and Chen [2024].

### A.3 A comment on step-size

Our choice of the step-size is motivated from the extensive literature of asymptotics of various stochastic approximation algorithm. In particular, it is well-known that SGD with a constant step-size is asymptotically biased Dieuleveut et al. [2020], Li et al. [2024], Glasgow et al. [2022], whereas central limit theory based on polynomially decaying schedule $\eta_t \propto t^{-\beta}$, $\beta \in (1/2, 1)$ has an extensive literature for different algorithms. In practice, often a combination of the two kinds of step-size is used, where a constant-step size algorithm provides a warm start, and after discarding initial few iterates pre-specified by the fixed *burn-in* period $k_0$, `local SGD` can be run with the polynomially decaying step-size to ensure appropriate convergence. This is tantamount to the step-size choice $\eta_t = \eta_0(t - k_0)^{-\beta}$, $t > k_0$, which is also covered by our theory.

## B Proof of Theorems 2.1 and 2.2

In this section we rigorously derive the Berry-Esseen bounds on $\bar{Y}_n$ and $Y_n$, as stated in Theorems 2.1 and 2.2 respectively. Similar to the simpler analysis for stochastic gradient descent in Samsonov et al. [2024], we aim to leverage Theorem 2.1 of Shao and Zhang [2022]. However, the regular synchronization step, as well as the general connection matrix $\mathbf{C}$, induces some significant non-triviality in the problem, requiring, in particular, careful analysis of the difference between client-wise estimates and the aggregated estimate. Before we delve deeper into the mathematical details, we summarize the road-map to prove Theorem 2.1 below.

- In Section B.1.1, we decompose the `local SGD` updates into a linear component and the remainder terms.

- In Section B.1.2, we echo the Lindeberg method, and define a coupling for the remainder terms. In particular, our choice of the coupling is novel, and rooted into the uniqueness of the decentralized setting.

- Finally, in Sections B.1.3-B.1.7, we control the different terms arising out of the application of the abstract Theorem 2.1 of Shao and Zhang [2022] to the steps above. We remark that this is where our treatment diverges from the preceding works proving Berry-Esseen in a stochastic approximation framework. To accommodate an increasing number of clients $K$ as well as to control the error of the each local client-level iterates, we derive and apply the Auxiliary results 4-8.

The proof for Theorem 2.2 will follow a similar structure.

## B.1 Proof of Theorem 2.1

### B.1.1 Linearization

Noting that $Y_t = K^{-1} R_t \mathbf{1}$ no matter if $t \in E_\tau$ or $t \notin E_\tau$, it is easy to observe

$$Y_t = Y_{t-1} - \eta_t \sum_{k=1}^K w_k \nabla f_k(\theta_{t-1}^k, \xi_t^k), \ t \in [n], \ Y_0 = K^{-1} \sum_{k=1}^K \theta_0^k. \tag{B.1}$$

Write (B.1) as follows:

$$Y_t - \theta_K^\star = Y_{t-1} - \theta_K^\star - \eta_t \nabla F(Y_{t-1}) + \eta_t \sum_{k=1}^K w_k (\nabla F_k(Y_{t-1}) - \nabla F_k(\theta_{t-1}^k)) + \eta_t \sum_{k=1}^K w_k g_k(\theta_{t-1}^k, \xi_t^k)$$

$$= (I - \eta_t A)(Y_{t-1} - \theta_K^\star) + \eta_t \big( A(Y_{t-1} - \theta_K^\star) - \nabla F(Y_{t-1}) \big)$$

$$+ \eta_t \sum_{k=1}^K w_k (\nabla F_k(Y_{t-1}) - \nabla F_k(\theta_{t-1}^k)) + \eta_t \sum_{k=1}^K w_k g_k(\theta_{t-1}^k, \xi_t^k), \tag{B.2}$$

where $g_k(\theta, \xi) := \nabla F_k(\theta) - \nabla f_k(\theta, \xi)$ denote the gradient noise. Denote $\mathcal{A}_s^t = \prod_{j=s+1}^t (I - \eta_j A)$, $\mathcal{A}_t^t = I$ with $A := \nabla_2 F(\theta_K^\star)$, and define $Q_s = \eta_s \sum_{j=s}^n \mathcal{A}_s^j$. Recursively, (B.2) can be simplified to

$$Y_t - \theta_K^\star = \mathcal{A}_0^t (Y_0 - \theta_K^\star) + \sum_{s=1}^t \eta_s \mathcal{A}_s^t \Big( \big( A(Y_{s-1} - \theta_K^\star) - \nabla F(Y_{s-1}) \big) +$$

$$+ \sum_{k=1}^K w_k (\nabla F_k(Y_{s-1}) - \nabla F_k(\theta_{s-1}^k)) + \sum_{k=1}^K w_k g_k(\theta_{s-1}^k, \xi_s^k) \Big), \tag{B.3}$$

which immediately yields,

$$\bar{Y}_n - \theta_K^\star = n^{-1} \eta_0^{-1} Q_0 (Y_0 - \theta_K^\star) + n^{-1} \sum_{s=1}^n Q_s \mathcal{N}_s + n^{-1} \sum_{s=1}^n Q_s \Big( \big( A(Y_{s-1} - \theta_K^\star) - \nabla F(Y_{s-1}) \big)$$

$$+ \sum_{k=1}^K w_k (\nabla F_k(Y_{s-1}) - \nabla F_k(\theta_{s-1}^k)) + \sum_{k=1}^K w_k \big( g_k(\theta_{s-1}^k, \xi_s^k) - g_k(\theta_K^\star, \xi_s^k) \big) \Big), \tag{B.4}$$

where we define that $\mathcal{N}_t = \sum_{k=1}^K w_k W_t^k$, with $W_t^k = g_k(\theta_K^\star, \xi_t^k)$. Let $H = n^{-1/2} \sum_{s=1}^n Q_s \mathcal{N}_s$, and let $\Sigma_n = \mathbb{E}[HH^\top]$. Then, (B.4) can be re-written as

$$\sqrt{n} \Sigma_n^{-1/2} (\bar{Y}_n - \theta_K^\star) = W + D_1 + D_2 + D_3 + D_4, \tag{B.5}$$

where

$$W = \Sigma_n^{-1/2} H = \sum_{s=1}^n u_s, \text{ where } u_s = \frac{1}{\sqrt{n}} \Sigma_n^{-1/2} Q_s \mathcal{N}_s, \tag{B.6}$$

$$D_1 = \frac{1}{\sqrt{n}\eta_0} \Sigma_n^{-1/2} Q_0 (Y_0 - \theta_K^\star), \tag{B.7}$$

$$D_2 = \frac{1}{\sqrt{n}} \Sigma_n^{-1/2} \sum_{s=1}^n Q_s \big( A(Y_{s-1} - \theta_K^\star) - \nabla F(Y_{s-1}) \big), \tag{B.8}$$

$$D_3 = \frac{1}{\sqrt{n}} \Sigma_n^{-1/2} \sum_{s=1}^n Q_s \Big( \sum_{k=1}^K w_k (\nabla F_k(Y_{s-1}) - \nabla F_k(\theta_{s-1}^k)) \Big), \tag{B.9}$$

$$D_4 = \frac{1}{\sqrt{n}} \Sigma_n^{-1/2} \sum_{s=1}^n Q_s \Big( \sum_{k=1}^K w_k \big( g_k(\theta_{s-1}^k, \xi_s^k) - g_k(\theta_K^\star, \xi_s^k) \big) \Big). \tag{B.10}$$

### B.1.2 Definition of the Lindeberg Coupling

Note that

$$|D_3|_2 \le C \frac{b_2\sqrt{L}}{K\sqrt{n}} |\Sigma_n^{-1/2}|_F \sum_{s=1}^n \sum_{k=1}^K |Y_{s-1} - \theta_{s-1}^k|_2 \tag{B.11}$$

$$\le C \frac{b_2\sqrt{L}}{\sqrt{n}K} |\Sigma_n^{-1/2}|_F \sum_{s=1}^n \sqrt{\sum_{k=1}^K |Y_{s-1} - \theta_{s-1}^k|^2} \tag{B.12}$$

$$= C \frac{b_2\sqrt{L}}{\sqrt{nK}} |\Sigma_n^{-1/2}|_F \sum_{s=1}^n |\boldsymbol{\Theta}_s (I - J)|_F := \Delta_3. \tag{B.13}$$

In the above series of inequalities, (B.11) follows from $w_k \le b_2 K^{-1}$, $\max_s |Q_s|_F \le C$, and Assumption A.2; (B.12) is a trivial consequence of Cauchy-Schwarz inequality. Additionally, define $\Delta_l = |D_l|_2$ for $l = 1, 2, 4$. Let $\Xi_t = (\xi_t^1, \ldots, \xi_t^K)$, and for each $i \in [n]$, let us denote

$$\Xi_{t,\{i\}} = \begin{cases} \Xi_t, & t \ne i \\ \Xi_i' := (\xi_t^{1'}, \ldots, \xi_t^{K'}), & t = i, \end{cases}$$

where $\xi_t^{k'}, \xi_t^k \overset{i.i.d.}{\sim} \mathcal{P}_k$, $k \in [K], t \in [n]$. For each $i \in [n]$, define the coupled DFL iterates as

$$\boldsymbol{\Theta}_{t,\{i\}} = (\boldsymbol{\Theta}_{t-1,\{i\}} - \eta_t \mathbf{G}_{t,\{i\}}) C_t, \quad \boldsymbol{\Theta}_{0,\{i\}} = \boldsymbol{\Theta}_0, \tag{B.14}$$

where $\mathbf{G}_{t,\{i\}} = K(w_1 \nabla f_1(\theta_{t-1,\{i\}}^1, \xi_{t,\{i\}}^1), \ldots, w_K \nabla f_K(\theta_{t-1,\{i\}}^K, \xi_{t,\{i\}}^K))$. Let $Y_{t,\{i\}} = K^{-1} \boldsymbol{\Theta}_{t,\{i\}} \mathbf{1}$. Based on (B.14), we can define coupled versions of $\Delta_l$, $l = 2, 3, 4$ as follows:

$$\Delta_{2,\{i\}} = \frac{1}{\sqrt{n}} \Big| \Sigma_n^{-1/2} \sum_{s=1}^n Q_s \big( A(Y_{s-1,\{i\}} - \theta_K^\star) - \nabla F(Y_{s-1,\{i\}}) \big) \Big|, \tag{B.15}$$

$$\Delta_{3,\{i\}} = C \frac{b_2\sqrt{L}}{\sqrt{nK}} |\Sigma_n^{-1/2}|_F \sum_{s=1}^n \big| \Theta_{s,\{i\}}(I - J) \big|_F, \tag{B.16}$$

$$\Delta_{4,\{i\}} = \frac{1}{\sqrt{n}} \Big| \Sigma_n^{-1/2} \sum_{s=1}^n Q_s \Big( \sum_{k=1}^K w_k \big( g_k(\theta_{s-1,\{i\}}^k, \xi_{s,\{i\}}^k) - g_k(\theta_K^\star, \xi_{s,\{i\}}^k) \big) \Big) \Big|. \tag{B.17}$$

Note that $D_{1,\{i\}} = D_1$ for all $i \in [n]$. With these definitions, along with the fact that $\mathbb{E}[WW^\top] = I$ allows us to apply Shao and Zhang [2022], Theorem 2.1 on (B.5) to obtain

$$d_C(\sqrt{n}\Sigma_n^{-1/2}(\bar{Y}_n - \theta_K^\star), Z) \le c_1\sqrt{d}\Upsilon_n + \mathbb{E}[|W||\Delta_n|] + \sum_{i=1}^n \mathbb{E}[|u_i| |\Delta_n - \Delta_{n,\{i\}}|], \tag{B.18}$$

where $Z \sim N(0, I)$, $\Upsilon_n = \sum_{s=1}^n \mathbb{E}[|u_s|^3]$, $\Delta_n = \sum_{l=1}^4 |\Delta_l|$, and $\Delta_{n,\{i\}} = \sum_{l=1}^4 \Delta_{l,\{i\}}$.

### B.1.3  Bound on $\sum_{i=1}^{n} \mathbb{E}[|u_i| \, |\Delta_n - \Delta_{n,\{i\}}|]$

Recall that $\max_k |\mathrm{Var}[W_s^k]| = O(1)$. Clearly $\mathcal{N}_s$ are i.i.d. and $\mathbb{E}[\mathcal{N}_s \mathcal{N}_s^\top] = \sum_{k=1}^{K} w_k^2 \mathrm{Var}[W_s^k]$, which directly implies $|\Sigma_n| = O(K^{-1})$ in view of the fact $w_k \asymp K^{-1}$ for $k \in [K]$. Therefore, from (B.6) it follows $\mathbb{E}[|u_s|^2] = O(1/n)$, and consequently,

$$\sum_{i=1}^{n} \mathbb{E}[|u_i| \, |\Delta - \Delta_{n,\{i\}}|] \lesssim \frac{1}{\sqrt{n}} \sum_{l=2}^{4} \sum_{i=1}^{n} \sqrt{\mathbb{E}[|\Delta_l - \Delta_{l,\{i\}}|^2]}. \tag{B.19}$$

We will deal with the three terms in the right side of (B.19) one-by-one.

**Bound on** $\Delta_2 - \Delta_{2,\{i\}}$  We start with controlling $\mathbb{E}[|\Delta_2 - \Delta_{2,\{i\}}|^2]$. It is easy to see from (B.8) and (B.15) that

$$\mathbb{E}[|\Delta_2 - \Delta_{2,\{i\}}|^2] \lesssim \frac{K}{n} \mathbb{E}\left[\left|\sum_{s=i}^{n} \big(A(Y_s - Y_{s,\{i\}}) - \nabla F(Y_s) + \nabla F(Y_{s,\{i\}})\big)\right|^2\right]$$

$$\lesssim K \sum_{s=i}^{n} \mathbb{E}[|Y_s - Y_{s,\{i\}}|^4] = O(Kn^{1-4\beta} - Ki^{1-4\beta}), \tag{B.20}$$

where (B.20) follows from Cauchy-Schwarz inequality and Proposition 8.

### B.1.4  Bound on $\Delta_3 - \Delta_{3,\{i\}}$

Note that, since $\boldsymbol{\Theta}_{t,\{i\}} = \boldsymbol{\Theta}_t$ for all $t < i$, hence we must have

$$\mathbb{E}[|\Delta_3 - \Delta_{3,\{i\}}|^2] \lesssim \frac{1}{n} \mathbb{E}\left[\left(\sum_{s=i}^{n} \big|(\boldsymbol{\Theta}_s - \boldsymbol{\Theta}_{s,\{i\}})(I - J)\big|_F\right)^2\right]$$

$$\leq \sum_{s=i}^{n} \mathbb{E}[\big|(\boldsymbol{\Theta}_s - \boldsymbol{\Theta}_{s,\{i\}})(I - J)\big|_F^2]$$

$$= \mathbb{E}[\big|(\boldsymbol{\Theta}_i - \boldsymbol{\Theta}_{i,\{i\}})(I - J)\big|_F^2] + \sum_{s=i+1}^{n} \mathbb{E}[\big|(\boldsymbol{\Theta}_s - \boldsymbol{\Theta}_{s,\{i\}})(I - J)\big|_F^2]. \tag{B.21}$$

Note that,

$$\mathbb{E}[\big|(\boldsymbol{\Theta}_i - \boldsymbol{\Theta}_{i,\{i\}})(I - J)\big|_F^2] = \eta_i^2 \mathbb{E}[\big|(\mathbf{G}_s - \mathbf{G}_{s,\{i\}})(C_i - J)\big|_F^2]$$

$$\leq \eta_i^2 \mathbb{E}[|\sum_{k=1}^{K} w_k \big(g_k(\theta_{i-1}^k, \xi_i^k) - g_k(\theta_{i-1}^k, \xi_i^{k'})\big)|^2]$$

$$\leq 2\eta_i^2 \mathbb{E}[|\sum_{k=1}^{K} w_k g_k(\theta_{i-1}^k, \xi_i^k)|^2] = O\left(\frac{\eta_i^2}{K}\right). \tag{B.22}$$

Hence, Proposition 5 and (B.22) simultaneously imply via (B.21) that

$$\mathbb{E}[|\Delta_3 - \Delta_{3,\{i\}}|^2] \lesssim \frac{\eta_i^2}{K} + K \sum_{s=i+1}^{n} \eta_s^4 = O(i^{-2\beta} K^{-1} + Kn^{1-4\beta} - Ki^{1-4\beta}). \tag{B.23}$$

### B.1.5  Bound on $\Delta_4 - \Delta_{4,\{i\}}$

This term is the simplest to deal with. In view of the facts (i) $\sum_{k=1}^{K} w_k \big(g_k(\theta_{t-1}^k, \xi_t^k) - g_k(\theta_{t-1,\{i\}}^k, \xi_t^k)\big)$ is a martingale difference sequence adapted to the filtration $\mathcal{F}_t = \sigma(\Xi_s : s \leq t) \bigvee \sigma(\Xi_i')$, and (ii) for a fixed $t$, $g_k(\theta_{t-1}^k, \xi_t^k) - g_k(\theta_{t-1,\{i\}}^k, \xi_t^k)$ are independent conditional on

$\mathcal{F}_{t-1}$, one readily obtains

$$
\begin{aligned}
\mathbb{E}[|\Delta_4 - \Delta_{4,\{i\}}|^2] \lesssim & \frac{K}{n}\bigg(\sum_{s=i}^{n-1}\mathbb{E}[|\sum_{k=1}^{K} w_k(g_k(\theta_s^k, \xi_{s+1}^k) - g_k(\theta_{s,\{i\}}^k, \xi_{s+1}^k))|^2] \\
& + \mathbb{E}[|\sum_{k=1}^{K} w_k(g_k(\theta_{i-1}^k, \xi_i^k) - g_k(\theta_K^\star, \xi_i^k) - g_k(\theta_{i-1}^k, \xi_i^{k'}) + g_k(\theta_K^\star, \xi_i^{k'}))|^2]\bigg) \\
\lesssim & \frac{K}{n}\big(K^{-2}\sum_{s=i}^{n-1}\sum_{k=1}^{K}\mathbb{E}[|\theta_s^k - \theta_{s,\{i\}}^k|^2] + 2K^{-2}\mathbb{E}[\sum_{k=1}^{K}|\theta_{i-1}^k - \theta_K^\star|^2]) \\
\lesssim & \frac{1}{nK}\bigg(\sum_{s=i}^{n-1}\mathbb{E}[|\boldsymbol{\Theta}_s(I-J)|_F^2 + K|Y_s - Y_{s,\{i\}}|^2 + |\Theta_{s,\{i\}}(I-J)|_F^2] \\
& + 2\mathbb{E}[|\Theta_{i-1}(I-J)|_F^2 + K|Y_{i-1} - \theta_K^\star|^2]\bigg) \\
\lesssim & \frac{\eta_i}{nK} + \frac{1}{n}\sum_{s=i-1}^{n-1}\eta_s^2 = O(\frac{i^{-\beta}}{nK} + \frac{n^{1-2\beta} - i^{1-2\beta}}{n}),
\end{aligned}
\tag{B.24}
$$

where (B.24) follows from Theorem 2.(ii) and Lemma S16 of Gu and Chen [2024], and Proposition 4.

Combining (B.20), (B.23) and (B.24), for (B.19) we obtain

$$
\begin{aligned}
\sum_{i=1}^{n}\mathbb{E}[|u_i|\,|\Delta - \Delta_{n,\{i\}}|] \lesssim & \frac{1}{\sqrt{n}}\sum_{i=1}^{n}\Big(\frac{i^{-\beta/2}}{\sqrt{nK}} + \frac{i^{-\beta}}{\sqrt{K}} + \sqrt{K}(n^{1/2-2\beta} - i^{1/2-2\beta}) + \frac{n^{\frac{1}{2}-\beta} - i^{\frac{1}{2}-\beta}}{\sqrt{n}}\Big) \\
\lesssim & \frac{n^{\frac{1}{2}-\beta} + n^{-\frac{\beta}{2}}}{\sqrt{K}} + \sqrt{K}n^{1-2\beta}.
\end{aligned}
\tag{B.25}
$$

### B.1.6 Bound on $\mathbb{E}[|W||\Delta_n|]$

From $\mathbb{E}[|u_s|^2] \lesssim n^{-1/2}$, we have $\mathbb{E}[|W|^2] = O(1)$, where $O(\cdot)$ hides constants involving $d$. Moreover, we also have $\mathbb{E}[|\Delta_n|^2] \lesssim \sum_{l=1}^{4}\mathbb{E}[|\Delta_l|^2]$. For $\Delta_1$, observe that from (B.7),

$$
\mathbb{E}[|D_1|^2] \lesssim \frac{K}{n}|Q_0|_F^2 \lesssim \frac{K}{n}\exp(-C_\beta n^{1-\beta}) \text{ for some constant } C_\beta > 0.
\tag{B.26}
$$

On the other hand, for $\Delta_2$, we can invoke Assumption A.3 and Minkowsky's inequality to deduce

$$
\begin{aligned}
\sqrt{\mathbb{E}[|\Delta_2|^2]} \lesssim & \sqrt{\frac{K}{n}}\sum_{s=1}^{n}\sqrt{\mathbb{E}[|A(Y_{s-1} - \theta_K^\star) - \nabla F(Y_{s-1})|^2]} \\
\lesssim & \sqrt{\frac{K}{n}}\sum_{s=1}^{n}\sqrt{\mathbb{E}[|Y_{s-1} - \theta_K^\star|^4]} \\
\lesssim & \sqrt{\frac{K}{n}}\sum_{s=1}^{n}(\frac{\eta_s}{K} + \eta_t^2) = O(\frac{n^{\frac{1}{2}-\beta}}{\sqrt{K}} + \sqrt{K}n^{\frac{1}{2}-2\beta}),
\end{aligned}
\tag{B.27}
$$

where (B.27) follows from Proposition 7. Moving on, for $\Delta_3$, it is immediate that

$$
\sqrt{\mathbb{E}[\Delta_3^2]} \lesssim \frac{1}{\sqrt{n}}\sum_{s=1}^{n}\sqrt{\mathbb{E}[|\boldsymbol{\Theta}_s(I-J)|_F^2]} \lesssim \frac{1}{\sqrt{n}}\sum_{s=1}^{n}\eta_s\sqrt{K} = O(n^{\frac{1}{2}-\beta}\sqrt{K}).
\tag{B.28}
$$

Finally, for $\Delta_4$, we recall the argument in (B.24) to provide

$$
\begin{aligned}
\mathbb{E}[|\Delta_4|^2] &\lesssim \frac{K}{n} \sum_{s=1}^{n} \mathbb{E}[|\sum_{k=1}^{K} w_k\big(g_k(\theta_{s-1}^k, \xi_s^k) - g_k(\theta_K^\star, \xi_s^k)\big)|^2] \\
&\lesssim \frac{1}{nK} \sum_{s=1}^{n} \big(|\Theta_{s-1}(I - J)|_F^2 + K|Y_{t-1} - \theta_K^\star|^2\big) \\
&\lesssim \frac{1}{nK} \sum_{s=1}^{n} \big(\eta_s^2 K + \eta_s\big) \\
&\lesssim \frac{n^{-\beta}}{K} + n^{-2\beta}.
\end{aligned}
\tag{B.29}
$$

Combining (B.26)-(B.29), we obtain

$$
\mathbb{E}[|W||\Delta_n|] \lesssim \sqrt{\frac{K}{n}} \exp(-C_\beta n^{1-\beta}) + n^{\frac{1}{2}-\beta}\sqrt{K} + \frac{n^{-\frac{\beta}{2}}}{\sqrt{K}} + n^{-\beta}.
\tag{B.30}
$$

### B.1.7 Final Berry Esseen bound

Note that, for any $t \in [n]$, Pinelis-Rosenthal inequality (Theorem 4.1 of Pinelis [1994]) applies to yield

$$
\mathbb{E}[|\Sigma_n^{-1/2}\mathcal{N}_t|^3] \lesssim K^{3/2}\mathbb{E}[|\sum_{k=1}^{K} w_k W_t^k|^3] = O(K^{-1/2}),
$$

which immediately implies that

$$
\Upsilon_n \lesssim \sum_{s=1}^{n} \mathbb{E}[n^{-3/2}|\Sigma_n^{-1/2}\mathcal{N}_s|^3] = O(\frac{1}{\sqrt{nK}}).
\tag{B.31}
$$

Therefore, combining (B.25), (B.30) and (B.31), we have that

$$
d_C(\sqrt{n}\Sigma_n^{-1/2}(\bar{Y}_n - \theta_K^\star), Z) \lesssim \frac{1}{\sqrt{nK}} + n^{\frac{1}{2}-\beta}\sqrt{K} + \frac{n^{-\frac{\beta}{2}}}{\sqrt{K}},
$$

which completes the proof.

### B.2 Proof of Theorem 2.2

Let $\Gamma = \mathrm{Var}(\sum_{s=1}^{n} \eta_s \mathcal{A}_s^n \mathcal{N}_s) = \sum_{s=1}^{n} \eta_s^2 \mathcal{A}_s^n \mathcal{V}_K \mathcal{A}_s^{n^\top}$. Clearly, $|\Gamma|_F \lesssim \frac{n^{-\beta}}{K}$. Define $v_s = \Gamma^{-1/2}\eta_s \mathcal{A}_s^n \mathcal{N}_s$. Recall (B.3), and rewrite it as

$$
\Gamma^{-1/2}(Y_n - \theta_K^\star) = \tilde{W} + \tilde{D}_1 + \tilde{D}_2 + \tilde{D}_3 + \tilde{D}_4,
\tag{B.32}
$$

where

$$
\begin{aligned}
\tilde{W} &= \sum_{s=1}^{n} v_s, \\
\tilde{D}_1 &= \Gamma^{-1/2}\mathcal{A}_0^n(Y_0 - \theta_K^\star), \\
\tilde{D}_2 &= \Gamma^{-1/2}\sum_{s=1}^{n} \eta_s \mathcal{A}_s^n\big(A(Y_{s-1} - \theta_K^\star) - \nabla F(Y_{s-1})\big), \\
\tilde{D}_3 &= \Gamma^{-1/2}\sum_{s=1}^{n} \eta_s \mathcal{A}_s^n \sum_{k=1}^{K} w_k\big(\nabla F_k(Y_{s-1}) - \nabla F_k(\theta_{s-1}^k)\big), \\
\tilde{D}_4 &= \Gamma^{-1/2}\sum_{s=1}^{n} \eta_s \mathcal{A}_s^n \sum_{k=1}^{K} w_k g_k(\theta_{s-1}^k, \xi_s^k).
\end{aligned}
$$

Let $\tilde{\Delta}_l = |\tilde{D}_l|_2$ for $l = 1, 2, 4$, and let

$$\tilde{\Delta}_3 := |\Gamma^{-1/2}|_F K^{-1/2} \sum_{s=1}^{n} \eta_s \mathcal{A}_s^n |\boldsymbol{\Theta}_s(I - J)|_F.$$

Note that $|\tilde{D}_3|_2 \leq \tilde{\Delta}_3$. The terms $|\tilde{\Delta}_l|$ and $|\tilde{\Delta}_{l,\{i\}}|$ are defined and controlled very similarly to Theorem 2.1, and the details are omitted. The $\frac{n^{-\beta/2}}{\sqrt{K}}$ appears by controlling $\tilde{\Upsilon}_n := \sum_{s=1}^{n} \mathbb{E}[|v_s|^3]$, which we show below. Since $|H|_F \lesssim n^{-\beta} K^{-1}$, therefore

$$\sum_{s=1}^{n} \mathbb{E}[|v_s|^3] \lesssim K^{3/2} n^{\frac{3\beta}{2}} \sum_{s=1}^{n} \eta_s^3 |\mathcal{A}_s^n|^3 \mathbb{E}[|\mathcal{N}_s^3|] \lesssim n^{-\beta/2} K^{-1/2}.$$

## B.3 Application of Section 2: weighted multiplier bootstrap

In the context of vanilla SGD, Fang et al. [2018], Fang [2019], Sheshukova et al. [2025] introduced a novel multiplier bootstrap paradigm that precludes the necessity of estimating $\Sigma_n$ while performing inference. In this section, we adapt this approach for the particular decentralized setting, and hint towards the applicability of our Berry-Esseen theorems 2.1 and 2.2. Specifically, for each client $k \in [K]$, let $\mathbb{P}_W^k$ be a distribution of a random variable with $\mathbb{E}[W^k] = 1$ and $\mathrm{Var}[W^k] = \sigma_k^2$, $W^k \sim \mathbb{P}_W^k$. For the validity of the bootstrap procedure, we assume that, for all $k$, $\sigma_k^2 \leq C_0$ for some constant $C_0 > 0$. Moreover, we assume that $W^k$'s uniformly bounded, i.e. that there exists universal constants $c_1, c_2 > 0$ such that $c_1 \leq W^k \leq c_2$ for all $k \in [K]$ almost surely. For $b \in [B]$ where $B$ is the number of bootstrap samples, consider the augmented `local SGD` updates

$$\boldsymbol{\Theta}_t^{\{b\}} = (\boldsymbol{\Theta}_{t-1}^{\{b\}} - \eta_t \mathbf{G}_t^{\{b\}})C_t,$$

where

$$\mathbf{G}_t^{\{b\}} = K(w_1 W_{t,1}^{\{b\}} \nabla f_1(\theta_{t-1}^1, \xi_t^1), \ldots, w_K W_{t,K}^{\{b\}} \nabla f_K(\theta_{t-1}^K, \xi_t^K))$$

and for each $k \in [K]$, $\{W_{t,k}^{\{b\}}\}$ are i.i.d. random variables from $\mathbb{P}_W^k$, $t \in [n], b \in [B]$. For each $b \in [B]$, define $\bar{Y}_n^{\{b\}} = n^{-1} K^{-1} \sum_{t,k} \theta_t^{k\{b\}}$. Suppose $\mathcal{F}_n := \sigma(\xi_s^k : s \in [n], k \in [K])$. Following standard arguments (see Theorem 3 of Sheshukova et al. [2025]), adapting the proof of Theorem 2.1 as well as off-the-self Gaussian comparison results Chernozhukov et al. [2017], Devroye et al. [2018], it is possible to show that

$$\sup_{A \in \mathcal{B}(\mathbb{R}^d): A \text{ convex}} \left| \mathbb{P}(\sqrt{n}(\bar{Y}_n^{\{b\}} - \bar{Y}_n) \in A | \mathcal{F}_n) - \mathbb{P}(\sqrt{n}(\bar{Y}_n - \theta_K^\star) \in A) \right| \lesssim n^{1/2-\beta}\sqrt{K},$$

modulo logarithmic factors, with high probability with respect to $\mathcal{F}_n$. This result enables one to approximate the distribution of $\bar{Y}_n - \theta_K^\star$ via the bootstrap samples $\bar{Y}_n^{\{b\}}$. We remark that this approach works when our focus is on $\bar{Y}_n$; we do not expect this multiplier bootstrap to approximate the entire process $\{Y_t\}$. We leave the detailed derivations to future work, since the focus of this paper is on establishing the fundamental Gaussian approximation theorems.

## C Proof of Theorem 2.3

Recall that $\Sigma_n = n^{-1} \sum_{s=1}^{n} Q_s \mathcal{V}_K Q_s^\top$, and $\Sigma = KA^{-1}\mathcal{V}_K A^{-\top}$. We aim to decompose $\Sigma_n - K^{-1}\Sigma$ into manageable terms, and then control them piecemeal. To be precise, write

$$\Sigma_n - K^{-1}\Sigma = \frac{1}{n} \sum_{s=1}^{n} \left( (Q_s - A^{-1})\mathcal{V}_K A^{-\top} + A^{-1}\mathcal{V}_K(Q_s - A^{-1})^\top + (Q_s - A^{-1})\mathcal{V}_K(Q_s - A^{-1})^\top \right).$$

Crucial to our proof is the observation that $\mathcal{A}_s^t - \mathcal{A}_{s-1}^t = \eta_s A \mathcal{A}_s^t$ for all $s, t \in [n]$. Therefore,

$$\sum_{s=1}^{n} Q_s = \sum_{s=1}^{n} \sum_{j=s}^{n} \eta_s \mathcal{A}_s^j = \sum_{j=1}^{n} \sum_{s=1}^{j} \eta_s \mathcal{A}_s^j = \sum_{j=1}^{n} \sum_{s=1}^{j} A^{-1}(\mathcal{A}_s^j - \mathcal{A}_{s-1}^j) = A^{-1} \sum_{j=1}^{n}(I - \mathcal{A}_0^j),$$

$$(\text{C.1})$$

where the last equality is via a telescoping argument. From (C.1), we obtain $\sum_{s=1}^{n}(Q_s - A^{-1}) = -A^{-1}\sum_{j=1}^{n}\mathcal{A}_0^j$. Consequently, recalling that $|\mathcal{V}_K|_F = K^{-1/2}$, it is immediate that

$$n^{-1}|A^{-1}\mathcal{V}_K|_F \big| \sum_{s=1}^{n}(Q_s - A^{-1}) \big| \lesssim \frac{1}{n\sqrt{K}} \sum_{j=1}^{n}|\mathcal{A}_0^j| \lesssim \frac{1}{n\sqrt{K}} \int_1^n \exp(-x^{1-\beta})\,\mathrm{d}x = O(K^{-1/2}n^{\beta-1}).$$

Moving on, the term $(Q_s - A^{-1})\mathcal{V}_K(Q_s - A^{-1})^\top$ can be similarly controlled by $K^{-1/2}n^{\beta-1}$ from Lemma A.5 of Wu et al. [2024] (also see Lemma 11 and 12 of Sheshukova et al. [2025]). This completes the proof of (2.7). Finally, (2.8) follows from (2.7) on the account of Proposition 9, and the fact that $\Sigma_n$ is positive-definite, and hence maps a convex set to a convex set.

# D    Proofs of Section 3

In this section, we derive the time-uniform Gaussian approximation results Theorem 3.1 and 3.2. Our proofs are divided into four successive approximation steps. We summarize our arguments in the following. In the step I, we control the difference between the aggregated and the local client-level `local SGD` updates. In step II, we replace the martingale structure of the gradient noise by i.i.d. mean zero noise. In step III, we further linearize the `local SGD` updates, which we finally approximate by a stochastically linear Gaussian process such as (3.4) or (3.6) in Step IV.

## D.1    Proof of Theorem 3.1

### D.1.1    Step I

Consider $\mathbf{\Theta}_0^\circ = (\theta_K^\star, \ldots, \theta_K^\star) \in \mathbb{R}^{d \times K}$, and let

$$\mathbf{\Theta}_t^\circ = (\mathbf{\Theta}_{t-1}^\circ - \eta_t \mathbf{G}_t^\circ)C_t, \tag{D.1}$$

where $\mathbf{G}_t^\circ$ is defined similar to $\mathbf{G}_t$ in (2.2), but with $\theta_t^{k\circ}$ instead of $\theta_t^k$. Moreover, let $Y_t^\circ = K^{-1}\mathbf{\Theta}_t^\circ\mathbf{1} \in \mathbb{R}^K$. Suppose $R_t = (r_t^1, \ldots, r_t^K) = \mathbf{\Theta}_{t-1} - \eta_t\mathbf{G}_t$, and $R_t^\circ$ is defined likewise. Recall (B.1) and (B.2). Define two more intermediate oracle processes:

$$\tilde{Y}_t = \tilde{Y}_{t-1} - \eta_t\nabla F(\tilde{Y}_{t-1}) + \eta_t\sum_{k=1}^{K}w_k g_k(\theta_{t-1}^k, \xi_t^k),\ t \in [n],\ \tilde{Y}_0 = Y_0 \tag{D.2}$$

$$\tilde{Y}_t^\circ = \tilde{Y}_{t-1}^\circ - \eta_t\nabla F(\tilde{Y}_{t-1}^\circ) + \eta_t\sum_{k=1}^{K}w_k g_k(\theta_{t-1}^{k\circ}, \xi_t^k),\ t \in [n],\ \tilde{Y}_0^\circ = Y_0. \tag{D.3}$$

For a random variable $X$, let $\|X\| = (\mathbb{E}[|X|^2])^{1/2}$ be the random variable $\mathcal{L}_2$-norm. Then,

$$\|Y_t - \tilde{Y}_t\| \le \|(Y_{t-1} - \tilde{Y}_{t-1}) - \eta_t(\nabla F(Y_{t-1}) - \nabla F(\tilde{Y}_{t-1}))\| + \eta_t\|\sum_{k=1}^{K}w_k(F_k(Y_{t-1}) - F_k(\theta_{t-1}^k)\| := A + B. \tag{D.4}$$

Now, for the term $A$ in (D.4), invoking Assumptions A.1 and A.2, it is easy to observe that

$$A^2 = \|Y_{t-1} - \tilde{Y}_{t-1}\|^2 + \eta_t^2\|\nabla F(Y_{t-1}) - \nabla F(\tilde{Y}_{t-1})\|^2 - 2\eta_t\mathbb{E}[(Y_{t-1} - \tilde{Y}_{t-1})^\top(\nabla F(Y_{t-1}) - \nabla F(\tilde{Y}_{t-1}))]$$
$$\le (1 - 2\eta_t\mu + \eta_t^2 L^2)\|Y_{t-1} - \tilde{Y}_{t-1}\|^2. \tag{D.5}$$

On the other hand, for $B$ in (D.4), Assumption A.2 entails,

$$B^2 \le \eta_t^2 K\sum_{k=1}^{K}w_k^2\|F_k(Y_{t-1}) - F_k(\theta_{t-1}^k)\|^2 \le \eta_t^2 b_2^2 L^2 K^{-1}\mathbb{E}[\|\mathbf{\Theta}_t(I - J)\|_F^2] = O(\eta_t^4), \tag{D.6}$$

through an application of Lemma S.16 of Supplement of Gu and Chen [2024]. Combining (D.5) and (D.6) and choosing a $c > \mu \vee L$, it must hold that

$$\|Y_t - \tilde{Y}_t\| \le (1 - \eta_t c)\|Y_{t-1} - \tilde{Y}_{t-1}\| + O(\eta_t^2),$$

which readily yields

$$\|Y_t - \tilde{Y}_t\| = O(\eta_t). \tag{D.7}$$

Very similarly, one can show that $\|Y_t^\circ - \tilde{Y}_t^\circ\| = O(\eta_t)$. Finally it remains to show that $\tilde{Y}_t$ and $\tilde{Y}_t^\circ$ is approximately close. We show it as follows.

$$\|\tilde{Y}_t - \tilde{Y}_t^\circ\|^2$$

$$= \|\tilde{Y}_{t-1} - \tilde{Y}_{t-1}^\circ - \eta_t(\nabla F(\tilde{Y}_{t-1}) - \nabla F(\tilde{Y}_{t-1}^\circ))\|^2 + \eta_t^2 \sum_{k=1}^{K} w_k^2 \|g_k(\theta_{t-1}^k, \xi_t^k) - g_k(\theta_{t-1}^{k\circ}, \xi_t^k)\|^2 \tag{D.8}$$

$$\leq (1 - \eta_t c)\|\tilde{Y}_t - \tilde{Y}_t^\circ\|^2 + 4\eta_t^2 b_2^2 K^{-2} \sum_{k=1}^{K}(\|\theta_{t-1}^k - Y_{t-1}\|^2 + \|Y_{t-1} - \theta_K^\star\|^2 + \|\theta_{t-1}^{k\circ} - Y_{t-1}^\circ\|^2 + \|Y_{t-1}^\circ - \theta_K^\star\|^2) \tag{D.9}$$

$$\leq (1 - \eta_t c)\|\tilde{Y}_t - \tilde{Y}_t^\circ\|^2 + 4\eta_t^2 b_2^2 (2\frac{\eta_t}{K^2} + 2\frac{\eta_t^2}{K}) \tag{D.10}$$

$$\leq (1 - \eta_t c)\|\tilde{Y}_t - \tilde{Y}_t^\circ\|^2 + O(\frac{\eta_t^3}{K^2} + \frac{\eta_t^4}{K}). \tag{D.11}$$

Here, (D.8) employs Assumption A.3 to deduce that $g_k(\theta_{t-1}^k, \xi_t^k) - g_k(\theta_{t-1}^{k\circ}, \xi_t^k)$ are mean-zero martingale differences adapted to $\mathcal{F}_t := \sigma(\xi_s^k, s \leq t, k \in [K])$; moreover, since $\{\xi_t^k\}_{k=1}^{K}$ are independent for a fixed $t$, hence $g_k(\theta_{t-1}^k, \xi_t^k) - g_k(\theta_{t-1}^{k\circ}, \xi_t^k)$ are also uncorrelated. Additionally, (D.9) uses a treatment analogous to (D.5) along with applying Assumption A.2 to the $g_k$ terms; and (D.10) involves applications of Lemmas S.16 and Theorem 2(ii) from Gu and Chen [2024]. Finally, (D.11) immediately implies that

$$\|\tilde{Y}_t - \tilde{Y}_t^\circ\|^2 = O(\eta_t^2 K^{-2} + \eta_t^3 K^{-1}),$$

which, coupled with (D.7), yields,

$$\max_{1 \leq t \leq n} |\sum_{s=1}^{t}(Y_s - \tilde{Y}_s^\circ)| \leq \sum_{t=1}^{n}|Y_t - \tilde{Y}_t^\circ| = O_{\mathbb{P}}(n^{1-\beta}). \tag{D.12}$$

### D.1.2   Step II

Moving on, we approximate $\tilde{Y}_t^\circ$ by another oracle descent sequence, given by

$$Y_t^\dagger = Y_{t-1}^\dagger - \eta_t \nabla F(Y_{t-1}^\dagger) + \eta_t \sum_{k=1}^{K} w_k g_k(\theta_K^\star, \xi_t^k), \ Y_0^\dagger = Y_0. \tag{D.13}$$

Importantly, (D.13) can be leveraged to linearize the original sequence $Y_{t-1}$ in (B.1). Before we proceed in that direction, we still need to approximate $\tilde{Y}_t^\circ$ by $Y_t^\dagger$. From (D.3) and (D.13), it follows very similarly to (D.8)-(D.11), that,

$$\|\tilde{Y}_t^\circ - Y_t^\dagger\|^2$$

$$= \|\tilde{Y}_{t-1}^\circ - Y_{t-1}^\dagger - \eta_t(\nabla F(\tilde{Y}_{t-1}^\circ) - \nabla F(Y_{t-1}^\dagger))\|^2 + \eta_t^2 \sum_{k=1}^{K} w_k^2 \|g_k(\theta_{t-1}^{k\circ}, \xi_t^k) - g_k(\theta_K^\star, \xi_t^k)\|^2$$

$$\leq (1 - \eta_t c)\|\tilde{Y}_{t-1}^\circ - Y_{t-1}^\dagger\|^2 + \eta_t^2 L^2 b_2^2 K^{-2}(\mathbb{E}[|\Theta_t^\circ(I - J)|_F^2] + C\eta_t + C\eta_2^2 K)$$

$$\leq (1 - \eta_t c)\|\tilde{Y}_{t-1}^\circ - Y_{t-1}^\dagger\|^2 + O(\eta_t^3 K^{-2} + \eta_t^4 K^{-1}),$$

which immediately yields $\|\tilde{Y}_t^\circ - Y_t^\dagger\| = O(\eta_t K^{-1} + \eta_t^{3/2} K^{-1/2})$. Similar to (D.12), here too we finally obtain

$$\max_{1 \leq t \leq n} |\sum_{s=1}^{t}(\tilde{Y}_s^\circ - Y_s^\dagger)| = O_{\mathbb{P}}(n^{1-\beta} K^{-1} + n^{1-3\beta/2} K^{-1/2}). \tag{D.14}$$

### D.1.3 Step III

Define $\tilde{Y}_t^\dagger$ as

$$\tilde{Y}_t^\dagger = Y_{t-1}^\dagger - \eta_t \nabla F(\tilde{Y}_{t-1}^\dagger) + \eta_t \sum_{k=1}^K w_k g_k(\theta_K^\star, \xi_t^k), \ \tilde{Y}_0 = \theta_K^\star.$$

Then it trivially follows that

$$\|Y_t^\dagger - \tilde{Y}_t^\dagger\|^2 = \|(Y_{t-1}^\dagger - \tilde{Y}_{t-1}^\dagger) - \eta_t(\nabla F(Y_{t-1}^\dagger) - \nabla F(\tilde{Y}_{t-1}^\dagger))\|^2$$
$$\leq (1 - \eta_t c)\|Y_{t-1}^\dagger - \tilde{Y}_{t-1}^\dagger\|^2 \lesssim \exp(-t^{1-\beta})|Y_0 - \theta_K^\star|^2, \quad \text{(D.15)}$$

which implies $\max_{1 \leq t \leq n} |\sum_{s=1}^t (Y_s^\dagger - \tilde{Y}_s^\dagger)| = O_\mathbb{P}(1)$, since $\int_1^n \exp(-t^{1-\beta}) \, \mathrm{d}t = O(1)$. Moving on, to linearize $\tilde{Y}_t^\dagger$, write (D.13) as

$$\tilde{Y}_t^\dagger - \theta_K^\star = (I - \eta_t A)(\tilde{Y}_{t-1}^\dagger - \theta_K^\star) - \eta_t(\nabla F(\tilde{Y}_{t-1}^\dagger) - A(\tilde{Y}_{t-1}^\dagger - \theta_K^\star)) + \eta_t \sum_{k=1}^K w_k g_k(\theta_K^\star, \xi_t^k),$$
$$\text{(D.16)}$$

where $A = \nabla_2 F(\theta_K^\star)$. Note that, Assumption A.1 along with $\sum w_k = 1$ implies that $A \succeq \mu I$. Mimicking (D.16), define

$$Y_t^\diamond = (I - \eta_t A)Y_{t-1}^\diamond + \eta_t \sum_{k=1}^K w_k g_k(\theta_K^\star, \xi_t^k), \ Y_0^\diamond = 0. \quad \text{(D.17)}$$

Clearly, it follows that

$$\mathbb{E}[|\tilde{Y}_t^\dagger - \theta_K^\star - Y_t^\diamond|] \leq (1 - \eta_t \mu)\mathbb{E}[|\tilde{Y}_{t-1}^\dagger - \theta_K^\star - Y_{t-1}^\diamond|] + \eta_t \mathbb{E}[|\nabla F(\tilde{Y}_{t-1}^\dagger) - A(Y_{t-1}^\dagger - \theta_K^\star)|]$$
$$\leq (1 - \eta_t \mu)\mathbb{E}[|\tilde{Y}_{t-1}^\dagger - \theta_K^\star - Y_{t-1}^\diamond|] + L_Q \eta_t \mathbb{E}[|\tilde{Y}_{t-1}^\dagger - \theta_K^\star|^2] \quad \text{(D.18)}$$
$$\leq (1 - \eta_t \mu)\mathbb{E}[|\tilde{Y}_{t-1}^\dagger - \theta_K^\star - Y_{t-1}^\diamond|] + O(\eta_t^2 K^{-1} + \eta_t^3), \quad \text{(D.19)}$$

where, (D.18) follows from Assumption A.3 , and (D.19) is a trivial consequence of Theorem 2.(ii) of Gu and Chen [2024]. Finally, (D.19) yields that

$$\max_{1 \leq t \leq n} |\sum_{s=1}^t (Y_s^\dagger - \theta_K^\star - Y_s^\diamond)| = O_\mathbb{P}(n^{1-\beta}K^{-1/2} + n^{1-2\beta}). \quad \text{(D.20)}$$

### D.1.4 Step IV

Note that $Y_t^\diamond$ is a linear process, and thus we can hope to bear down standard strong invariance principle results Komlós et al. [1975], Sakhanenko [2006], Gëttse and Zaĭtsev [2009] on it to yield an asymptotically optimal Gaussian approximation. In particular, let $\mathcal{V}_K = \mathrm{Var}(\sum_{k=1}^K w_k g_k(\theta_K^\star, \xi^k))$, $\xi^k \sim \mathcal{P}_k, k \in [K]$. Note that, Assumption 4.2 in Gu and Chen [2024] can also be summarized as $\|K\mathcal{V}_K\|_F \asymp 1$. We pursue two different type of Gaussian approximation. Let $W_t^k = g_k(\theta_K^\star, \xi_t^k)$, and $\mathcal{N}_t = \sum_{k=1}^K w_k W_t^k$. By Gëttse and Zaĭtsev [2009], there exists i.i.d. $Z_1, \ldots, Z_n \overset{i.i.d.}{\sim} N(0, K\mathcal{V}_K)$, such that $\max_{1 \leq t \leq n} |\sum_{s=1}^t (\sqrt{K}\mathcal{N}_s - Z_s)| = o_\mathbb{P}(n^{1/p})$. Write (D.17) as

$$Y_t^\diamond = (I - \eta_t A)Y_{t-1}^\diamond + \eta_t \mathcal{N}_t,$$

which immediately yields

$$\sum_{s=1}^t Y_s^\diamond = \sum_{s=1}^t \eta_s \mathcal{N}_s B_{s,t}, \ B_{s,t} = \sum_{j=s}^t \mathcal{A}_s^j, \quad \text{(D.21)}$$

where $\mathcal{A}_s^t = \prod_{j=s+1}^t (I - \eta_j A)$, $\mathcal{A}_t^t = I$. Mimicking $Y_t^\diamond$, define $Y_{t,1}^G$ as in (3.4), to which we can simplify $\sum_{s=1}^t Y_{s,1}^G = K^{-1/2} \sum_{s=1}^t \eta_s Z_s \sum_{j=s}^t \mathcal{A}_s^j$. Note that,

$$\max_{1 \leq t \leq n} |\sum_{s=1}^t (Y_s^\diamond - Y_{s,1}^G)| \leq \max_{1 \leq t \leq n} \Omega_t \max_{1 \leq t \leq n} |\sum_{s=1}^t (\mathcal{N}_s - K^{-1/2} Z_s)| = o_\mathbb{P}(\max_{1 \leq t \leq n} \Omega_t \, n^{1/p} K^{-1/2}).$$
$$\text{(D.22)}$$

where $\Omega_t := |B_{1,t}|_F + \sum_{s=2}^t |B_{s,t} - B_{s-1,t}|_F$. The proof of (3.5) is completed after combining (D.12), (D.14), (D.20) and (D.22) in view of Proposition 2.

## D.2 Proof of Theorem 3.2 and Proposition 1

In this subsection, we pursue a finer, client-level Gaussian approximation, with slight sacrifice to the optimality in terms of error rate. In particular, the steps I, II and II from the proof of Theorem 3.1 carry forward verbatim. Consequently, it enables us to invoke from Theorem 2.1 of Mies and Steland [2023] so that for each $k \in [K]$, there exists $Z_1^k, \ldots, Z_n^k \sim N(0, \text{Var}(W^k))$, such that

$$\max_{1 \leq t \leq n} \sum_{k=1}^{K} |\sum_{s=1}^{t} (W_s^k - Z_s^k)| = O_{\mathbb{P}}(K^{\frac{3}{4} - \frac{1}{2p}} n^{\frac{1}{2p} + \frac{1}{4}} \sqrt{\log n}). \tag{D.23}$$

Here $W^k$ denotes a generic $g_k(\theta_K^\star, \xi^k)$. For $\tilde{\theta}_t^{1^G}, \ldots, \tilde{\theta}_t^{K^G} \in \mathbb{R}^d$, define $\tilde{\Theta}_t^G = (\tilde{\theta}_t^{1^G} \ldots \tilde{\theta}_t^{K^G}) \in \mathbb{R}^{d \times k}$, and simultaneously define the recursion (3.6). Letting $Y_{t,2}^G = K^{-1} \tilde{\Theta}_t^G \mathbf{1}$, one arrives at the recursion

$$Y_{t,2}^G = (I - \eta_t A) Y_{t-1,2}^G + \eta_t \sum_{k=1}^{K} w_k Z_t^k, \tag{D.24}$$

to which, from (D.23), one has

$$\max_{1 \leq t \leq n} |\sum_{s=1}^{t} (Y_t^\diamond - Y_{t,2}^G)| \leq \max_{1 \leq t \leq n} \Omega_t |\sum_{s=1}^{t} \sum_{k=1}^{K} w_k (W_t^k - Z_t^k)| \leq b_2 \log n \max_{1 \leq t \leq n} K^{-1} \sum_{k=1}^{K} |\sum_{s=1}^{t} (W_t^k - Z_t^k)|$$

$$= O_{\mathbb{P}}((n/K)^{\frac{1}{4} + \frac{1}{2p}} (\log n)^{3/2}), \tag{D.25}$$

where the second inequality is due to Proposition 2 and $\max_k w_k \leq b_2 K^{-1}$. Again, we conclude (3.7) in light of (D.12), (D.14), (D.20) and (D.25). Finally, Proposition 1 follows trivially from Theorems 3.1 and 3.2.

## E Auxiliary propositions

In this section, we present some technical results required to prove our main theorems. Propositions 2 and 3 relates the `local SGD` updates to its asymptotic covariance matrices. In particular, Proposition 2 controls the implicit total variation between the linearized `local SGD` updates, and as such, is crucial in deriving the time-uniform approximations `Aggr-GA` and `Client-GA`.

**Proposition 2.** *Let $A \in \mathbb{R}^{d \times d}$ be a positive definite matrix with smallest eigen value $\lambda_{\min} > 0$, and define $\mathcal{A}_s^t = \prod_{j=s+1}^{t} (I - \eta_j A), \mathcal{A}_t^t = I$. If*

$$\Omega_t := |B_{1,t}|_F + \sum_{s=2}^{t} |B_{s,t} - B_{s-1,t}|_F,$$

*then it holds that $\max_{1 \leq t \leq n} \Omega_t = O(\log n)$.*

**Proposition 3.** *Let $B_{s,t}$ be as in Proposition 2. Then, for all $s \geq 1, t \geq s$, it holds that*

$$|B_{s,t} - A^{-1}|_F \lesssim s^{-1} + \exp\left(-c_\beta(t^{1-\beta} + s^{1-\beta})\right),$$

*where $c_\beta$ is some constant depending on $\beta, \lambda_{\min}$.*

Propositions 4-8 characterizes the various properties of the `local SGD` updates and its difference with its corresponding Lindeberg coupling. These results hold under the conditions of Theorem 2.1, and can be considered as its building blocks.

**Proposition 4.** *Under the assumptions of Theorem 2.1, it holds that for all $i \in [n], t \geq i$, it holds that*

$$\mathbb{E}[|Y_t - Y_{t,\{i\}}|^2] = O(\eta_t^2). \tag{E.1}$$

**Proposition 5.** *Under the conditions of Theorem 2.1, for all $i \in [n], t > i$, it holds that*

$$\mathbb{E}[|(\Theta_t - \Theta_{t,\{i\}})(I - J)|_F^2] = O(\eta_t^4 K).$$

**Proposition 6.** *Under the conditions of Theorem 2.1, it holds that*

$$\mathbb{E}[|\mathbf{\Theta}_t(I - J)|_F^4] = O(\eta_t^4 K^2). \tag{E.2}$$

**Proposition 7.** *Under the conditions of Theorem 2.1, it holds that*

$$\mathbb{E}[|Y_t - \theta_K^\star|^4] = O\left(\frac{\eta_t^2}{K^2} + \eta_t^4\right). \tag{E.3}$$

**Proposition 8.** *Grant the assumptions of Theorem 2.1. Then, for $t \geq i$, it holds that*

$$\mathbb{E}[|Y_t - Y_{t,\{i\}}|^4] = O(\eta_t^4). \tag{E.4}$$

Proposition 9 is a typical Gaussian comparison results that relates the finite-sample covariance $\Sigma_n$ to the asymptotic covariance $\Sigma$ in terms of the corresponding normal distributions. This result enables theorem 2.3 to reflect the computation-communication trade-off of Remark 2.2.

**Proposition 9** (Gaussian comparison lemma; Theorem 1.1, Devroye et al. [2018])**.** *Let $\Sigma_1$ and $\Sigma_2$ be positive definite covariance matrices in $\mathbb{R}^{p \times p}$. Let $X \sim \mathcal{N}(0, \Sigma_1)$ and $Y \sim \mathcal{N}(0, \Sigma_2)$. Then*

$$d_{\mathrm{TV}}(X, Y) \leq \frac{3}{2} \left\| \Sigma_2^{-1/2} \Sigma_1 \Sigma_2^{-1/2} - I_p \right\|_{\mathrm{F}}.$$

## E.1 Proofs of the auxiliary results

*Proof of Proposition 2.* In the following, all $\lesssim$ solely depend on $\beta$ and $\lambda_{\min}$. Observe that for $s < t$, $B_{s,t} = \eta_s\left(I + \eta_{s+1}^{-1}(I - \eta_s A)B_{s+1,t}\right)$. Therefore, it can be written that

$$B_{s,t} - B_{s-1,t} = \frac{\eta_s - \eta_{s-1}}{\eta_s} B_{s,t} + \eta_{s-1}(B_{s,t}A - I) := I_1 + I_2. \tag{E.5}$$

The $I_1$ term is relatively straightforward by noting that $\max_{s,t} |B_{s,t}|_F = O(1)$, and $\left|\frac{\eta_s - \eta_{s-1}}{\eta_s}\right| = O(s^{-1})$. On the other hand, for $I_2$, Proposition 3 instructs that

$$\eta_{s-1}|B_{s,t}A - I|_F \lesssim s^{-\beta-1} + s^{-\beta} \exp\left(-c_\beta(t^{1-\beta} + s^{1-\beta})\right). \tag{E.6}$$

Combining (E.5) and (E.6), we obtain

$$|B_{s,t} - B_{s-1,t}|_F \lesssim s^{-1} + s^{-\beta} \exp\left(-c_\beta(t^{1-\beta} + s^{1-\beta})\right),$$

which immediately shows

$$\Omega_t \lesssim \sum_{s=1}^t s^{-1} + \exp(-c_\beta t^{1-\beta}) \int_1^t s^{-\beta} \exp(c_\beta s^{1-\beta}) \lesssim \log t,$$

which completes the proof. $\qquad\square$

*Proof of Proposition 3.* Decompose $B_{s,t} = \eta_s \sum_{j=s}^t \mathcal{A}_j^s$ as

$$B_{s,t} - A^{-1} = -A^{-1}\mathcal{A}_s^t + \sum_{j=s}^{t-1}(\eta_{j+1} - \eta_s)\mathcal{A}_j^s + \eta_s \mathcal{A}_s^t, \tag{E.7}$$

where the sum $\sum_{j=s}^{t-1}$ is interpreted as 0 if $s = t$. For the term $A^{-1}\mathcal{A}_s^t$ in (E.7), we deduce $|\mathcal{A}_s^t| \lesssim \exp(-c_\beta(t^{1-\beta} + s^{1-\beta}))$. On the other hand,

$$\sum_{j=s}^{t-1}(\eta_{j+1} - \eta_s)|\mathcal{A}_j^s|_F \lesssim s^{-\beta-1} \exp(c_\beta s^{1-\beta}) \sum_{j=s}^{t-1}(j - s) \exp(-j^{1-\beta}) \lesssim s^{-1}.$$

This completes the proof. $\qquad\square$

*Proof of Proposition 4.* From (B.2) we write

$$Y_t - Y_{t,\{i\}} = \begin{cases} \eta_i \sum_{k=1}^K \big(g_k(\theta_{i-1}^k, \xi_i^k) - g_k(\theta_{i-1}^k, \xi_i^{k'})\big), & t = i, \\ (Y_{t-1} - Y_{t-1,\{i\}}) - \eta_t(\nabla F(Y_{t-1}) - \nabla F(Y_{t-1,\{i\}})) \\ \quad + \eta_t \sum_{k=1}^K w_k\big(\nabla F_k(Y_{t-1}) - \nabla F(Y_{t-1,\{i\}}) - \nabla F_k(\theta_{t-1}^k) + \nabla F_k(\theta_{t-1,\{i\}}^k)\big) \\ \quad + \eta_t \sum_{k=1}^K w_k\big(g_k(\theta_{t-1}^k, \xi_t^k) - g_k(\theta_{t-1,\{i\}}, \xi_t^k)\big), & t > i. \end{cases}$$

(E.8)

Clearly, when $t = i$, we have trivially that $\mathbb{E}[|Y_i - Y_{i,\{i\}}|^2] = O(\eta_i^2 K^{-1})$. Hence, we focus on $t > i$. Consider the observation that $\sum_{k=1}^K w_k\big(g_k(\theta_{t-1}^k, \xi_t^k) - g_k(\theta_{t-1,\{i\}}, \xi_t^k)\big)$ is a martingale difference sequence adapted to the filtration $\mathcal{F}_t = \sigma(\Xi_s : s \le t) \bigvee \sigma(\Xi_i')$. Moreover, for a fixed $t$, $g_k(\theta_{t-1}^k, \xi_t^k) - g_k(\theta_{t-1,\{i\}}, \xi_t^k)$ are independent conditional on $\mathcal{F}_{t-1}$. Therefore, rewriting (E.8) as

$$Y_t - Y_{t,\{i\}} = T_1 + T_2 + T_3 \tag{E.9}$$

with

$$T_1 = (Y_{t-1} - Y_{t-1,\{i\}}) - \eta_t(\nabla F(Y_{t-1}) - \nabla F(Y_{t-1,\{i\}})),$$

$$T_2 = \eta_t \sum_{k=1}^K w_k\big(\nabla F_k(Y_{t-1}) - \nabla F(Y_{t-1,\{i\}}) - \nabla F_k(\theta_{t-1}^k) + \nabla F_k(\theta_{t-1,\{i\}}^k)\big), \text{ and,}$$

$$T_3 = \eta_t \sum_{k=1}^K w_k\big(g_k(\theta_{t-1}^k, \xi_t^k) - g_k(\theta_{t-1,\{i\}}, \xi_t^k)\big),$$

it is easy to see that $\mathbb{E}[T_1^\top T_3] = \mathbb{E}[T_2^\top T_3] = 0$. Consequently, from (E.8), one computes

$$\mathbb{E}[|Y_t - Y_{t,\{i\}}|^2] = \mathbb{E}[|T_1|^2] + \mathbb{E}[|T_2|^2] + \mathbb{E}[|T_3|^2] + 2\mathbb{E}[T_1^\top T_2]. \tag{E.10}$$

Now all that is required is to build a recursion by analyzing (E.10) term-by-term. Note that standard arguments invoking Assumptions A.2 and A.1 yields

$$\mathbb{E}[|T_1|^2] \le (1 - \eta_t c)\mathbb{E}[|Y_{t-1} - Y_{t-1,\{i\}}|^2]. \tag{E.11}$$

On the other hand, for $T_3$, we proceed as follows:

$$\begin{aligned} \mathbb{E}[|T_3|^2] &= \eta_t^2 \sum_{k=1}^K w_k^2 \mathbb{E}[|g_k(\theta_{t-1}^k, \xi_t^k) - g_k(\theta_{t-1,\{i\}}^k, \xi_t^k)|^2] \\ &\lesssim \eta_t^2 \sum_{k=1}^K w_k^2\Big(\mathbb{E}[|\theta_{t-1}^k - Y_{t-1}|^2] + \mathbb{E}[|\theta_{t-1,\{i\}}^k - Y_{t-1,\{i\}}|^2] + \mathbb{E}[|Y_{t-1} - Y_{t-1,\{i\}}|^2]\Big) \\ &\lesssim \frac{\eta_t^2}{K} \mathbb{E}[|Y_{t-1} - Y_{t-1,\{i\}}|^2] + O(\frac{\eta_t^4}{K}), \end{aligned} \tag{E.12}$$

where $O(\eta_t^4 K^{-1})$ bound in (E.12) is derived upon applying Lemma S16 of Gu and Chen [2024]. Very similarly, one can bound $T_2$ as

$$\mathbb{E}[|T_2|^2] \lesssim \eta_t^2 K \sum_{k=1}^K w_k^2 \mathbb{E}[|Y_{t-1} - \theta_{t-1}^k|^2 + |Y_{t-1,\{i\}} - \theta_{t-1,\{i\}}^k|^2] = O(\eta_t^4). \tag{E.13}$$

Finally we tackle the cross-product term in (E.10). Again, Assumption A.2 and yet another application of Lemma S16 of Gu and Chen [2024] produces

$$\begin{aligned} \mathbb{E}[T_1^\top T_2] &\le \eta_t \sqrt{\mathbb{E}[|T_1|^2]} \sqrt{\mathbb{E}[|\sum_{k=1}^K w_k\big(\nabla F_k(Y_{t-1}) - \nabla F(Y_{t-1,\{i\}}) - \nabla F_k(\theta_{t-1}^k) + \nabla F_k(\theta_{t-1,\{i\}}^k)\big)|^2]} \\ &\le \eta_t \sqrt{\mathbb{E}[|T_1|^2]} \frac{b_2\sqrt{L}}{\sqrt{K}} \sqrt{\sum_{k=1}^K \mathbb{E}[|Y_{t-1} - \theta_{t-1}^k|^2 + |Y_{t-1,\{i\}} - \theta_{t-1,\{i\}}^k|^2]} \\ &\lesssim \eta_t^2 \sqrt{\mathbb{E}[|T_1|^2]} \\ &\le \eta_t(\frac{1}{4c}\eta_t^2 + c\mathbb{E}[|T_1|^2]) \\ &\le \eta_t \frac{c}{4} \mathbb{E}[|T_1|^2] + O(\eta_t^3), \end{aligned}$$

(E.14)

(E.15)

where (E.14) involves an application of Young's inequality $xy \leq \epsilon x^2 + (4\epsilon)^{-1} y^2$ with $\epsilon = (4c)^{-1}$, where $c$ is as in (E.11). Therefore, in view of $(1 + \eta_t \frac{c}{2})(1 - \eta_t c) \leq 1 - \eta_t \frac{c}{2}$, we combine (E.11) -(E.15) into (E.10) to obtain

$$\mathbb{E}[|Y_t - Y_{t,\{i\}}|^2] \leq (1 - \eta_t \frac{c}{2} + \frac{\eta_t^2}{K})\mathbb{E}[|Y_{t-1} - Y_{t-1,\{i\}}|^2] + O(\eta_t^3 + \frac{\eta_t^4}{K}), \ t > i,$$

which immediately shows (E.1) with standard manipulations (see Lemma A.1 and A.2 of Zhu et al. [2023]; Polyak and Juditsky [1992] $\qquad\square$

*Proof of Proposition 5.* Recall $C_t$ from (2.2). Let $r_{t,s}$ be the number of synchronization steps between $s-1$ and $t$, satisfying $\lfloor \frac{t-s}{\tau} \rfloor + 1 \geq r_{t,s} \geq \lfloor \frac{t-s}{\tau} \rfloor$. Further note that $\mathbf{C}^{r_{t,s}} = \prod_{j=s}^{t} C_j$. From (2.2) and (B.14), it is easy to see that

$$(\boldsymbol{\Theta}_t - \boldsymbol{\Theta}_{t,\{i\}})(I - J) = -\sum_{s=i}^{t} \eta_s (\mathbf{G}_s - \mathbf{G}_{s,\{i\}})(\mathbf{C}^{r_{t,s}} - J), \tag{E.16}$$

where we have repeatedly used the fact that $\mathbf{C1} = \mathbf{1}$. Moreover, it also holds that

$$\|\mathbf{C}^{r_{t,s}} - J\|_2 \leq \left(\rho^{\frac{1}{\tau}}\right)^{\max\{t-s-(\tau-2),0\}} = 1_{\{t-s<\tau-1\}} + 1_{\{t-s\geq\tau-1\}}\tilde{\rho}^{t-s-(\tau-1)} := \kappa_{\rho,\tau}(t,s), \tag{E.17}$$

where $\tilde{\rho} = \rho^{1/\tau}$. Equation E.17 also appears as (S7) in Gu and Chen [2024]. In view of (E.17), one can expand (E.16) as follows:

$$\mathbb{E}[|\sum_{s=i}^{t} \eta_s(\mathbf{G}_s - \mathbf{G}_{s,\{i\}})(\mathbf{C}^{r_{t,s}} - J)|_F^2]$$

$$\leq \sum_{s=i}^{t} \kappa_{\rho,\tau}^2(t,s)\eta_s^2 \mathbb{E}[|\mathbf{G}_s - \mathbf{G}_{s,\{i\}}|_F^2]$$

$$+ \sum_{s=i}^{t}\sum_{l=i,l\neq s}^{t} \kappa_{\rho,\tau}(t,s)\kappa_{\rho,\tau}(t,l)\eta_s\eta_l \mathbb{E}\Big[\mathrm{Tr}\big[(\mathbf{G}_s - \mathbf{G}_{s,\{i\}})^\top(\mathbf{G}_l - \mathbf{G}_{l,\{i\}})\big]\Big] \tag{E.18}$$

$$\leq \sum_{s=i}^{t} \kappa_{\rho,\tau}^2(t,s)\eta_s^2 \mathbb{E}[|\mathbf{G}_s - \mathbf{G}_{s,\{i\}}|_F^2]$$

$$+ \sum_{s=i}^{t}\sum_{l=i,l\neq s}^{t} \kappa_{\rho,\tau}(t,s)\kappa_{\rho,\tau}(t,l)2^{-1}\mathbb{E}\Big(\eta_s^2|\mathbf{G}_s - \mathbf{G}_{s,\{i\}}|_F^2 + \eta_l^2|\mathbf{G}_l - \mathbf{G}_{l,\{i\}}|_F^2\Big)$$

$$\leq \sum_{s=i}^{t} \kappa_{\rho,\tau}^2(t,s)\eta_s^2 \mathbb{E}[|\mathbf{G}_s - \mathbf{G}_{s,\{i\}}|_F^2] + \sum_{s=i}^{t} \kappa_{\rho,\tau}(t,s)\eta_s^2 \mathbb{E}[|\mathbf{G}_s - \mathbf{G}_{s,\{i\}}|_F^2]\Big(\sum_{l=i,l\neq s}^{t} \kappa_{\rho,\tau}(t,l)\Big). \tag{E.19}$$

Now we are required to tackle $\mathbb{E}[|\mathbf{G}_s - \mathbf{G}_{s,\{i\}}|_F^2]$. To that end, observe that for $s > i$

$$\mathbb{E}[|\mathbf{G}_s - \mathbf{G}_{s,\{i\}}|_F^2] = K^2 \sum_{k=1}^{K} w_k^2 \mathbb{E}[|\nabla f_k(\theta_{s-1}^k, \xi_s^k) - \nabla f_k(\theta_{s-1,\{i\}}^k, \xi_s^k)|^2]$$

$$\leq 2b_2^2 L \sum_{k=1}^{K} \mathbb{E}[|\theta_{s-1}^k - \theta_{s-1,\{i\}}^k|^2]$$

$$\lesssim \sum_{k=1}^{K} \mathbb{E}[|\theta_{s-1}^k - Y_{s-1}|^2] + \sum_{k=1}^{K} \mathbb{E}[|\theta_{s-1,\{i\}}^k - Y_{s-1,\{i\}}|^2] + \sum_{k=1}^{K} \mathbb{E}[|Y_{s-1} - Y_{s-1,\{i\}}|^2]$$

$$= O(\eta_s^2 K), \tag{E.20}$$

where (E.20) follows from Lemma S16 of Gu and Chen [2024] and Proposition 4 respectively. Putting (E.20) back into (E.19), we obtain

$$\mathbb{E}[|(\boldsymbol{\Theta}_t - \boldsymbol{\Theta}_{t,\{i\}})(I - J)|_F^2] \lesssim \sum_{s=i}^{t} \eta_s^4 K \kappa_{\rho,\tau}^2(t,s) \leq \sum_{s=i}^{t} \eta_s^4 \tilde{\rho}^{t-s} = O(\eta_t^4 K),$$

where the last assertion uses $\int_1^n x^{-a} e^{yx} \mathrm{d}x \lesssim n^{-a} e^{ny}$ for $a, y > 0$, where $\lesssim$ is independent of $n$. This completes the proof. □

*Proof of Proposition 6.* We can re-purpose significant portions of the proof of Lemma S16 of Gu and Chen [2024] to prove (E.2). Indeed, writing $\Theta_t = \sum_{s=1}^{t} \eta_s \mathbf{G}_s C_s$, we have from the referenced proof that

$$\mathbb{E}[|\Theta_t(I - J)|_F^4] = \mathbb{E}[|\sum_{s=1}^{t} \eta_s \mathbf{G}_s (\mathbf{C}^{r_{t,s}} - J)|_F^4]$$

$$\leq 2\mathbb{E}\left[\left(\sum_{s=1}^{t} \eta_s^2 \kappa_{\rho,\tau}^2(t,s)|\mathbf{G}_s|^2\right)^2\right] + 2\mathbb{E}\left[\left(\sum_{s=1}^{t} \kappa_{\rho,\tau}(t,s) \sum_{l=1,l\neq s}^{t} \kappa_{\rho,\tau}(t,l)\eta_s\eta_l|\mathbf{G}_s^\top \mathbf{G}_l|\right)^2\right]$$

$$:= S_1 + S_2. \tag{E.21}$$

For $S_1$ in (E.21), it is straightforward to obtain

$$S_1 \lesssim \sum_{s=1}^{t} \eta_s^4 \kappa_{\rho,\tau}^4(t,s)\mathbb{E}[|\mathbf{G}_s|^4] + \sum_{s=1}^{t} \sum_{l=1,l\neq s}^{t} \eta_s^2\eta_l^2 \kappa_{\rho,\tau}^2(t,s)\kappa_{\rho,\tau}^2(t,l)\mathbb{E}[|\mathbf{G}_s|^2|G_l|^2]$$

$$\lesssim \sum_{s=1}^{t} \eta_s^4 \kappa_{\rho,\tau}^4(t,s)\mathbb{E}[|\mathbf{G}_s|^4] + \sum_{s=1}^{t} \kappa_{\rho,\tau}^2(t,s)\eta_s^4\mathbb{E}[|\mathbf{G}_s|^4] \max_s \sum_{l=1,l\neq s}^{t} \kappa_{\rho,\tau}^2(t,l) \tag{E.22}$$

$$\lesssim \sum_{s=1}^{t} K^2 \eta_s^4 (\kappa_{\rho,\tau}^4(t,s) + \kappa_{\rho,\tau}^2(t,s)) = O(\eta_t^4 K^2), \tag{E.23}$$

where, in (E.22) we apply AM-GM inequality to derive

$$\eta_s^2\eta_l^2\mathbb{E}[|\mathbf{G}_s|^2|\mathbf{G}_l|^2] \leq \frac{\eta_s^4\mathbb{E}[|\mathbf{G}_s|^4] + \eta_l^4\mathbb{E}[|\mathbf{G}_l|^4]}{2}.$$

A very similar treatment yields the same bound on $S_2$, completing the proof of (E.2). □

*Proof of Proposition 7.* Write

$$R_t := Y_t - \theta_K^\star = E_1 + E_2 + E_3, \text{ where,}$$

$$E_1 = R_{t-1} - \eta_t \nabla F(Y_{t-1}),$$

$$E_2 = \eta_t \sum_{k=1}^{K} w_k(\nabla F_k(Y_{t-1}) - \nabla F_k(\theta_{t-1}^k)), \text{ and}$$

$$E_3 = \eta_t \sum_{k=1}^{K} w_k g_k(\theta_{t-1}^k, \xi_t^k). \tag{E.24}$$

Note that trivially, Assumptions A.1 and A.2 imply that

$$\mathbb{E}[|E_1|^4] \leq (1 - \eta_t c)\mathbb{E}[|R_{t-1}|^4]. \tag{E.25}$$

Moving on, for $E_2$ we proceed as follows:

$$\mathbb{E}[|E_2|^4] = \eta_t^4 \mathbb{E}[|\sum_{k=1}^{K} w_k(\nabla F_k(Y_{t-1}) - \nabla F_k(\theta_{t-1}^k))|^4]$$

$$\leq C_2 \frac{\eta_t^4}{K^2} \mathbb{E}\left[\left(\sum_{k=1}^{K} |Y_{t-1} - \theta_{t-1}^k|^2\right)^2\right]$$

$$\leq C_2 \frac{\eta_t^4}{K^2} \mathbb{E}[|\Theta_{t-1}(I - J)|_F^4] \leq C_2' \eta_t^8, \tag{E.26}$$

for some constants $C_2, C_2' > 0$, where the final assertion is drawn from Proposition 6. Finally, for $E_3$, we obtain,

$$\mathbb{E}[|E_3|^4] = \eta_t^4 \mathbb{E}[|\sum_{k=1}^{K} w_k g_k(\theta_{t-1}^k, \xi_t^k)|^4] \leq C_3 \frac{\eta_t^4}{K^2}, \tag{E.27}$$

for some constant $C_3 > 0$, where we have used the fact that $g_k(\theta_{t-1}^k, \xi_t^k)$ are mean-zero and independent random vectors conditional on $\mathcal{F}_t$. Now, we will leverage (E.25)-(E.27) to develop bounds on the cross-product terms. In particular,

$$\mathbb{E}[|E_1|^2|E_2|^2] \leq \sqrt{\mathbb{E}[|E_1|^4]}\sqrt{\mathbb{E}[|E_l|^4]}$$
$$\leq C_3 \eta_t^4 \sqrt{\mathbb{E}[|E_1|^4]}$$
$$\leq C_3 \min\{\eta_t \varepsilon \mathbb{E}[|E_1|^4] + (4\varepsilon)^{-1}\eta_t^7, \ \eta_t^2 \varepsilon \mathbb{E}[|E_1|^4] + (4\varepsilon)^{-1}\eta_t^6\}, \tag{E.28}$$

and similarly

$$\mathbb{E}[|E_1|^2|E_3|^2] \leq C_3 \min\{\eta_t \varepsilon \mathbb{E}[|E_1|^4] + (4\varepsilon)^{-1}\frac{\eta_t^3}{K^2}, \ \eta_t^2 \varepsilon \mathbb{E}[|E_1|^4] + (4\varepsilon)^{-1}\frac{\eta_t^2}{K^2}\}, \tag{E.29}$$

where the final assertions follow from Young's inequality. Here, $\varepsilon$ is chosen to be small enough, however it remains a constant; the explicit choice of $\varepsilon$ will be indicated towards the end of the proof, when we collect terms to establish the recursion. Note that, quite trivially, from (E.26) and (E.27), one has

$$\mathbb{E}[|E_2|^2|E_3|^2] \leq C_4 \frac{\eta_t^6}{K} \text{ for some constant } C_4 > 0. \tag{E.30}$$

Rest of the cross-products are strictly dominated by some combinations of the terms analyzed till now. For example, for $l, r, q \in \{1, 2, 3\}$, Cauchy-Schwarz and AM-GM inequalities implies that

$$\mathbb{E}[(E_l^\top E_q)^2] \leq \mathbb{E}[|E_l|^2|E_q|^2],$$
$$\mathbb{E}[|E_l|^2(E_r^\top E_q)] \leq \sqrt{\mathbb{E}[|E_l|^4]}\sqrt{\mathbb{E}[(E_r^\top E_q)^2]},$$
$$\mathbb{E}[|(E_l^\top E_r)(E_l^\top E_q)|] \leq 2^{-1}(\mathbb{E}[(E_l^\top E_r)^2] + \mathbb{E}[(E_l^\top E_q)^2]). \tag{E.31}$$

A careful collection of terms from (E.25)-(E.31) yields

$$\mathbb{E}[|R|_t^4] \leq (1 - \eta_t c)(1 + \eta_t C_0 \varepsilon)\mathbb{E}[|R_{t-1}|^4] + O(\frac{\eta_t^3}{K^2} + \eta_t^5), \tag{E.32}$$

where $C_0$ is a large constant depending upon $C_2, C_3$ and $C_4$. Now, choose $\varepsilon > 0$ so that $C_0 \varepsilon < c/2$, upon which we immediately obtain $(1 - \eta_t c)(1 + \eta_t C_0 \varepsilon) < 1 - \eta_t c/2$. Therefore, (E.32) immediately yields (E.3). □

*Proof of Proposition 8.* Recall (E.8). Clearly, for $t = i$, the result is trivial. For $t > i$, we leverage (E.9). A proof very similar to (E.3), which uses a similar decomposition (E.24), can then be followed. The crucial term is $\mathbb{E}[|T_1|^2|T_3|^2]$, which is computed below. Note that

$$\mathbb{E}[|T_3|^4] \leq \frac{\eta_t^4}{K}\mathbb{E}\left[\sum_{k=1}^{K}|g_k(\theta_{t-1}^k, \xi_t^k) - g_k(\theta_{t-1,\{i\}}^k, \xi_t^k)|^4\right]$$

$$\leq L'\frac{\eta_t^4}{K}\mathbb{E}\left[\sum_{k=1}^{K}|\theta_{t-1}^k - \theta_{t-1,\{i\}}^k|^4\right]$$

$$\leq 27L'\frac{\eta_t^4}{K}\mathbb{E}\left[\sum_{k=1}^{K}|\theta_{t-1}^k - Y_{t-1}|^4 + K|Y_{t-1} - Y_{t-1,\{i\}}|^4 + \sum_{k=1}^{K}|\theta_{t-1,\{i\}} - Y_{t-1,\{i\}}|^4\right]$$

$$\lesssim \frac{\eta_t^4}{K}\mathbb{E}[K^{-1}|\Theta_{t-1}(I - J)|_F^4 + K^{-1}|\Theta_{t-1,\{i\}}(I - J)|_F^4 + K|Y_{t-1} - Y_{t-1,\{i\}}|^4]$$

$$\lesssim \eta_t^4 \mathbb{E}[|Y_{t-1} - Y_{t-1,\{i\}}|^4] + O(\eta_t^8). \tag{E.33}$$

Therefore, from (E.33), it follows

$$
\begin{aligned}
\mathbb{E}[|T_1|^2|T_3|^2] &\leq \sqrt{\mathbb{E}[|T_1|^4]}\sqrt{\mathbb{E}[|T_3|^4]} \\
&\leq \sqrt{\mathbb{E}[|T_1|^4]}\big(\sqrt{L}\eta_t^2\sqrt{\mathbb{E}[|Y_{t-1} - Y_{t-1,\{i\}}|^4]} + O(\eta_t^4)\big) \\
&\lesssim \eta_t^2\mathbb{E}[|Y_{t-1} - Y_{t-1,\{i\}}|^4] + \eta_t^4\sqrt{\mathbb{E}[|T_1|^4]} \\
&\lesssim \eta_t^2\mathbb{E}[|Y_{t-1} - Y_{t-1,\{i\}}|^4] + \eta_t^3\mathbb{E}[|T_1|^4] + O(\eta_t^5).
\end{aligned}
\tag{E.34}
$$

Rest of the terms are computed similar to Proposition 7, and the details are omitted. The final recursion can be derived to be

$$
\mathbb{E}[|Y_t - Y_{t,\{i\}}|^4] \leq (1 - \eta_t c)\mathbb{E}[|Y_{t-1} - Y_{t-1,\{i\}}|^4] + O(\eta_t^5) \text{ for some small constant } c > 0,
$$

which immediately yields (E.4). □

# F    Additional Simulations

## F.1    Effect of $n$ and $K$ on the Berry-Esseen rate

In this subsection, we empirically investigate the behavior of the Berry-Esseen error $d_C(\sqrt{n}(\bar{Y}_n - \theta_K^\star), Z)$ for $Z \sim N(0, \Sigma_n)$ with varying choices of the number of iterations $N$ and the number of clients $K$. If the bound (2.5) is sharp, we expect the Berry-Esseen error to decay with increasing $N$, and increase with an increasing number of clients. Since the distance $d_C$ involves taking a supremum over all convex sets, which is computationally infeasible, we restrict ourselves to the following measure of the approximation error:

$$
\tilde{d}_c = \sup_{x \in [0,c]} \big|\mathbb{P}(|\sqrt{n}\Sigma_n^{-1/2}(\bar{Y}_n - \theta_K^\star)| \leq x) - \mathbb{P}(|Z| \leq x)\big|, \; Z \sim N(0, I).
$$

For large enough $c > 0$, we expect $\tilde{d}_c$ to be a reasonable proxy for $d_C$. For our numerical exercises to quantify $\tilde{d}_c$, we analyze the output $\bar{Y}_n$ of the `local SGD` algorithm under a federated random effects model, hereafter denoted as `FRand-eff`. We describe the set-up below.

### F.1.1    `FRand-eff` formulation

Consider a positive definite matrix $\Gamma \in \mathbb{R}^{d \times d}$ and $\beta_0 \in \mathbb{R}^d$, and let $\mathcal{D}^K := \{\beta_1, \ldots, \beta_K\} \overset{\text{i.i.d.}}{\sim} N_d(\beta_0, \Gamma)$. Moreover, consider $\Sigma^K := \{\sigma_1^2, \ldots, \sigma_K^2\} \subset \mathbb{R}_{>0}^K$. For $k \in [K]$ and at $t \in [n]$-th iteration, suppose that the $k$-th client has access to data $(y_{tk}, x_{tk}) \in \mathbb{R} \times \mathbb{R}^d$ generated from the linear model $y_{tk} \sim N(x_{tk}^\top \beta_k, \sigma_k^2)$. If the weights are chosen such that $w_1 = \ldots = w_K = K^{-1}$, then clearly $\theta_K^\star = \sum_{k=1}^K w_k \beta_k \to \beta_0$ as $K \to \infty$. Therefore, `local SGD` can be employed, and we expect $\bar{Y}_n$ to consistently estimate $\beta_0$ as $n$ and $K$ grow. This model highlights the need for information-sharing across client, since unless $\Gamma = 0$, the output of local vanilla SGD for any particular client is inconsistent for $\beta_0$.

For the purpose of the numerical exercises in this section, we choose $d = 2$ and $\beta_0 = (2, -3)^\top$, and let $\Gamma = \gamma I$ with $\gamma \geq 0$. In particular, $\gamma = 0$ corresponds to a fixed effect $\beta_0$ from which each client generates their observations. For each $K$, we generate $\Sigma^K$ uniformly from the set $\{1, \ldots, 5\}$, $\mathcal{D}^K$ from the specification above, and keep them fixed throughout the corresponding experiments as $n$ varies. The underlying connection matrix C is taken as $C_{ij} = \frac{1}{3}I\{|i - j| \leq 1\}$, $i, j \in [K]$. In other words, every client is connected to only its two immediate neighbors.

### F.1.2    $\tilde{d}_c$ versus $n$ and $K$

In this set, we analyze the behavior of $\tilde{d}_c$ versus $n$ for different choices of $K, \tau$, and $\gamma$. In particular, we aim to verify the Berry-Esseen error rate of $n^{1/2-\beta}\sqrt{K}$ from Theorems 2.1 and 2.2. Let $\gamma = 1$. Consider the following two separate settings corresponding to $n, K$, and $\tau$.

- **Setting 1.** Let $K = 10$, and vary $n \in \{100, 200, 300, 400, 500\}$, and $\tau \in \{10, 15, 20\}$. For each pair of $(n, \tau)$, we plot $\tilde{d}_c$ against $n$.

- **Setting 2.** fix $n = 300$, and vary $K \in \{20, 40, 60, 80, 100\}$, and $\tau \in \{10, 15, 20\}$. For each pair of $(K, \tau)$, we plot $\tilde{d}_c$ against $K$.

We provide the practical details behind empirically estimating $\tilde{d}_c$. For each of the experimental settings described above, we generate $(y_{tk}, x_{tk}) \in \mathbb{R} \times \mathbb{R}^d, k \in [K], t \in [n]$ from the `FRand-eff` specification described above, and run the `local SGD` algorithm with step size $\eta_t = 0.3t^{-0.75}$ for $t \in [n]$. For the large choice of $c = 100$, $\tilde{d}_c$ is empirically estimated by $n_{\text{sim}} = 1000$ many independent Monte-Carlo repetitions of our experiments.

Figure 1 allows us to draw important practical insights from the rates of Theorems 2.1 and 2.3. Firstly, from the Settings 1 and 3, the synchronization parameter $\tau$ does not seem to have a significant effect on the behavior of $\tilde{d}_c$. Moreover, Figure 1(left) seems to corroborate well with the conclusion of Theorem 2.1, with $\tilde{d}_c$ decaying with $n$ for a fixed $K$. On the other hand, for Setting 2, Figure 1(right) seems to point towards a trade-off in terms of $K$ for fixed $n$. This particular behavior becomes clearer as we recall (2.5). For fixed $n = 300$, the initial decay of $\tilde{d}_c$ (and by extension, $d_C$) with increasing $K$, is caused by the $n^{-\beta/2}K^{-1/2}$ term. However, as $K$ increases, the term $n^{1-\beta/2}\sqrt{K}$ starts to dominate, leading the error $\tilde{d}_c$ to increase with increasing $K$. This numerical exercise establish the sharpness of our upper-bound (2.5), complementing the discussions in Remark 2.1.

To investigate the behavior of $\tilde{d}_c$ further, we also consider the case $\gamma = 5$. In Figure 4, due to the increased heterogeneity across $\beta_k$, the effect of synchronization becomes more pronounced; for the same values of $n, K, \tau$, the $\tilde{d}_c$ values are much lesser compared to that in Figure 1. In particular, in Figure 4(right), the inflection point in $K$ beyond which $n^{1/2-\beta}\sqrt{K}$ starts to dominate, has shifted to the right. This is understandable, since increased variability among $\beta_k$ means a greater reward for sharing information, and thus the effect of increasing the clients leads to lowering the error $\tilde{d}_c$ for a longer regime, before the asymptotics of $n^{1/2-\beta}\sqrt{K}$ eventually kicks in.

## F.2 Computation-communication trade-off

In this section, we numerically investigate the computation-communication trade-off hinted at in Remark 2.2. There, we noted that if $K \asymp n^c$ for $c > 1/2$, then, based on our upper bounds, we argued that for no $\beta \in (1/2, 1)$ does $d_C$ converge to 0. In particular, this observation is trivial for $\beta \in (1/2, 1/2 + c/2]$ since the central limit theory itself fail to help in view of violation of $K \gtrsim n^{2\beta-1}$. Of particular interest is the range $\beta \in (1/2 + c/2, 1)$, where, as per Theorem 3 of Gu and Chen [2024], central limit theory continues to hold, but (2.8) suggests that the upper bound to $d_C$ is no longer $o(1)$.

To explore this phenomena through numerical examples, we invoke `FRand-eff` for $\gamma = 0$, and let $n \in \{100, 200, 300, 400, 500\}$, and $K = \lfloor n^r \rfloor$ for $r \in \{0.2, 0.6\}$. In conjunction with Theorem 2.3, we consider the following error:

$$d_c^\dagger = \sup_{x \in [0,c]} \left| \mathbb{P}(|\sqrt{n}\Sigma^{-1/2}(\bar{Y}_n - \theta_K^\star)| \le x) - \mathbb{P}(|Z| \le x) \right|,$$

where $\Sigma = A^{-1}SA^{-\top}$. Moreover, we consider the `local SGD` algorithm with $\tau = 5$, and $\eta_t = 0.5t^{-\beta}$. In light of $1/2 + r/2 \in \{0.6, 0.8\}$, we ensure the validity of central limit theory by letting $\beta \in \{0.85, 0.9, 0.95\}$. Finally, for each value of $r$, we plot $\tilde{d}_c$ against $n$ for the different choices of $\beta$. For $r = 0.2$ and $r = 0.6$, the evident decreasing and increasing trends of $\tilde{d}_c$ in Figure 2 respectively, vindicate not only the sharpness of our Berry-Esseen bounds Theorems 2.1-2.3, but also clearly highlights trade-off at the region $\sqrt{n} \ll K \ll n$.

## F.3 Effect of heterogeneity

In this section, we characterize the effect of data hetero-geneity on the Berry-Esseen errors of Theorem 2.1. To start with, we note the following generalized version of Theorem 2.1, whose proof merely follows from careful tracking of constants, and is therefore omitted.

**Corollary F.1.** *For $\xi^{(k)} \sim \mathcal{P}_k$ (the data distribution of client $k \in [K]$), define $\mu_{l,k} = \mathbb{E}[|\nabla F_k(\theta_K^\star) - \nabla f_k(\theta_K^\star, \xi^{(k)})|^l]$ for $l = 2, 3, 4$. Denote $M_l = \sum_k \mu_{k,l}$, and $M_{22} = \sum_{k \ne l}^K \mu_{2,k}^2 \mu_{2,l}^2$. Assume that for*

*some constant $c > 0$, $\min_k \mu_{4,k} \geq c$. Further, assume $\max_\theta \sum_{k=1}^{K} w_k \|\nabla F(\theta) - \nabla F_k(\theta)\|^4 \leq \kappa^4$ for some $\kappa \geq 0$. As long as $\tau \leq \theta n$, for some $\theta \in (0,1)$, under the assumptions of Theorem 2.1, it follows that:*

$$d_{\mathrm{C}}(\sqrt{n}(\bar{Y}_n - \theta_K^\star), Z)$$

$$\lesssim (\kappa^c + 1)\left(\sqrt{\tau} + \frac{1}{\sqrt{1 - \rho^{1/\tau}}}\right)\left(\frac{M_3}{\sqrt{n}M_2^{3/2}} + \frac{n^{-\beta/2}}{\sqrt{M_2}} + n^{1/2-\beta}\left(\sqrt{K} + \frac{\sqrt{M_4 + M_{22}}}{K\sqrt{M_2}} + \frac{\sqrt{M_2}}{K}\right)\right.$$

$$\left. + n^{1-2\beta}\left(\sqrt{K}\sqrt{\frac{M_4}{M_2}} + \sqrt{M_2}\right) + n^{1/2-2\beta}\frac{M_4}{\sqrt{M_2}} + n^{-\beta}\right), \tag{F.1}$$

*for some constant $c \geq 0$.*

The effect of heterogeneity can also be verified empirically, by noticing that $\gamma$ plays the role of heterogeneity in the `Frand-eff` formulation in Section F.1.1 in the supplement. The corresponding simulation results can be seen in the following.

| $\gamma$ | $n = 100$ | $n = 200$ | $n = 300$ |
|---|---|---|---|
| 1 | 0.222 | 0.224 | 0.154 |
| 2 | 0.258 | 0.188 | 0.112 |
| 3 | 0.322 | 0.322 | 0.302 |
| 4 | 0.374 | 0.272 | 0.232 |
| 5 | 0.542 | 0.292 | 0.302 |

Table 1: Berry-Esseen error across different levels of heterogeneity under step-size= $0.3t^{-0.75}$, number of clients $K = 10$, $\tau = 2$, $\mathbf{C}_{ij} = 1/3I\{|j - i| \leq 1\}$, dimension $d = 2$. All the results are based on 500 Monte-Carlo simulations.

## F.4 Effect of Synchronization

Note that, from the verbose Corollary F.1, one can glean the following result that condenses the effect of synchronization in an interpretable manner.

**Corollary F.2.** *As long as $\tau \leq \theta n$, for some $\theta \in (0,1)$, under the assumptions of Theorem 2.1 it follows that:*

$$d_{\mathrm{C}}(\sqrt{n}(\bar{Y}_n - \theta_K^\star), Z) \lesssim \left(\sqrt{\tau} + \frac{1}{\sqrt{1 - \rho^{1/\tau}}}\right)\left(\frac{1}{\sqrt{nK}} + n^{\frac{1}{2}-\beta}\sqrt{K} + \frac{n^{-\frac{\beta}{2}}}{\sqrt{K}}\right). \tag{F.2}$$

*for some constant $c \geq 0$.*

In particular, we allow the synchronization parameter $\tau$ to grow linearly with $n$. Clearly, as the number of local updates $\tau$ increases, the errors increase with a $\sqrt{\tau}$ rate. This have also been verified empirically in the following table.

| $\tau$ | $n = 100$ | $n = 200$ | $n = 300$ |
|---|---|---|---|
| 10 | 0.087 | 0.100 | 0.110 |
| 20 | 0.118 | 0.146 | 0.155 |
| 30 | 0.139 | 0.168 | 0.171 |
| 40 | 0.155 | 0.183 | 0.185 |
| 50 | 0.164 | 0.196 | 0.211 |
| 60 | 0.176 | 0.225 | 0.220 |
| 70 | 0.175 | 0.218 | 0.231 |
| 80 | 0.176 | 0.230 | 0.252 |
| 90 | 0.176 | 0.234 | 0.253 |
| 100 | 0.175 | 0.241 | 0.261 |

Table 2: Comparison of Berry-Esseen error across different values of $\tau$ under step-size= $0.3t^{-0.75}$, number of clients $K = 10$, , dimension $d = 2$, Connection matrix $\mathbf{C}_{ij} = 1/3I\{|j - i| \leq 1\}$.

## F.5 Effect of connection matrix

In the following, we provide experimental results on how the Berry-Esseen errors depend on the network topology ($\rho$), where $\rho$ is the second largest eigen value of the connection graph $\mathbf{C}$. As the network topology becomes less connected ($\rho \uparrow 1$), the GA error scales as $(1 - \rho^{1/\tau})^{-1/2}$. On the other hand, when $\rho = 0$, the network is densest as $\mathbf{C} = K^{-1}\mathbf{1}_K\mathbf{1}_K^\top$, and the algorithm essentially becomes a centralized one.

| $\rho$ | $n = 100$ | $n = 200$ | $n = 300$ |
|------|-----------|-----------|-----------|
| 0.1 | 0.094 | 0.084 | 0.089 |
| 0.2 | 0.106 | 0.107 | 0.104 |
| 0.3 | 0.121 | 0.126 | 0.124 |
| 0.4 | 0.136 | 0.143 | 0.138 |
| 0.5 | 0.149 | 0.164 | 0.154 |
| 0.6 | 0.165 | 0.192 | 0.175 |
| 0.7 | 0.184 | 0.217 | 0.203 |
| 0.8 | 0.203 | 0.245 | 0.250 |
| 0.9 | 0.212 | 0.275 | 0.294 |

Table 3: Comparison of Berry-Esseen error across different values of $\rho$ under step-size$= 0.3t^{-0.75}$, number of clients $K = 10$, , dimension $d = 2$, synchronization parameter $\tau = 10$. For each $\rho$, $\mathbf{C} = \rho I_K + (1 - \rho)K^{-1}\mathbf{11}_K$.

## F.6 Performance of the time-uniform Gaussian approximations

This section devotes itself to numerical studies to validate the efficacy of the Gaussian approximations `Aggr-GA` and `Client-GA`, discussed in Section 3. Consider the quantities

$$U_n = \max_{1 \le t \le n} |\sum_{s=1}^{t}(Y_s - \theta_K^\star)|, \; U_n^{\texttt{Aggr-GA}} = \max_{1 \le t \le n} |\sum_{s=1}^{t} Y_{s,1}^G|, \text{ and } U_n^{\texttt{Client-GA}} = \max_{1 \le t \le n} |\sum_{s=1}^{t} Y_{s,2}^G|,$$

where $Y_{s,1}^G$ and $Y_{s,2}^G$ are defined as in Theorem 3.1. Moreover, we also consider the Brownian motion approximation by functional central limit theorem as another competitor, and as such, consider

$$U_n^{\texttt{f-CLT}} = \max_{1 \le t \le n} |\sum_{s=1}^{t} Z_s|, \; Z_1, \ldots, Z_n \overset{i.i.d.}{\sim} N(0, \Sigma),$$

where $\Sigma$ is as in Section 2.2.1. In order to compare the distributions of $U_n^{\texttt{Aggr-GA}}$, $U_n^{\texttt{Client-GA}}$ and $U_n^{\texttt{f-CLT}}$ to that of $U_n$, we resort to Q-Q plots. Fix $N = 500$, $\tau = 20$, and let $K \in \{10, 25, 50\}$. For each triplet of $(N, K, \gamma)$, we simulate $n_{\text{sim}} = 500$ parallel independent `local SGD` chains with step-sizes $\eta_t = 0.7t^{-0.85}$, and observations from the `FRand-eff` model in order to empirically simulate $U_n$. Concurrently, we also simulate $n_{\text{sim}}$ independent observations from the distributions of $U_n^{\texttt{Aggr-GA}}$, $U_n^{\texttt{Client-GA}}$ and $U_n^{\texttt{f-CLT}}$ by running the corresponding chains in parallel. The QQ-plots are shown in Figure 3.

The sub-optimality of the functional CLT as a time-uniform Gaussian approximation to $\{Y_t\}$ is empirically evident from the QQ-plots. Both our proposals `Aggr-GA` and `Client-GA` uniformly dominate the approximation via Brownian motion across different settings. Moreover, as $K$ increases from left panel to the right, `Aggr-GA` out-performs `Client-GA`. This in line with Proposition 1 (i) and (ii) underpinning the sharper approximation rate for `Aggr-GA`. However, we must recall that `Client-GA` requires local covariance estimation for each client, thus protecting the privacy of the federated setting.

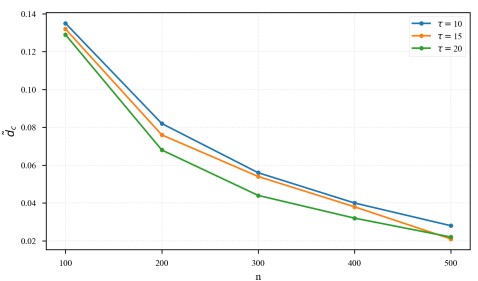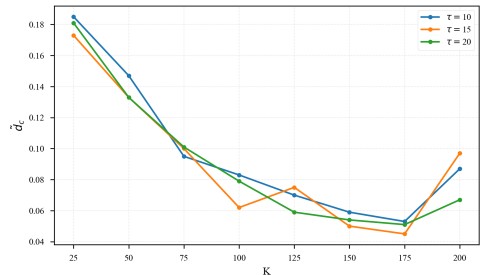

Figure 4: Plot of $\tilde{d}_c$ against $n$ and $K$ for $\gamma = 5$, and Settings 1(left), and 2(right).

### F.7 Ablation Studies

In the following, we further investigate the affect of heterogeneity and synchronization for the time-uniform Gaussian approximations. In particular, we carry out two ablation studies for the parameters $\tau$ and $\gamma$ in the `FRand-eff` formulation. In the first experiment, we fix $N = 500, K = 15, d = 2, \mathbf{C}_{ij} = 1/3I\{|j - i| \leq 1\}$ and $\gamma = 1$, and vary $\tau = 5, 50, 100$. For each particular setting, we report the following quantities:

$$Q_{\texttt{f-CLT}} = \max_{\alpha \in (0,1)} \frac{|q_{1-\alpha}(U_n) - q_{1-\alpha}(U_n^{\texttt{f-CLT}})|}{q_{1-\alpha}(U_n)}, \ Q_{\texttt{Aggr-GA}} = \max_{\alpha \in (0,1)} \frac{|q_{1-\alpha}(U_n) - q_{1-\alpha}(U_n^{\texttt{Aggr-GA}})|}{q_{1-\alpha}(U_n)}, \text{ and,}$$

$Q_{\texttt{Client-GA}} = \max_{\alpha \in (0,1)} \frac{|q_{1-\alpha}(U_n) - q_{1-\alpha}(U_n^{\texttt{Client-GA}})|}{q_{1-\alpha}(U_n)}$. The following table summarizes the results.

| $\tau$ | $Q_{\texttt{f-CLT}}$ | $Q_{\texttt{Aggr-GA}}$ | $Q_{\texttt{Client-GA}}$ |
|---|---|---|---|
| 5 | 1.495 | 0.214 | 0.327 |
| 10 | 2.009 | 0.44 | 0.53 |
| 15 | 2.476 | 0.663 | 0.883 |

Clearly, as the number of local steps $\tau$ increase, the efficacy of each Gaussian approximation worsens; however, the two Gaussian approximations proposed, `Aggr-Ga` and `Client-GA` consistently outperforms a functional-CLT based approach, mirroring our results from Section 4.3. Moreover, `Aggr-GA` consistently provides the sharpest approximation, vindicating the theory outlined in Section 3.

Moreover, we also fix $N = 500, K = 15, \tau = 20$, and vary the heterogeneity parameter $\gamma = 1, 5, 10$. The results are as follows.

| $\gamma$ | $Q_{\texttt{f-CLT}}$ | $Q_{\texttt{Aggr-GA}}$ | $Q_{\texttt{Client-GA}}$ |
|---|---|---|---|
| 1 | 1.047 | 0.237 | 0.275 |
| 5 | 2.047 | 0.605 | 0.646 |
| 10 | 2.805 | 0.547 | 0.782 |

We again see the worsening performance of the Gaussian approximations with increasing heterogeneity.

## G  Experiments on attack instance detection via time-uniform approximations

### G.1  Attack instance detection

To round off our discussion in Section 3.1, one can exploit the time-uniform approximation guarantees of Theorems 3.1 and 3.2 to propose valid, Gaussian bootstrap-based algorithms for attack instance detection. For convenience, we only state an algorithm based on Theorem 3.1; a corresponding algorithm based on Theorem 3.2 can be likewise constructed.

---

**Algorithm 2** `Time-uniform Gaussian bootstrap`

---

**Input:** Initializations $\Theta_0 = (\theta_0^1, \ldots, \theta_0^K) \in \mathbb{R}^{d \times K}$; Connection matrix $\mathbf{C}$; Synchronization parameter $\tau$; Loss functions $f_k(\cdot, \xi^k), \xi^k \sim \mathcal{P}_k, k \in [K]$, weights $\{w_k\}_{k=1}^K$, number of iterations $n$, step-size schedules $\{\eta_t\}_{t=1}^n$, Hessian $A$; number of bootstrap samples $B$; covariance matrix $\mathcal{V}_K$.

- Let $E_\tau = \{\tau, 2\tau, \ldots, L\tau\}$, where $L = \lfloor \frac{n}{\tau} \rfloor$. Initialize $t = 1$. Stopping time $\hat{T}_0 = 1$, estimated attack instance $\hat{s}_0 = +\inf$.
- While $t \le n$:
  1. Store the local SGD iterates $Y_t$, and calculate $R_t = \max_{1 \le s \le t} s|\bar{Y}_s - \bar{Y}_t|$ and $s_t = \arg\max_{1 \le s \le t} s|\bar{Y}_s - \bar{Y}_t|$.
  2. For $B = 1, \ldots, B$:
     Draw $Z_t^{(b)} \sim N(0, \mathcal{V}_K)$, and do $Y_{t,1}^{G,(b)} = (I - \eta_t A)Y_{t-1,1}^{G,(b)} + \eta_t Z_t^{(b)} K^{-1/2}, Y_{0,1}^{G,(b)} = \mathbf{0}$. Calculate $R_t^{G,(b)} = \max_{1 \le s \le t} s|\bar{Y}_{s,1}^{G,(b)} - \bar{Y}_{t,1}^{G,(b)}|$
- $\hat{q}_{1-\alpha} \leftarrow$ sample quantile($\{R_t^{G,(b)}\}$).
- **Thresholding**: If $R_t > \hat{q}_{1-\alpha} + c\sqrt{n}$:
  $$\hat{T}_0 \leftarrow t, \hat{s}_0 \leftarrow s_{\hat{T}_0}. \text{ Stop.}$$
  Else $t+ = 1$.

**Output:** $\hat{T}_0 I\{\hat{T}_0 < n\}, s_{\hat{T}_0}$.

---

We remark that Algorithm 2 is directly motivated from (3.3); it not only detects the attack instance, but detects it as soon as possible in a sequential manner. We provide some numerical experiments validating this algorithm in Section G.2.

## G.2 Numerical experiments on attack instance detection

For a corresponding numerical validation, we consider the `Frand-eff` model in Section F, and consider an attack at time point $t_0 = T/2$ for $K_0 = K/2$ many clients, where their corresponding parameters $\beta_k$ change to $\beta_k' = \beta_k + \mu$. We take $T = 500$, $K = 10$, $\tau = 20$ and for each setting, the above algorithm is run for $B = 500$ bootstrap samples. The empirical power of the described algorithm is reported below, based on 500 Monte-carlo simulations.

| $\mu$ | Probability of detection | Attack instance (mean, 95% CI) | Stopping time $\hat{T}_0$ (mean, 95% CI) |
|---|---|---|---|
| 0 (No attack) | 0.046 (False positive) | – | – |
| 0.5 | 0.172 | 201.233, (58.75, 290.875) | 400.67, (141.25, 496.875) |
| 1 | 0.966 | 265.203, (144.05, 309) | 412.49, (345.05, 482) |
| 1.5 | 1 | 255.772, (155.9, 310.525) | 389.61, (292.475, 470.525) |
| 2 | 1 | 249.672, (118.85, 286.575) | 356.32, (311.475, 397.525) |
| 2.5 | 1 | 247.098, (113.8, 281) | 343.39, (294.95, 379.525) |
| 3 | 1 | 249.572, (94.275, 276) | 334.57, (282.95, 367) |

Table 4: Simulation results of attack detection.

Clearly, the higher the severity of the attack ($\mu$ being large), the more probable it is to be detected, and the quicker it gets detected. Moreover, the estimated attack time also stabilizes around the correct attack instance. Finally, we note that the algorithm can be modified to perform the sequential test only at the synchronization steps, instead of testing for all $t$.

### G.2.1 Experiments based on MNIST dataset

As a further application of Algorithm 2, we work on a federated learning (FL) setup with $K = 5$ clients collaboratively training a linear classifier on **MNIST** data. Let each image be $x_i \in \mathbb{R}^{28 \times 28}$. To get rid of high-dimensionality, a PCA transform $P : \mathbb{R}^{28 \times 28} \to \mathbb{R}^d$ with $d = 3$ is fitted on the full training set $z_i = P(x_i) \in \mathbb{R}^3$. At time point $t$, each client $k \in [K]$ sequentially receives $\xi_t^k = (y_t^k, z_t^k)$'s where $y_t^k$ are corresponding digit labels. Following the notation of the paper, the loss function of client $k$ is the *cross-entropy loss*:

$$f_k(W, b; \xi_t^k) = -y_t^k \log(\hat{y}_t^k) - (1 - y_t^k) \log(1 - \hat{y}_t^k), \ \hat{y}_t^k = \sigma(W z_t^k + b),$$

where $\sigma$ is the sigmoid/logit function, and $W \in \mathbb{R}^{10 \times 3}$ and $b \in \mathbb{R}^{10}$, so the parameter vector $\theta = \text{Vec}([W : b]) \in \mathbb{R}^d$ with $d = 40$. The weight parameters for the aggregated objective function are $w_k = K^{-1}, k \in [K]$, and the step-size is $\eta_t = 0.3t^{-0.75}$ in conjunction with our empirical exercises. The synchronization parameter is $\tau = 5$, and we use the connection matrix from the simulation: $\mathbf{C}_{ij} = 1/3I\{|j - i| \leq 1\}$. Finally, we run the local SGD iterates for $n = 200$ iterations.

For a randomly selected set of $K_0 = 3$ clients, a *label-flipping attack* (the label of image of digits $1, 2, 4$ are switched to $7, 5, 8$, and vice-versa) is injected at $t = 50$, and Algorithm 2 is employed to detect this attack. The matrices $A$ and $\mathcal{V}_K$ are inputs to this algorithm, and are therefore estimated by a pre-traininng/warm-start transient phase. The results are summarized in the following table.

|  | Probability of detection | Attack instance $\hat{s}_0$ (mean, 95% CI) | Stopping time $\hat{T}_0$ (mean, 95% CI) |
| --- | --- | --- | --- |
| No attack | 0.06 | – | – |
| Label flipping attack | 0.90 | 58.49, (31.7, 96.25) | 95.67, (89.1, 98.9) |

Table 5: Simulation results for label-flipping attack detection.

We note that the empirically validity under level $0.05$ is approximately maintained, and the algorithm also achieves high detection power under the label-flipping attack. We remark that the detection can be made earlier by tuning the constant $c$ in the thresholding step appropriately by eg. cross-validation, but overall this experiment shows the applicability of such Gaussian-bootstrap based algorithm beyond theoretical rates, in practical scenarios.

