# OpenReview forum: "Sharp Gaussian approximations for Decentralized Federated Learning"
_NeurIPS.cc/2025/Conference — NeurIPS 2025 spotlight_

### Official Review · Reviewer_CKRL · 2025-06-23

**Clarity:** 3
**Significance:** 3
**Originality:** 3
**Rating:** 4
**Confidence:** 2

**Summary:**

This paper investigates the statistical properties of local SGD in decentralized federated learning (DFL) settings, focusing not only on convergence but also on refined distributional approximations. The authors propose Berry-Esseen bounds for final iterates and develop time-uniform Gaussian approximations to enable inference and adversarial attack detection. Two bootstrap frameworks, Aggr-GA and Client-GA, are introduced, and extensive simulations validate the theoretical results.

**Questions:**

1.	Are the results still valid under significant client heterogeneity?
2.	All analysis assumes a fixed step size, can the results extend to adaptive or momentum-based variants, e.g. Adam?

**Ethical Concerns:**

["NO or VERY MINOR ethics concerns only"]

**Final Justification:**

My main concern lies in the real-world evaluation. While the theoretical contributions are solid and well-motivated, I believe that a more extensive evaluation on real-world benchmarks is necessary to fully demonstrate the practical impact of this work. The additional experiments in the above response are a welcome step in that direction, but their scope remains limited. Therefore, I have decided to maintain my current score.

That said, I appreciate the theoretical depth of the paper and recognize its potential for future applications. Best wishes for your continued work on this promising line of research.

**Limitations:**

The paper lacks real-world experiments on standard federated learning tasks (e.g., image or text classification).

**Paper Formatting Concerns:**

NA.

**Quality:**

3

**Strengths And Weaknesses:**

**Strengths:**

1.	This paper addresses an important gap in the statistical understanding of local SGD beyond convergence.

2.	It provides novel Berry-Esseen bounds and time-uniform Gaussian approximations. The proposed approximations support practical applications such as bootstrap inference and attack detection. Solid empirical validations that match theoretical insights.

3.	Both centralized (Aggr-GA) and decentralized (Client-GA) solutions, can balances accuracy and privacy trade-offs.

**Weaknesses:**

1.	While simulations are extensive, real-world experiments (e.g., NLP or vision FL benchmarks) are lacking.

2.	Presentation could be improved, the notation and organization are occasionally dense and hard to follow.

---

> ### Author Rebuttal · Authors · 2025-07-30
>
> We firstly thank the reviewer for the detailed and encouraging comments on our novelty. We have strived to address each of them in the following. We are happy to provide more explanation if you feel any of your comments is not adequately addressed; we also welcome any other suggestions to help improve the quality of this paper. In case you do not have any other concerns about this manuscript, we would greatly appreciate if you can increase your score. Our rebuttals are as follows.
>
> ### Weaknesses
>
> 1. ***While simulations are extensive, real-world experiments (e.g., NLP or vision FL benchmarks) are lacking.***
>
>    *Answer*:
>    We appreciate the reviewer's important question. Our paper is theoretical in nature, and we do not claim to compete with any SOTA algorithm. We provide two novel, sharp, theoretically valid Gaussian approximations. As the reviewer **6krr** pointed out, our work is the first one to establish strong asymptotic guarantees in a decentralized federated learning setting. We believe that our Gaussian approximation results open door for many potential applications in these areas, but they may require developing case-by-case algorithms to be usable; in that regard, we hope that this opens door to an exciting research direction.
>
>    To illustrate our above point, we have added a real-world experiment on one of these applications from the paper, which we describe in the following paragraph. We work on a federated learning (FL) setup with $K = 5$ clients collaboratively training a linear classifier on *MNIST* data. Let each image be $x_i \in \mathbb{R}^{28\times 28}$. To get rid of high-dimensionality, a PCA transform $P: \mathbb{R}^{28\times 28} \rightarrow \mathbb{R}^d$ with $d = 3$ is fitted on the full training set $z_i = P(x_i) \in \mathbb{R}^3$. At time point $t$, each client $k \in [K]$ sequentially receives $\xi_t^k=(y_t^k, z_t^k)$ where $y_t^k$ are corresponding digit labels. Following the notation of the paper, the loss function of client $k$ is the *cross-entropy loss*:
> $$
> f_k\bigl(W, b; \xi_t^k\bigr)
> = -\,y_t^k\log\bigl(\hat y_t^k\bigr)
>   -\,(1 - y_t^k)\log\bigl(1 - \hat y_t^k\bigr),
> \quad
> \hat y_t^k = \sigma\bigl(W z_t^k + b\bigr).
> $$
>
> where $\sigma$ is the sigmoid function, $W\in\mathbb{R}^{10\times3}$, $b\in\mathbb{R}^{10}$, and hence
>
> $$
> \theta = \operatorname{Vec}\bigl([W : b]\bigr)\in\mathbb{R}^d,\quad d = 40.
> $$
>  The weight parameters for the aggregated objective function are $w_k=K^{-1}, k \in [K]$, and the step-size is $\eta_t=0.3 t^{-0.75}$ in conjunction with our empirical exercises. The synchronization parameter is $\tau=5$, and we use the connection matrix from the simulation: $C_{ij}=1/3 I\{|j-i|\leq 1\}$. Finally, we run the local SGD iterates for $n=100$ iterations.
>
>    For a randomly selected set of $K_0=3$ clients, a *label-flipping attack* (labels of images of digits $1, 2, 4$ are switched to $7,5,8$, and vice-versa) is injected at $t = 50$, and the following Gaussian bootstrap algorithm is employed to detect this attack.
>
> **Algorithm: Attack detection using time-uniform Gaussian bootstrap**
>
> - **Input:** Connection matrix $C$; Synchronization parameter $\tau$; Loss functions $f_k(\theta, \xi^k)$, $\xi^k \sim P_k, k \in [K]$; weights $w_k$; number of iterations $n$; step-size schedules $\eta_{t}$; Hessian $A$; number of bootstrap samples $B$; covariance matrix $V_K$. Type-1 error level $\alpha$.
> -  Initialize $t = 1$. Stopping time $\hat{T}_0 = 1$, estimated attack instance $\hat{s}_0 = +\infty$.
> - While $t \leq n$:
> 1. Store the local SGD iterates $Y_t$, and calculate
>    $$ R_t = \max_{1 \le s\le t} s\bigl|\bar Y_s-\bar Y_t\bigr|
>      ,\quad
>      s_t = \max_{1 \le s\le t} s\bigl|\bar Y_s-\bar Y_t\bigr|.$$
> 2. For $b = 1,\dots,B$:
>    - Draw $Z_t^{(b)} \sim \mathcal N(0, V_K)$, and set
>      $$Y_{t,1}^{G,(b)}
>        = (I - \eta_t A)\,Y_{t-1,1}^{G,(b)} + \eta_t\,Z_t^{(b)}\,K^{-1/2},
>        \quad
>        Y_{0,1}^{G,(b)} = 0.$$
>    - Calculate
>      $$R_t^{G,(b)}=\max_{1 \le s\le t}s\bigl|\bar Y_{s,1}^{G,(b)} - \bar Y_{t,1}^{G,(b)}\bigr|.$$
> 3. Compute
>    $\hat q_{1-\alpha}(t)$ = sample‑quantile $(R_t^{G,(b)})_{b=1}^B.$
>
> 4. **Thresholding:**
>    If
>    $$R_t > \hat q_{1-\alpha}(t) + c\,\sqrt n$$
>    then set $\hat T_0 = t$, $\hat s_0 = s_{\hat T_0}$ and **stop**,
>    otherwise $t\leftarrow t+1$.
>
> - Output:
> $$\hat T_0 I_{\{\hat T_0 < n\}}\; s_{\hat T_0}.$$
>
> The matrices $A$ and $V_K$ are inputs to this algorithm, and are therefore estimated by a pre-training/warm-start transient phase. The results are summarized in the following table.
>
>    |                      | Probability of detection   | Attack instance $\hat{s}_0$  (mean, 95% CI)     | Stopping time $\hat{T}_0$ (mean, 95% CI) |
>    |----------------------|---------------------------|--------------------------------------------------|-------------------------------------------|
>    | No attack            | 0.06 (false positive detection) | -                                   | -                                         |
>    | Label flipping attack| 0.90                      | 58.49, (31.7, 96.25)                           | 95.67, (89.1, 98.9)                       |
>
>    We note that the empirical validity under level $0.05$ is approximately maintained, and the algorithm also achieves high detection power under the label-flipping attack. We remark that the detection can be made earlier by tuning the constant $c$ in the thresholding step appropriately by e.g. cross-validation, but overall this experiment shows the applicability of such Gaussian-bootstrap based algorithms in practical scenarios.
>
> 2. ***Presentation could be improved, the notation and organization are occasionally dense and hard to follow.***
>
>    *Answer*: We sincerely regret that the paper was hard to follow. Our paper presents novel Gaussian approximation results, and so we want to retain their full mathematical statements with all the notations, for the readers more familiar with the literature. However, to improve readability for a broader audience, we have taken the following steps:
>    - **Informal Statements**. We present two “Informal Theorems” (Theorems 1.1 and 1.2) early in the paper, which summarizes the key contributions without notational obscurity. This gives non-technical readers immediate intuition before encountering the full statements.
>    - **Notations and Organization** We have added additional comments to re-introduce/explain many notations/contexts whenever they are used. For example, at the start of Section 2.2.1., we have modified:
>
>    > ``In Theorem 2.1, the ```local SGD``` updates are scaled by the **covariance matrix** $\Sigma_n$, which is not usually known or estimable.''
>
> The statement of the Theorem 3.2 is also modified.
>
> > ``Recall the client-wise gradient noise $W_k$ from Theorem 3.1. Under the assumptions of Theorem 3.1....''
>
> On the organizational front, we have added a new Section before Section 2, named ``Gaussian approximations in DFL: An introduction'' that serves as an extension of Section 1.1., providing mild introduction to our set-up, and explaining key assumptions, and implications of our results, in terminologies aimed at a broader audience. We hope that these changes enhance clarity and presentation adequately for a general audience.
>
> ### Questions
>
> 1. ***Are the results still valid under significant client heterogeneity?***
>
>    *Answer*: Yes, the results are valid even under client heterogeneity. As an example, we show the following generalized version of our Theorem 2.1 and verify it empirically.
>
>    **Generalization of Theorem 2.1.**
>    Let the heterogeneity parameter $\kappa$ be such that $\max_\theta \sum_{k=1}^K w_k \|\nabla F(\theta) - \nabla F_k(\theta)\|^4 \leq \kappa^4$ for some $\kappa \geq 0$. Then, under the assumptions of Theorem 2.1 in the main manuscript it follows that:
> $$
>    d_{\mathrm{C}}( \sqrt{n}(\bar{Y}_n - \theta_K^\star) , Z)
>    \lesssim
>    (\kappa \vee \kappa^2+1)\left(\frac{1}{\sqrt{nK}}  + n^{\frac{1}{2}-\beta} \sqrt{K} + \frac{n^{-\frac{\beta}{2}}}{\sqrt{K}}\right),
> $$
> Clearly, as the heterogeneity increases, the Berry-Esseen error rates worsens polynomially. Similar results are proved for the time-uniform Gaussian approximations. The effect of $\kappa$ can also be verified empirically, by noticing that $\gamma$ plays the role of heterogeneity in the ```Frand-eff``` formulation in Section F.1.1 in the supplement. The corresponding simulation results can be seen in the following.
>
>    | $\gamma$ | $n=100$ | $n=200$ | $n=300$ |
>    |----------|---------|---------|---------|
>    | 1        | 0.222   | 0.224   | 0.154   |
>    | 2        | 0.258   | 0.188   | 0.112   |
>    | 3        | 0.322   | 0.322   | 0.302   |
>    | 4        | 0.374   | 0.272   | 0.232   |
>    | 5        | 0.542   | 0.292   | 0.302   |
>
>    *Table: Berry-Esseen error across different levels of heterogeneity under step-size $\eta_t=0.3t^{-0.75}$, number of clients $K=10$, $\tau=2$, $C_{ij}=1/3 I_{|j-i|\leq 1\}$, dimension $d=2$.*
>
> 2. ***All analysis assumes a fixed step size, can the results extend to adaptive or momentum-based variants, e.g. Adam?***
>
>    *Answer*: We respectfully point out that even $\mathcal L_2$ or in probability convergence properties for ADAM is not known; can in fact fail to converge, (Reddi et. al., [*On the Convergence of Adam and Beyond*, ICLR 2018]). Therefore, we think existence of valid Gaussian approximation, or even a central limit theory is much harder to prove, and is heavily conditional on some proof of in-probability convergence of ADAM iterates. However, the proof techniques here give us confidence that it can be extended, albeit non-trivially, to allow for some adaptive momentum-based variants that already have provable convergence guarantee along with almost sure bounds on gradients.
>
> ### Limitations
>
> ***The paper lacks real-world experiments on standard FL tasks (e.g., image or text classification).***
>
> *Answer*:  Answered above in our response to comment 1 of the **Weakness** section

---

> > ### Comment · Reviewer_CKRL · 2025-08-04
> >
> > Thank you for the response. My concerns have been addressed, and I will maintain my score.

---

> ### Author Response · Authors · 2025-08-05
>
> Thank you very much for confirming that we have successfully addressed your concerns. Given that your original concerns have now been resolved, we hope you might consider whether this further clarifies the contribution and impact of our paper sufficiently to justify a slightly higher score. We would also greatly appreciate any additional feedback or reconsideration that might improve the paper in your assessment.

---

> > ### Comment · Reviewer_CKRL · 2025-08-06
> >
> > Thank you for your response.
> >
> > My main concern lies in the real-world evaluation. While the theoretical contributions are solid and well-motivated, I believe that a more extensive evaluation on real-world benchmarks is necessary to fully demonstrate the practical impact of this work. The additional experiments in the above response are a welcome step in that direction, but their scope remains limited. Therefore, I have decided to maintain my current score.
> >
> > That said, I appreciate the theoretical depth of the paper and recognize its potential for future applications.
> > Best wishes for your continued work on this promising line of research.

---

> > > ### Author Response · Authors · 2025-08-06
> > >
> > > Thank you for your response and kind words about our theoretical development. Our paper is more on the theoretical side, but we appreciate your intuition about the practical application, and we agree with the potential scope of more experiments in an extended paper. Given the limited amount of time as well as page limit, we chose to focus on the ensuring the validity of our methods, while adding some real-data application based on your review. Thank you again for your consideration.

---

### Official Review · Reviewer_6krr · 2025-07-03

**Clarity:** 3
**Significance:** 3
**Originality:** 2
**Rating:** 4
**Confidence:** 5

**Summary:**

This paper studies stochastic gradient descent in the framework of decentralized federated learning, and provides a non-asymptotic versions of the Central limit theorem for the convergence of the model parameter to global optimum. In Section 2, the authors generalize Berry-Essen-type bounds for the last iterate of SGD as well as for the Polyak-Ruppert average to the case of federated learning. The proof of this section are based on the framework of randomized concentration inequalities by [Shao and Zhang, 2022].

In Section 3, the authors present two ‘time uniform’ approximations of SGD iterates averaged across clients. That is, they show that strong approximations in the form of Gaussian processes with explicit covariance structures depending on the parameters of the optimization problem can be constructed. The authors derive stochastic bounds on the deviation of the original SGD process from these approximations. The authors motivate their studies of ‘time uniform’ approximations by detecting anomalies in the SGD trajectory

**Questions:**

1. What is the effect of data heterogeneity across different clients in Theorem 2.1? In the current version, I guess, it is completely hidden in the constants, at the same time, it will be very interesting to quantify, how this factor affects the non-asymptotic CLT.

**Ethical Concerns:**

["NO or VERY MINOR ethics concerns only"]

**Final Justification:**

The authors did a good job of obtaining, to the best of my knowledge, an original Berry–Esseen bound for the federated SGD setting. The results are sound, and my main concern was related to a number of typos in the presentation. The authors have promised to revise this part and include additional results to quantify the impact of heterogeneity. Given this modification, I continue to support accepting the paper.

Justification for not giving a higher grade: The main results of the paper rely heavily on the [Shao and Zhang, 2022] approach to deriving Berry–Esseen bounds for nonlinear statistics. Thus, the paper does not create a novel tool for the analysis that enables tackling the federated setting, but rather is the first to adopt existing tools from [Shao and Zhang, 2022] to this setting.

**Limitations:**

Yes

**Paper Formatting Concerns:**

No concerns.

**Quality:**

3

**Strengths And Weaknesses:**

***Strengths***

The authors obtain novel theoretical results in the federated learning setting. To the best of my knowledge, this is the first time Berry-Eseen type bounds have been proven for the federated learning setup. The considered strong approximation result for SGD training trajectories by a Gaussian process is also novel, to the best of my knowledge.

***Weaknesses***
Theorem 2.2 in its current form is unclear to me. First, a well-known asymptotic normality results on the last iterate of SGD suggest that one should consider $(\theta_n - \theta^\star) / \sqrt{\eta_n}$ in order to have a non-trivial limiting variance. Hence, I guess, there is a normalizing factor $n^{\beta/2}$ missing in the left-hand side. Second, the claim in lines 208-209:  "In particular, this rate is slower than the corresponding rate for the Polyak-Ruppert averaged version" is incorrect - when setting $\beta = 1$, you have exactly the same rate $1/\sqrt{n}$. The problem here is not in the rate, rather in the realizability of the corresponding step sizes (in such a case $\eta_k = c/k$ with some lower bound on the constant $c$, which is instance-dependent). Another issue with the last iterate is that it yields a suboptimal limiting covariance matrix, unlike the Polyak-Ruppert procedure.

***Minor points***
1. Statements of "informal" theorems are too vague, especially the one of theorem 1.1. I would indicate arguments in $d_{Berry-Esseen}$, or at least mention in the statement that the Polyak-Ruppert averaged version of SGD is considered;
2. There is a clash of notations related to $A$. It is used both for generic matrix in notations, generic convex set in the definition of the convex distance (see eq. (2.3)), at the same time, due to Theorem 2.1, it has a separate meaning (Hessian at the optimum). Please change.
3. Vectors $g_k$ are intensively used in the statement of theorem 2.1 and below, but are not defined in the main text. From the context I can infer that $g_k (\theta, \xi) = \nabla f_k (\theta, \xi)$. Please fix this discrepancy;
4. Line 188, still statement of Theorem 2.1:
***assume that $\max_{k \in [K]} \mathbb{E}[|g_k * (\theta_K^\star,\xi)|^2] = \mathcal{O}(1)$***
First, there is a typo here, you do not need $*$. Second, writing here $\mathcal{O}(1)$ is very confusing. Please write the assumption you impose formally.
5. Appendix A, line 989: "Assumption A.3 is much weaker compared to assumption A2(p) in [Sheshukova, 2025]" It is not true, Assumption A.3 is exactly assumption A2(p)- (iii) in the mentioned paper. At the same time, assumption A.2 in the current paper is indeed weaker than the a.s. Lipshitzness imposed in A2(p)-(ii) in [Sheshukova, 2025].
5. What is $E$ in the line 1054 in appendix?
6. There is a $\nabla$ symbol missing in assumption A.2, see line 979
7. Some constructions are clearly too verbose. As a particular example: line 257, “Hence posing a hindrance to performing” --> why not just "hindering"? As I have already mentioned in the weaknesses section, I recommend the authors to re-read the main text carefully and update the ambiguous or verbose phrasing.

---

> ### Author Rebuttal · Authors · 2025-07-30
>
> We sincerely thank the reviewer for the detailed comments, particularly regarding our mathematical results and proofs. We are grateful for your recognition that our paper provides the *first* Berry-Esseen-type bound and strong approximation result for local SGD training trajectories. We warmly welcome any additional suggestions you may have to enhance our manuscript. If our rebuttal adequately addresses your concerns, we kindly request you to consider increasing your score. Now let us address the comments individually.
>
> ### Weaknesses
>
> ***Theorem 2.2 in its current form is unclear to me....***
>
> *Answer*.
> - Your observation is indeed correct about the correct scaling factor. It was a typo on our part that we kept the scaling factor $n^{-\beta}$ multiplied to the variance. We have added the correct version of Theorem 2.2:
>
> **Theorem 2.2**: Under the assumptions of Theorem 2, it holds that
> $$
> d_{\mathrm C}( n^{\beta/2}(Y_n - \theta_K^\star) , Z)  \lesssim \frac{n^{-\beta/2}}{\sqrt{K}}  + n^{\frac{1}{2}-\beta} \sqrt{K},
> $$
> where $Z \sim N(0, \tilde \Sigma_n)$ with
> $$
> \tilde \Sigma_n := n^{\beta}\sum_{s=1}^n \operatorname{Var}\left(\mathcal A_s^n \sum_{k=1}^K \eta_{s,k}w_k g_k(\theta_K^\star, \xi_s^k)\right).
> $$
>
> - Regarding the subsequent remark, we mentioned the slower rate since usually we restrict the $\beta \in (1/2, 1)$; only in the limiting case of $\beta=1$ do we have the $n^{-1/2}$ scaling. However, again you are absolutely right in pointing this out, and we have accordingly removed this comment. We are thankful to the reviewer for mentioning the point of suboptimal limiting covariance matrix, which we have added now.
>
> ### Minor Comments
>
> 1. ***Statements of "informal" theorems are too vague, especially the one of theorem 1.1. I would indicate arguments in $d_{Berry-Esseen}$, or at least mention in the statement that the Polyak-Ruppert averaged version of SGD is considered;***
>
>    *Answer*: We provided the informal statements to ease those readers into our paper who may not be entirely familiar into this literature, and we are sorry if we erred on the side of being too vague. In cognizance of your suggestion, we have made the following modifications.
>
> **Theorem 1.1, informal:** For a decentralized federated learning set-up with $K$ clients, the Polyak-Ruppert averaged iterates $\bar Y_n$ of the local SGD algorithm with $n$ iterations, and step size $\eta_t \asymp t^{-\beta}$, achieves
>    $$
>    \texttt{Berry-Esseen error}\lesssim n^{1/2 - \beta}\sqrt{K}.
>    $$
>
>    **Theorem 1.2, informal:** If the local SGD algorithm with $K$ clients runs $n$ iterations with step size $\eta_t \asymp t^{-\beta}$, then there exists a Gaussian process $Y_t^G = (I-\eta_tA) Y_{t-1}^G + \eta_t Z_t$ with $Z_t$ i.i.d. $N(0,\Gamma)$ for some matrix $\Gamma$ and $A$ being the Hessian of the problem, such that for the local SGD iterates $Y_n$, it holds,
>    $$
>    \max_{1\leq t \leq n} \left| \sum_{s=1}^t (Y_s - Y_s^G) \right| \approx o_{\mathbb{P}}(n^{1-\beta} + \frac{n^{1/p}}{\sqrt{K}} ),
>    $$
>    where $p\geq 2$ denotes the maximum number of finite moments the data generating mechanism of each client has.
>
> 2. ***There is a clash of notations related to $A$. It is used both for generic matrix in notations, generic convex set in the definition of the convex distance (see eq. (2.3)), at the same time, due to Theorem 2.1, it has a separate meaning (Hessian at the optimum). Please change.***
>
>    *Answer*: We have replaced the $A$ in the notation section by "for $M \in \mathbb{R}^{m\times n}$, $\lVert M\rVert_F$ denotes its Frobenius norm." and the $A$ in equation (2.3) by
>    $$
>    d_{\mathrm C}(Y,Z):= \sup_{\aleph \in \mathcal B(\mathbb R^d): A \text{ convex}} \left|\mathbb P(Y \in \aleph)- \mathbb P(Z \in \aleph) \right|.
>    $$
>    This keeps the notation $A$ only reserved for the Hessian. We are sorry for the ambiguity caused.
>
> 3. ***Vectors $g_k$ are intensively used in the statement of theorem 2.1 and below, but are not defined in the main text. From the context I can infer that $g_k(\theta, \xi) = \nabla f_k(\theta, \xi)$. Please fix this discrepancy..***
>
>    *Answer*: Thanks for catching this. Actually the vectors $g_k$ are mean-zero versions of the noisy gradient (referred to as gradient noise), and are given by $g_k(\theta, \xi) = \nabla F_k(\theta) - \nabla f_k(\theta, \xi)$. We have clarified this in the statement of the Theorem 2.1, where it were earlier used without definition.
>
> 4. ***Line 188, still statement of Theorem 2.1 "assume that
> $
> \max_{k \in [K]} \mathbb E[|g_k\star (\theta^{\star}_K, \xi)|^2] = O(1)
> $"
> First, there is a typo here, you do not need a $\star$. Second, writing here $O(1)$ is very confusing. Please write the assumption you impose formally.***
>
>    *Answer*: We have modified the statement in Theorem 2.1 as follows.
>
>  **Theorem 2.1.** Define $\mathcal A_s^t := \prod_{j=s+1}^t (I - \eta_t A)$, $\mathcal A_t^t = I$, where $A := \nabla_2 F(\theta_K^\star)$ for $t \in [n]$. Further, for $s \in [n]$, define the random vectors
> $$
> u_s = \eta_s \sum_{k=1}^K w_k \left( \sum_{j=s}^n \mathcal A_s^j \right) g_k(\theta_K^\star, \xi_s^k),
> $$
> with
> $$
> \Sigma_n := n^{-1} \sum_{s=1}^n \mathbb E[u_s u_s^\top], \quad g_k(\theta, \xi^k) = \nabla F_k(\theta) - \nabla f_k(\theta, \xi^k).
> $$
>
> Let there exist a constant $C$ such that for $\xi^k \sim \mathcal P_k$, $k \in [K]$, it holds $\max_{k \in [K]} \mathbb E[|g_k(\theta_K^\star, \xi^k)|^2] \leq C$. Suppose that the step-size schedules of the clients satisfy $\eta_t = \eta_0 (t + k_0)^{-\beta}$ for some fixed $\eta_0, k_0 > 0$, and $\beta \in (1/2, 1)$. Then, under Assumptions 2.1, A.1, A.2 and A.3 with $p = 4$, and $\bar Y_n$ as in (2.4), it holds that
>
> $$
> d_{\mathrm C} \left( \sqrt{n} (\bar Y_n - \theta_K^\star),\; Z \right) \lesssim \frac{1}{\sqrt{nK}} + n^{\frac{1}{2} - \beta} \sqrt{K} + \frac{n^{-\frac{\beta}{2}}}{\sqrt{K}},
> $$
>
> where $\lesssim$ hides constants involving $d, \beta, \mu, L$ and $\rho$, and $Z \sim N(0, \Sigma_n)$.
>
> 5. ***Appendix A, line 989: "Assumption A.3 is much weaker compared to assumption A2(p) in [Sheshukova, 2025]" It is not true, Assumption A.3 is exactly assumption A2(p)- (iii) in the mentioned paper. At the same time, assumption A.2 in the current paper is indeed weaker than the a.s. Lipshitzness imposed in A2(p)-(ii) in [Sheshukova, 2025].***
>
>    *Answer*: Again thank you for catching this detail. We indeed meant to write Assumption A.2, and it was a typo which has been corrected.
>
> 6. ***What is $E$ in line 1054 in the appendix? It is undefined.***
>
>    *Answer*: It should be $E_{\tau}=\{\tau, 2\tau, \ldots\}$ which is defined in the Algorithm 1 of our manuscript. We have corrected this.
>
> 7. ***There is a $\nabla$ symbol missing in assumption A.2, see line 979***
>
>    *Answer*: Yes, this has been corrected.
>
> 8. ***Some constructions are clearly too verbose. As a particular example: line 257, “Hence posing a hindrance to performing” --> why not just "hindering"? As I have already mentioned in the weaknesses section, I recommend the authors to re-read the main text carefully and update the ambiguous or verbose phrasing..***
>
>     *Answer*: Thank you again for your careful perusal. Based on this feedback, we have substantially revised the manuscript to make it concise while enhancing clarity and readability. Some of these modifications can be found in our rebuttal to the comment 2 of the **Weakness** section by the reviewer *CKRL*.
>
>
> ### Questions
>
> - ***What is the effect of data heterogeneity across different clients in Theorem 2.1? In the current version, I guess, it is completely hidden in the constants, at the same time, it will be very interesting to quantify, how this factor affects the non-asymptotic CLT.***
>
>    *Answer*: Thank you for asking this important question. This has been one of the most common questions that we received, and accordingly we have generalized Theorem 2.1 to reflect the heterogeneity.
>
>    **Generalization of Theorem 2.1.**
>    Let the heterogeneity parameter $\kappa>0$ be such that $\max_\theta \sum_{k=1}^K w_k \|\nabla F(\theta) - \nabla F_k(\theta)\|^4 \leq \kappa^4$. Then, under the assumptions of Theorem 2.1 in the main manuscript it follows that:
> $$
>    \begin{aligned}
>    d_{\mathrm C}( \sqrt{n}(\bar Y_n - \theta_K^\star) , Z) \lesssim & (\kappa \vee \kappa^2+1)(\frac{1}{\sqrt{nK}}  + n^{\frac{1}{2}-\beta} \sqrt{K} + \frac{n^{-\frac{\beta}{2}}}{\sqrt{K}}),
>    \end{aligned}
> $$
> Clearly, as the heterogeneity increases, the Berry-Esseen error rates worsens polynomially. Similar results are proved for the time-uniform Gaussian approximations.
>
> The effect of heterogeneity can also be verified empirically, by noticing that $\gamma$ plays the role of heterogeneity in the Frand-eff formulation in Section F.1.1 in the supplement. The corresponding simulation results can be seen in the following table:
>
>    |$\gamma$|$n=100$|$n=200$|$n=300$|
>    |-|-|-|-|
>    |1|0.222|0.224|0.154|
>    |2|0.258|0.188|0.112|
>    |3|0.322|0.322|0.302|
>    |4|0.374|0.272|0.232|
>    |5|0.542|0.292|0.302|
>
> *Table: Berry-Esseen error across different levels of heterogeneity under step-size $\eta_t= 0.3t^{-0.75}$, number of clients $K=10$, $\tau=2$, $C_{ij}=1/3 I\{|j-i|\leq 1\}$, dimension $d=2$. All the results are based on 500 Monte-Carlo simulations.*

---

> > ### Comment · Reviewer_6krr · 2025-08-05
> >
> > The authors addressed most of my concerns in their rebuttal. I suggest that the authors carefully proofread the manuscript, correct the outlined typos (as well as any others that may still be present), and include the heterogeneity result.
> >
> > That being said, I prefer to keep my score as is and wish the authors good luck.

---

> ### Author Response · Authors · 2025-08-05
>
> We are grateful for your careful perusal of our rebuttal and your valuable feedback, which, we believe, have improved the manuscript quality substantially. As we mentioned in our rebuttal, we have already done a thorough proofreading of our manuscript to weed out potential typos, and will duly include the heterogeneity result, based on your suggestion. We warmly appreciate your acknowledgment about us having addressed most of your concerns. We have also performed additional numerical experiments as per other reviewers' suggestions. At this point, we politely ask if there is any room that we could further improve our paper and you might consider raising your score.
>
> Thank you very much for your guidance and consideration.

---

### Official Review · Reviewer_mVKL · 2025-07-05

**Clarity:** 3
**Significance:** 4
**Originality:** 4
**Rating:** 5
**Confidence:** 4

**Summary:**

This paper presents sharp Gaussian approximation results for local SGD in decentralized federated learning (DFL), going beyond standard central limit theory to enable statistical inference with finite sample guarantees. The main contributions are a Berry-Esseen theorem for both the final and Polyak-Ruppert averaged iterates, and two distinct time-uniform Gaussian approximations (Aggr-GA and Client-GA) for the entire trajectory of local SGD updates. The theoretical results are validated through extensive simulations demonstrating improved approximation rates compared to traditional methods and revealing key computation-communication trade-offs.

**Questions:**

1. How do the approximation rates scale with non-IID data distributions across clients?
2. What are the computational trade-offs between Aggr-GA and Client-GA in practice?
3. How robust are the results to violations of the strong convexity assumption?
4. Could the methods be extended to non-convex settings?

**Ethical Concerns:**

["NO or VERY MINOR ethics concerns only"]

**Final Justification:**

Thanks to the authors clarification in rebuttal and I keep the same rating for their submission.

**Quality:**

4

**Strengths And Weaknesses:**

Strengths:
- Strong technical novelty in deriving first Berry-Esseen bounds and time-uniform Gaussian approximations for local SGD
- Rigorous theoretical analysis with comprehensive proofs
- Extensive empirical validation of theoretical claims through well-designed experiments
- Clear distinction from prior work by going beyond central limit theory
- Direct implications for statistical inference and anomaly detection in DFL

Weaknesses:
- Limited discussion of computational complexity of the proposed methods
- Lacks ablation studies comparing different variants of the Gaussian approximations
- Could benefit from more real-world applications beyond synthetic experiments
- Synchronization effects could be explored more thoroughly
- Some assumptions like strong convexity may be restrictive

---

> ### Author Rebuttal · Authors · 2025-07-31
>
> We sincerely thank the reviewer for detailed comments and encouraging words on our technical novelty, originality and practical applications. We address all of your concerns below. We also gladly welcome any more suggestions to improve this paper. Finally, in the event you no longer have any concerns or queries, we would greatly appreciate if you can consider increasing the score of the paper.
>
> ### Weaknesses
>
> 1. ***Limited discussion of computational complexity of the proposed methods***
>
> *Answer*:  We address this in our response to comment 2 on your **Question** section.
>
> 2. ***Lacks ablation studies comparing different variants of the Gaussian approximations***
>
>  *Answer*: To address this important point, we include two more simulation studies as follows. Recall that
> $$
> U_n=\max_{1\leq t\leq n}\left|\sum_{s=1}^t(Y_s-\theta_K^\star)\right|,\quad
> U_n^{\texttt{Aggr-GA}}=\max_{1\leq t\leq n}\left|\sum_{s=1}^tY_{s,1}^G\right|,\quad
> U_n^{\texttt{Client-GA}}=\max_{1\leq t\leq n}\left|\sum_{s=1}^tY_{s,2}^G\right|
> $$
> and $U_n^{\texttt{f-CLT}}=\max_{1\leq t\leq n}\left|\sum_{s=1}^tZ_s\right|$, where $Z_1,\ldots,Z_n\overset{i.i.d.}{\sim}N(0,\Sigma)$, $Y_{s,1}^G$ and $Y_{s,2}^G$ are defined as in Theorem 3.1 and 3.2, and $\Sigma$ is as in Section 2.2.1.
>
> In the first experiment, we fix $N=500,K=15,\gamma=1$, and vary $\tau=5,10,15$. For each setting, we compute the following:
> $$
> Q_{\texttt{f-CLT}} = \max_{\alpha \in (c,1-c)} \frac{|q_{1-\alpha}(U_n)-q_{1-\alpha}(U_n^{\texttt{f-CLT}})|}{q_{1-\alpha}(U_n)},\ Q_{\texttt{Aggr-GA}}=\max_{\alpha\in(c,1-c)} \frac{|q_{1-\alpha}(U_n) - q_{1-\alpha}(U_n^{\texttt{Aggr-GA}})|}{q_{1-\alpha}(U_n)},
> $$
> and $$Q_{\texttt{Client-GA}}=\max_{\alpha\in(c,1-c)}\frac{|q_{1-\alpha}(U_n)-q_{1-\alpha}(U_n^{\texttt{Client-GA}})|}{q_{1-\alpha}(U_n)}$$ for a small $c>0$. The following table summarizes the results.
>
> |$\tau$|$Q_{\texttt{f-CLT}}$|$Q_{\texttt{Aggr-GA}}$|$Q_{\texttt{Client-GA}}$|
> |-|-|-|-|
> |5|1.495|0.214|0.327|
> |10|2.009|0.44|0.53|
> |15|2.476|0.663|0.883|
>
> Clearly, the efficacy of each Gaussian approximation worsens with increasing $\tau$; moreover, the two Gaussian approximations proposed, `Aggr-GA` and `Client-GA`, consistently outperform a functional-CLT-based approach, mirroring our results from Section 4.3.
>
> We also fix $N=500,K=15,\tau=20$, and vary the heterogeneity parameter $\gamma=1,5,10$. The results are as follows.
>
> |$\gamma$|$Q_{\texttt{f-CLT}}$|$Q_{\texttt{Aggr-GA}}$|$Q_{\texttt{Client-GA}}$|
> |-|-|-|-|
> |1|1.047|0.237|0.275|
> |5|2.047|0.605|0.646|
> |10|2.805|0.547|0.782|
> We again see the worsening performance of the Gaussian approximations with increasing heterogeneity. We believe these two experiments provide a clear ablation study on the empirical performances of our proposed approximations.
>
> 3. ***Could benefit from more real-world applications beyond synthetic experiments***
>
> *Answer*: As an application of our time-uniform Gaussian approximation results (Theorem 3.1 and 3.2), we have added a real-world experiment in the paper. We describe this in the following. We work on a federated learning (FL) setup with $K=5$ clients collaboratively training a linear classifier on MNIST data. Let each image be $x_i\in\mathbb{R}^{28\times 28}$. To get rid of high-dimensionality, a PCA transform $P:\mathbb{R}^{28\times 28}\rightarrow\mathbb{R}^d$ with $d=3$ is fitted on the full training set $z_i=P(x_i)\in\mathbb{R}^3$. At time $t$, each client $k\in[K]$ sequentially receives $\xi_t^k=(y_t^k,z_t^k)$ where $y_t^k$ are the corresponding digit labels. The loss function of client $k$ is the cross-entropy loss:
> $$
> f_k(W,b;\xi_t^k)=-y_t^k\log(\hat y_t^k)-(1-y_t^k)\log(1-\hat y_t^k),\quad\hat y_t^k=\sigma(Wz_t^k+b),
> $$
> where $\sigma$ is the sigmoid/logit function, and $W\in\mathbb{R}^{10 \times 3}$ and $b\in \mathbb{R}^{10}$, so the parameter vector $\theta=\operatorname{Vec}([W:b])\in\mathbb{R}^{d}$ with $d=40.$ The weight parameters for the aggregated objective function are $w_k=K^{-1},\ k\in[K]$, and the step-size is $\eta_t=0.3t^{-0.75}$. The synchronization parameter is $\tau=5$, and we use the connection matrix $\mathbf{C}_{ij}=1/3I\{|j-i|\leq 1\}$. Finally, we run the local SGD iterates for $n=100$ iterations.
>
>  For a randomly selected set of $K_0=3$ clients, a label-flipping attack (the label of images of digits $1,2,4$ are switched to $7,5,8$, and vice-versa) is injected at $t = 50$. We have described the algorithm (named **Attack detection using time-uniform Gaussian bootstrap**) we employ to detect such attacks at our rebuttal to the **Limitation** section of the reviewer *PXEE*. The results are summarized in the following table.
>
> ||Probability of detection|Attack instance $\hat s_0$ (mean, 95% CI)|Stopping time $\hat T_0$ (mean, 95% CI)|
> |-|-|-|-|
> |No attack|0.06 (false positive)|-|-|
> |Label flipping attack|0.90|58.49, (31.7, 96.25)|95.67, (89.1, 98.9)|
>
> Note that empirical validity at the $0.05$ level is approximately maintained, and high detection power is achieved under the label-flipping attack. Earlier detection can be obtained by suitably tuning the constant $c$ in the threshold step (e.g., via cross-validation). Overall, these experiments demonstrate the practical applicability of Gaussian-bootstrap-based algorithms.
>
> 4. ***Synchronization effects could be explored more thoroughly***
>
> *Answer*: Addressing of your question, we have the following general version of the Theorem 2.1, where we allow the synchronization parameter $\tau$ to grow linearly with $n$.
>
> **Generalization of Theorem 2.1.**
>   As long as $\tau \leq \theta n$, for some $\theta \in (0,1)$, under the assumptions of Theorem 2.1 in the main manuscript it follows that:
> $$
>    d_{\mathrm{C}}(\sqrt{n}(\bar{Y}_n-\theta_K^\star),Z)\lesssim\left(\sqrt{\tau}+\frac{1}{\sqrt{1-\rho^{1/\tau}}}\right)\left(\frac{1}{\sqrt{nK}}+n^{\frac{1}{2}-\beta} \sqrt{K}+\frac{n^{-\frac{\beta}{2}}}{\sqrt{K}}\right)
> $$
> Clearly, as the number of local updates $\tau$ increases, the errors increase with a $\sqrt{\tau}$ rate. This has also been verified empirically in the following table.
>
> |$\tau$|$n=100$|$n=200$|$n=300$|
> |-|-|-|-|
> |20|0.118|0.146|0.155|
> |40|0.155|0.183|0.185|
> |60|0.176|0.225|0.220|
> |80|0.176|0.230|0.252|
> |100|0.175|0.241|0.261|
>
> 5. ***Some assumptions like strong convexity may be restrictive***
>
> *Answer*: Even though our theoretical results use strong convexity for convenience, in the supplementary Section A.2, we explicitly mention how the strong convexity assumption can be replaced by a much weaker, "local concordance property". For example, Logistic regression violates strong convexity but obeys local concordance property.
>
> ### Questions
>
> 1. ***How do the approximation rates scale with non-IID data distributions across clients?***
>
> *Answer*: We respectfully point out that **our results do not require IID data for each client**. In particular, we mention in **Section 2.1, Preliminaries**:
>
>    > "Here, $\mathcal{P}_k$ determines the distribution of the local noisy gradient for each client, realized by sampling $\xi^k\sim\mathcal{P}_k$. We allow for heterogeneity among the clients, i.e. $\mathcal{P}_k$'s are allowed to be different. However, noise sampling (i.e., the $\xi^k$) is assumed to be independent from one client to another."
>
> That said, if you think we misunderstood your question or further clarifications are required, we would appreciate further comments from you.
>
> 2. ***What are the computational trade-offs between Aggr-GA and Client-GA in practice?***
>
> *Answer*:  We thank the reviewer for raising this important point. At each iteration $t$, Aggr-GA has a computational complexity of $O(d^2)$, since it involves generating one random sample followed by a matrix-vector multiplication. But implementing this require complete peer-to-peer sharing. In contrast, Client-GA has a complexity of $O(Kd^2)$ per iteration. However, the structure of Client-GA naturally allows for parallel computation between synchronization steps, significantly reducing the computational burden while preserving periodic, *partial*, peer-to-peer communication.
>
> 3. ***How robust are the results to violations of the strong convexity assumption?***
>
> *Answer*: Answered above in response to comment 5 in the **Limitations** section.
>
> 4. ***Could the methods be extended to non-convex settings?***
>
> *Answer*: If there are only finitely many local minimas, our techniques along with *local strong convexity* assumption can yield the corresponding results conditional on in-probability convergence. See for example Zhong et. al., [*Online bootstrap inference with non convex stochastic gradient descent estimator*, Preprint 2023]. Unfortunately, for non-convex regimes with uncountably many local minimas, it is unclear what the central limit theory would look like even for vanilla SGD. This indeed represents an exciting research direction, and we believe, despite the significant non-triviality, the techniques developed here should be useful there too.

---

> > ### Author Response · Authors · 2025-08-07
> > **Appreciation for your review; follow-up on responses**
> >
> > Dear Reviewer,
> >
> > We hope you have had a chance to review our responses to your thoughtful comments. We sincerely appreciate your time and effort into evaluating our work.
> >
> > If you have any remaining questions or suggestions, we would be very happy to clarify or address them. We also hope that our responses adequately resolved your concerns. If so, we would be truly grateful if you would consider updating your score accordingly.
> >
> > Thank you again for your constructive feedback and for supporting the review process.
> >
> > Warm regards, The Authors.

---

### Official Review · Reviewer_SopJ · 2025-07-08

**Clarity:** 2
**Significance:** 3
**Originality:** 3
**Rating:** 4
**Confidence:** 3

**Summary:**

In this paper, the authors present two generalized Gaussian approximation results for decentralized federated learning with local updates and explore their theoretical and practical implications. The theoretical development appears solid and technically nontrivial.

**Questions:**

My major comments are as follows:

1. Motivation

The authors should elaborate on why traditional convergence analysis (i.e., focusing on optimization error) is not sufficient in modern decentralized learning settings. It is important to explain why the statistical properties investigated in this paper are important. A more thorough discussion connecting statistical properties with optimization and generalization error would greatly benefit readers unfamiliar with this line of work.

2. Choice of Online Setting

The paper considers an online federated setting, rather than the more common federated learning scenario where each client holds a local dataset. It would be valuable to consider, or at least discuss, how the proposed theoretical results would extend to or differ in the standard offline setting. Additionally, analyzing how sample size, local update number, and total iteration number affect the statistical properties in such settings would be more general and practically meaningful.

3. Clarification on Decentralization

The paper claims to investigate decentralized federated learning, but repeatedly refers to a ``moderator'' that aggregates client information and redistributes it according to some policy in Sections 2.1 and 2.1.1. This resembles centralized federated learning more than a truly decentralized settings. In typical decentralized Local SGD, messages are shared peer-to-peer over a communication graph.  It should be clearly justified and discussed.

4. Dependence on $\tau$, Network Topology, data heterogeneity

The authors mention that the data can be heterogeneous and describe the network topology in Section 2. However, none of the main theoretical results appear to explicitly capture the impact of $\tau$ (the number of local updates), the network topology, or data heterogeneity. The absence of such dependencies should be discussed.

5. Privacy and Robustness Claims Lack Support

The authors claim in Section 3 that analyzing the full trajectory of Local SGD can help maintain privacy and detect adversarial attacks. However, I was unable to find any formal results, theoretical justifications, or empirical validations to support these claims. More detailed discussion and supporting experiments are necessary.

**Ethical Concerns:**

["NO or VERY MINOR ethics concerns only"]

**Final Justification:**

Most of my concerns have been addressed, and I have accordingly raised my score.

**Limitations:**

Yes

**Quality:**

3

**Strengths And Weaknesses:**

In this paper, the authors present two generalized Gaussian approximation results for decentralized federated learning with local updates and explore their theoretical and practical implications. The theoretical development appears solid and technically nontrivial. However, the paper is somewhat difficult to follow. Many important concepts and theorem statements are introduced without sufficient explanation or discussion, making it challenging for readers who are not already familiar with this specific area to fully grasp the contributions and their implications.

---

> ### Author Rebuttal · Authors · 2025-07-31
>
> We are thankful to the reviewer for a detailed perusal as well as the positive comments on the theoretical part. Below, we address your concerns individually. We warmly welcome any further suggestions you may have to improve the paper. Finally, in case you find this rebuttal adequate, we humbly request you to consider increasing your score.
>
> ### Weaknesses
>
> ***Readability and accessibility***
>
> *Answer*: We sincerely regret that the paper was difficult to follow. Based on this comment, we have substantially revised the paper to include more explanations, implications and remarks to improve clarity. Some of these modifications can be found in our response to the *Weakness*, point 2, by the reviewer *CKRL*.
>
> ### Questions
>
> 1. ***Motivation***
>
> *Answer*: We thank the reviewer for this insightful suggestion. Indeed, traditional analyses typically focus on quantities such as $|Y_n-\theta_K^\star|$ or $|F(Y_n)-F(\theta_K^\star)|$, without addressing statistical properties explicitly. Optimization and generalization error bounds alone are insufficient for developing valid inferential tools. Distributional results like asymptotic normality explicitly quantify the randomness of DFL iterates, thereby enabling uncertainty quantification, confidence procedures, and hypothesis testing. Sensitivity analysis (using ROC curve) is an increasing concern of many federate-learning applications (Schneble and Thamilarasu, [*Attack Detection Using Federated Learning in
> Medical Cyber-Physical Systems*, ICCN, 2021]; Dayan et. al., [*Federated learning for predicting clinical outcomes in patients with COVID-19*, Nature Medicine 2019]). Our results reinforce these statistical motivations.
>
> 2. ***Choice of Online Setting***
>
> *Answer*: We are not aware of any central limit theory in offline gradient descent (GD) for e.g. M-estimation, let alone in a federated setting; whereas central limit theory for SGD is classical (Polyak and Juditsky, [*Acceleration of stochastic approximation by averaging*, SIAM Optimization, 1992]). This is because, once you fix the empirical objective, the GD trajectory becomes deterministic, and converges to the empirical minimizer. Therefore, conditional on the offline data, one can only provide an $\mathcal{L}_2$ bound; marginally (i.e. lifting the condition), the statistical property of the GD iterates mimic that of the empirical minimizer.
>
>  Similar argument also holds for the DFL setting. Marginally, the decentralized GD updates $Y_n$ will converge to the maximum likelihood estimate: $\hat \theta_{n,K}= \min_\theta\sum_{k=1}^Kw_k n_k^{-1}\sum_{i=1}^{n_k}f(\theta,\xi_i^{(k)})$. Its statistical properties, e.g. CLT, are classically known. The effect of sample size, local update number, and total iteration number, can be characterized in the usual $\mathcal L_2$ bound $\mathbb E\|Y_n-\hat \theta_{n,k}\|^2$, and has indeed been done in the literature (Stich, [*Local SGD converges fast and communicates little*, ICLR 2019]; Khaled et. al., [*Tighter Theory for Local SGD on Identical and Heterogeneous Data*, AISTATS 2020]).
>
> 3. ***Clarification on Decentralization***
>
> *Answer*: Our set-up is *decentralized*, where, during the synchronization, the messages are shared over the communication graph $\mathbf{C}$ (Section 2.1.1, and Assumption 2.1 in the manuscript) peer-to-peer. A "moderator" is mentioned only to explain the known weights, and the synchronization systems. We agree that none of them require the presence of a moderator, and have removed this word from the paper.
>
> 4. ***Dependence on $\tau$, Network Topology, data heterogeneity***
>
> *Answer*: In what follows, we provide versions of our results with the explicit constants depending on the number of local updates ($\tau$) and the network topology ($\rho$), where $\rho$ is the second largest eigenvalue of the connection graph $\mathbf{C}$.
>
> **Generalization of Theorem 2.1.**
> Let there exist $\kappa>0$ such that it holds $\max_\theta \sum_{k=1}^Kw_k\|\nabla F(\theta)-\nabla F_k(\theta)\|^4 \leq\kappa^4$. As long as $\tau\leq\theta n$, for some $\theta\in(0,1)$, under assumptions of Theorem 2.1 in the main draft, it follows that:
> $$\begin{aligned}
> d_{\mathrm C}(\sqrt{n}(\bar Y_n-\theta_K^\star),Z)\lesssim&(\kappa\vee\kappa^2+1)\Big(\sqrt{\tau}+\frac{1}{\sqrt{1-\rho^{1/\tau}}}\Big)(\frac{1}{\sqrt{nK}}+n^{\frac{1}{2}-\beta} \sqrt{K}+\frac{n^{-\frac{\beta}{2}}}{\sqrt{K}}),
> \end{aligned}$$
> The data heterogeneity is reflected by the $\kappa$ parameter.
>
> As the network topology becomes sparser ($\rho\uparrow1$), the GA error scales as $(1-\rho^{1/\tau})^{-1/2}$. Additionally, as the number of local updates $\tau$ increases, the errors increase with a $\sqrt{\tau}$ rate. Finally, as the heterogeneity increases, the Berry-Esseen error rates worsens polynomially. Similar results are proved for the time-uniform Gaussian approximations.
>  These theoretical intuitions can be verified empirically. In particular, note that the effect of heterogeneity can also be tracked through the effect of $\gamma$ in the `Frand-eff` model in Section F.1.1 in the supplement. The corresponding simulation results for $\rho,\tau$ and $\kappa$ are as follows.
>
> **Comparison of Berry-Esseen error under step-size $\eta_t= 0.3t^{-0.75}$, number of clients $K=10$, dimension $d=2$.**
>
> |$\rho$|$n=100$|$n=200$|$n=300$|
> |-|-|-|-|
> |0.2|0.106|0.107|0.104|
> |0.4|0.136|0.143|0.138|
> |0.6|0.165|0.192|0.175|
> |0.8|0.203|0.245|0.250|
> *Table: For each $\rho$, $\mathbf C= \rho I_K + (1-\rho)K^{-1}\mathbf{1}\mathbf{1}_K$.*
>
> |$\tau$|$n=100$|$n=200$|$n=300$|
> |-|-|-|-|
> |20|0.118|0.146|0.155|
> |40|0.155|0.183|0.185|
> |60|0.176|0.225|0.220|
> |80|0.176|0.230|0.252|
> |100|0.175|0.241|0.261|
>
> **Table: Here $
> \mathbf C_{ij} = \frac{1}{3}  I_{\,|j - i| \leq 1}
> $**
>
> |$\gamma$|$n=100$|$n=200$|$n=300$|
> |-|-|-|-|
> |1|0.222|0.224|0.154|
> |2|0.258|0.188|0.112|
> |3|0.322|0.322|0.302|
> |4|0.374|0.272|0.232|
> |5|0.542|0.292|0.302|
>
> **Table: Here $\tau=20$.**
>
> 5. ***Privacy and Robustness Claims Lack Support***
>
> *Answer*: A brief mathematical justification is given in Section 3.1, equation (3.1) and (3.2) of the original draft. We elaborate it here.
>
>  Assume that at time $t_0$, $\mathcal K_0$ clients become malicious. This model poisoning can be mathematically described by a change in their local risk functions $F_k,k\in\mathcal K_0$, which affects the distribution of the local SGD updates $Y_t$. To identify the time-point $t_0$ sequentially, we examine the widely-used CUSUM statistic $R_{t}:=\max_{1\leq s\leq t}s|\bar Y_s-\bar Y_t|$. We expect $R_{t}$ to be large for $t>t_0$ if an attack has altered the mean behavior of iterates. The null distribution (i.e. when no attack takes place) of $R_t$ is usually mathematically intractable, hence hindering valid inference. This necessitates a bootstrap procedure, e.g. leveraging Theorem 3.1, which we illustrate below. Let the type-one-error $\alpha$, be given.
>     Recall the aggr-GA approximation $Y_{s,1}^G$, and define $\mathcal G_t=\sum_{s=1}^tY_{s,1}^G$. Then, with $R_t^G:=\max_{1\leq s\leq t}|\mathcal G_s-\frac{s}{t}\mathcal G_t|,$ it follows from Theorem 3.1 that,
> $n^{-1/2}\max_{1\leq t\leq n}|R_t-R_t^G|\leq2\max_{1\leq t\leq n}|t\bar Y_t-t\theta_K^\star-\mathcal G_t|=o_{\mathbb P}(1).$
> Therefore, if $Q_{1-\alpha}(X)$ denotes the $(1-\alpha)$-th quantile of random variable $X$, then for a suitable positive sequence $\{a_n\}$,
> $$\mathbb P(R_t>Q_{1-\alpha}(R_t^G)+a_n\text{ for some }t\in[n])\leq\alpha+\mathbb P(\max_{1\leq t \leq n}|R_t-R_t^G|>a_n)\to\alpha,$$
> as long as $n^{-1/2}a_n\geq c$. Based on this, a bootstrap algorithm follows:
>
> **Algorithm: Attack detection using time-uniform Gaussian bootstrap**
>
> - **Input:** Connection matrix $C$; Synchronization parameter $\tau$; Loss functions $f_k(\cdot,\xi^k),\ \xi^k\sim\mathcal P_k,\ k\in [K]$; weights $w_k, k=1,.., K$; number of iterations $n$; step-sizes $\eta_t$; Hessian $A$; number of bootstrap samples $B$; covariance matrix $V_K$, level $\alpha$.
>
> - Let $E_{\tau}=\{\tau,2\tau,\dots,L\tau\}$, where $L=\lfloor n/\tau\rfloor$. Initialize $t=1$. Stopping time $\hat{T}_0=1$, attack instance $\hat s_0=+\infty$.
> - While $t\leq n$:
> 1. Store the local SGD iterates $Y_t$, and calculate $R_t=\max_{1\leq s\leq t}s|\bar Y_s-\bar Y_t|,$ and, $s_t=\max_{1\leq s\leq t}s|\bar Y_s-\bar Y_t|.$
>
> 2. For $b=1,...,B$:
>        -Draw $Z_t^{(b)}\sim N(0,V_K)$, and do $Y_{t,1}^{G,(b)}=(I-\eta_tA)Y_{t-1,1}^{G,(b)}+\eta_tZ_t^{(b)}K^{-1/2}$
> with $Y_{0,1}^{G,(b)}=0$.
>        -Calculate $R_t^{G,(b)}=\max_{1\leq s\leq t}|\bar Y_{s,1}^{G,(b)}-\bar Y_{t,1}^{G,(b)}|$
>
> 3. Compute $\hat q_{1-\alpha}(t)\leftarrow$ sample quantile $(\{R_t^{G,(b)}\})$.
>
> 4. **Thresholding:** If $R_t>\hat q_{1-\alpha}(t)+c\sqrt{n}$:
>       $\hat T_0\leftarrow t$, $\hat s_0 \leftarrow s_{\hat T_0}$. Stop.
>    Else, $t\leftarrow t+1$.
> - **Output:** $\hat T_0I_\{\hat T_0<n\},$  $s_{\hat T_0}$
>
> For a corresponding empirical validation, we consider the `Frand-eff` model in Section F of the paper, and consider an attack at time point $t_0=T/2$ for $K_0=K/2$ many clients, where their corresponding parameters $\beta_k$ change to $\beta_k'=\beta_k+\mu$. We take $T=500$, $K=10$, $\tau=20$ and for each setting, the above algorithm is run for $B=500$ bootstrap samples. The empirical power of the described algorithm is reported below, based on $500$ Monte-Carlo simulations.
>
> |$\mu$|Probability of detection|Attack instance $s_{\hat T_0}$ (mean, 95% CI)|Stopping time $\hat T_0$ (mean, 95% CI)|
> |-|-|-|-|
> |0(No attack)|0.046 (False positive)|-|-|
> |1|0.966|265.203, (144.05, 309)| 412.49, (345.05, 482)|
> |2|1|249.672, (118.85, 286.575)|356.32, (311.475, 397.525)|
> |3|1|249.572, (94.275, 276)|334.57, (282.95, 367)|
>
> Clearly, the more severe an attack (large $\mu$), the quicker it gets detected, and the attack instance is also estimated with higher precision. Finally, note the algorithm can be modified to perform sequential tests only at synchronization steps, rather than at every iteration $t$.

---

> > ### Comment · Reviewer_SopJ · 2025-08-01
> >
> > Thank you for the authors' response. Most of my concerns have been addressed. I have a small follow-up question regarding Point 2:
> >
> > In this paper, does each decentralized agent hold a local dataset and, in every iteration, perform SGD using a mini-batch of samples drawn from that dataset? Additionally, do the sample size and batch size affect the theoretical results presented in the paper?

---

> > > ### Author Response · Authors · 2025-08-01
> > >
> > > We thank the reviewer for the acknowledgment and for asking this insightful and relevant follow-up question. Our setting is *online*: at each iteration, every client/decentralized agent accesses exactly one sample, i.e., local SGD is performed with mini-batch size $B=1$. Consequently, if each client has a local dataset of size $n$, local SGD executes $n$ iterations.
> > >
> > > For the *Berry-Esseen theorems 2.1 and 2.2*, our results directly extend to mini-batch scenarios provided $n/B \to \infty$. The new Berry-Esseen error rate becomes $n^{1/2-\beta}B^{\beta-1/2} \sqrt{K}$., and the limiting covariance matrix $\Sigma$ in our Theorem 2.3 *does not change*. Noting that $\beta>1/2$, this is in line with the empirical observations that large batch sizes have diminished performance even for vanilla SGD (see Lin et al. [*Don't use large minibatches, use local SGD*, ICLR, 2020] and Goyal et al. [*Accurate, Large Minibatch SGD: Training ImageNet in 1 Hour*, Preprint, 2017]).
> > >
> > > However, for *time-uniform approximations* (Theorems 3.1 and 3.2), the issue is more nuanced due to the requirement for uniform control over iterates, and can improve with increasing mini-batch size to a certain extent. We expect that an optimal error rate would require selecting batch size $B \asymp n^c$, with exponent $c$ depending intricately on moment conditions, iterate dependencies, and possibly even number of client $K$; see Berkes et al. [*Komlós–Major–Tusnády approximation under dependence*, Annals of Probability, 2014] for related results of time-uniform approximations of partial sums under stationary dependence. A mini-batched version in our problem likely leads to a partial sums of blocks instead, which, to our advantage, is also (approximately) a stationary dependent series. Based on the familiarity of techniques in this literature, in practice we expect $B= n^{1/3}$ to work well. To summarize, determining optimal batch size as a function of iteration count $n$ and client number $K$ remains an exciting open problem for time-uniform approximations.

---

> > > > ### Comment · Reviewer_SopJ · 2025-08-02
> > > >
> > > > Thank you for your response. I appreciate the clarifications and will raise my score accordingly.

---

> > > > > ### Author Response · Authors · 2025-08-05
> > > > >
> > > > > Thank you very much for your positive feedback and for reconsidering your evaluation.

---

### Official Review · Reviewer_PXEE · 2025-07-20

**Clarity:** 4
**Significance:** 3
**Originality:** 3
**Rating:** 5
**Confidence:** 4

**Summary:**

This paper proposes two generalized Gaussian approximation methods for local SGD in Decentralized Federated Learning (DFL). First, the authors derive Berry-Esseen bounds for the final local SGD iterates, enabling valid multiplier bootstrap procedures. Second, they provide two sharp time-uniform Gaussian approximations over the local SGD trajectory. Specifically, the theoretical results proposed in this paper are related to the choice of the number of iterations $n$, and the number of clients $K$, the step size $\beta$. These results enable the development of valid and powerful statistical inference methods, thereby enhancing the performance of federated learning in terms of privacy, robustness, and computational efficiency. Empirical results not only validate the theoretical analysis in the paper but also demonstrate the effectiveness of the proposed methods compared to off-the-shelf Brownian-motion-based approximation.

**Questions:**

Are there any empirical results to indicate the effectiveness of the proposed GA methods for detecting adversarial attacks?

**Ethical Concerns:**

["NO or VERY MINOR ethics concerns only"]

**Limitations:**

This paper focuses on proposing the asymptotic statistical guarantees beyond convergence properties of local SGD under DFL. However, more experimental evidence could be provided to demonstrate the effectiveness and potential impact of the proposed sharp time-uniform Gaussian approximations.

**Paper Formatting Concerns:**

There is no formatting concern.

**Quality:**

3

**Strengths And Weaknesses:**

### Strengths
1. The idea proposed in the paper is easy to follow, and the proposed asymptotic statistical guarantees for local SGD under DFL are potentially impactful.
2. The elaboration of the proposed theoretical results is sufficient and intuitive.
3. Both theoretical analysis and empirical results are provided to justify the effectiveness of the proposed Aggr-GA and Client-GA.

### Weaknesses
1. The pseudocode for the two sharp time-uniform Gaussian approximations could be provided.
2. Some of the interpretations of empirical results are inconsistent when comparing the main body and the appendix. For example, $\tilde{d}_c$ does not linearly increases with $K$ for fixed $n$ in Figure 1 right, and is caused by the different dominate term in Eq. (2.5) when increasing $K$. However, the analysis in Section 4.1 says that $\tilde{d}_c$ increases with $K$ for fixed $n$.
3. Additional comparative analysis with other approaches in the literature could strengthen the findings.
4. Typos in line 320, "we discuss the Berry-Esseen error Berry-Esseen error $d_C$...".

---

> ### Author Rebuttal · Authors · 2025-07-30
>
> We sincerely thank the reviewer for carefully reading our manuscript and for the positive comments on its readability and thoroughness. Below, we address your comments individually. We warmly welcome any further suggestions to improve the paper. If our responses adequately address your concerns, we kindly ask you to consider increasing your score.
>
> ### Weaknesses
>
> 1. ***The pseudocode for the two sharp time-uniform Gaussian approximations could be provided.***
>
>       *Answer*:  The time-uniform Gaussian approximations follow an iterative scheme given by equations (3.3) and (3.5) in the paper. Here we present a pseudocode:
>
>      **Algorithm: Time-uniform Gaussian approximations**
>
>      Input: Client-level variances $H_k$, $k\in[K]$.
>
>     - Let $E_{\tau} = \{\tau,2\tau,\ldots,L\tau\}$, where $L = \lfloor n / \tau \rfloor$.
>     - For $t=1,..,n$:
>
>     A. Store the local SGD iterates $Y_t = K^{-1} \Theta_t \boldsymbol{1}$, where $\Theta_t = (\Theta_{t-1} - \eta_t G_t)C_t$
> with
> $C_t = \begin{cases}\mathbf{C}, & \text{if } t \in E_{\tau} \\\\I_K, & \text{otherwise.}\end{cases}$.
>
>    B. {Aggr-GA}: Draw $Z_t \sim N(0, K\sum_{k=1}^K H_kH_k^\top)$, and do $Y_{t,1}^G = (I - \eta_t A) Y_{t-1,1}^{G, (b)} + \eta_t Z_t^{(b)} K^{-1/2}$ with $Y_{0,1}^{G, (b)} = 0$.
>
>   C. {Client-GA}: Draw $Z_t^k\sim N(0, H_k)$ for $k=1,..,k$, and do $\Theta_t^{G} = \big((I -\eta_t A) \Theta_{t-1}^{G} + \eta_t M_t\big)C_t$, where $M_t:= K(w_1 Z_t^1, \ldots, w_K Z_t^K)$, and define $Y_{t,2}^G=K^{-1}\Theta_t^G 1_K$.
>
> What Theorem 3.1 (and Theorem 3.2) says is that the distribution of $Y_{t,1}^{G}$ is close to that of $\{Y_t\}$ in a very strong sense.
>
> 2. ***Some of the interpretations of empirical results are inconsistent when comparing the main body and the appendix. For example, $\tilde{d}_c$ does not linearly increases with $K$ for fixed $n$ in Figure 1 right, and is caused by the different dominate term in Eq. (2.5) when increasing $K$. However, the analysis in Section 4.1 says that $\tilde{d}_c$ increases with $K$ for fixed $n$.***
>
>   *Answer*: We sincerely regret the confusion caused; due to space constraints, we could not include every detail in the main manuscript. However, in the supplementary Appendix section F.1, we write:
>
>    > "On the other hand, for Setting 2, Figure 1 (right) seems to point towards a trade-off in terms of $K$ for fixed $n$. This particular behavior becomes clearer as we recall equation (2.5). For fixed $n=300$, the initial decay of ${d}_c$ with increasing $K$, is caused by the $n^{-\beta/2}K^{-1/2}$ term. However, as $K$ increases, the term $n^{1-\beta/2}\sqrt{K}$ starts to dominate, leading the error $d_c$ to increase with increasing $K$."
>
> This is in line with equation (2.5) of Section 2, where, in Remark 2.1, we also mention a similar trade-off. We hope this clears up the confusion, and we have added this part in the main draft.
>
> 3. ***Additional comparative analysis with other approaches in the literature could strengthen the findings.***
>
> *Answer*: We are not aware of similar results in the literature. As reviewers *mVKL* and *6krr* note, our work provides the *first* Berry-Esseen bounds and time-uniform Gaussian approximation bound in the Federated Learning/`local sgd` setup. Therefore, our paper primarily focuses on novel theoretical contributions, and does not aim at proposing or competing with SOTA algorithms. Nonetheless, we acknowledge that competing approaches might exist for some applications of our results, such as attack instance detection. However, to the best of our knowledge, the literature mostly revolves around robustness guarantees (error bounds, convergence rates) assuming a certain adversarial profile or detection of malicious clients (Blanchard et al., [*Machine Learning with Adversaries: Byzantine-Tolerant Machine Learning*], NeurIPS 2017; Wang et al., [*Attack of the Tails: Yes, You Really Can Backdoor Federated Learning*], NeurIPS 2020; Qian et al., [*ByMI: Byzantine Machine Identification with False Discovery Rate Control*], ICML 2024; Mapakshi et al., [*Temporal Analysis of Adversarial Attacks in Federated Learning, Machine Learning, Deep Learning and AI for Cybersecurity*, 2025] ). This makes our approach rather unique.
>
> 4. ***Typos in line 320: "we discuss the Berry-Esseen error Berry-Esseen error $d_C$...".***
>
> *Answer*: We have fixed it and tried removing any other typo in the paper.
>
> ### Questions
> ***Are there any empirical results to indicate the effectiveness of the proposed GA methods for detecting adversarial attacks?***
>
> *Answer*:  This has been addressed in detail below.
>
> ### Limitations
> ***This paper focuses on asymptotic statistical guarantees beyond convergence for local SGD under DFL. However, more experimental evidence could demonstrate the effectiveness and potential impact of the proposed sharp time-uniform Gaussian approximations.***
>
> *Answer*:  Thank you for this important suggestion. In our paper, we discussed in equations (3.1) and (3.2) (Section 3.1) how our Gaussian approximation results help detect model poisoning attacks. Below, we first provide pseudo-code for an algorithm leveraging our `aggr-GA` result (Theorem 3.1) to detect such attacks, followed by empirical exercises and real-data experiments. Similar algorithms can be devised using Theorem 3.2.
>
> **Algorithm: Attack detection using time-uniform Gaussian bootstrap**
>
> - **Input:** Connection matrix $C$; Synchronization parameter $\tau$; Loss functions $f_k(\theta, \xi^k)$, $\xi^k \sim P_k, k \in [K]$; weights $w_k$; number of iterations $n$; step-size schedules $\eta_{t}$; Hessian $A$; number of bootstrap samples $B$; covariance matrix $V_K$. Type-1 error level $\alpha$.
> -  Initialize $t = 1$. Stopping time $\hat{T}_0 = 1$, estimated attack instance $\hat{s}_0 = +\infty$.
> - While $t \leq n$:
> 1. Store the local SGD iterates $Y_t$, and calculate
>    $$ R_t = \max_{1 \le s\le t} s\bigl|\bar Y_s-\bar Y_t\bigr|
>      ,\quad
>      s_t = \max_{1 \le s\le t} s\bigl|\bar Y_s-\bar Y_t\bigr|.$$
> 2. For $b = 1,\dots,B$:
>    - Draw $Z_t^{(b)} \sim \mathcal N(0, V_K)$, and set
>      $$Y_{t,1}^{G,(b)}
>        = (I - \eta_t A)\,Y_{t-1,1}^{G,(b)} + \eta_t\,Z_t^{(b)}\,K^{-1/2},
>        \quad
>        Y_{0,1}^{G,(b)} = 0.$$
>    - Calculate
>      $$R_t^{G,(b)}=\max_{1 \le s\le t}s\bigl|\bar Y_{s,1}^{G,(b)} - \bar Y_{t,1}^{G,(b)}\bigr|.$$
> 3. Compute $\hat q_{1-\alpha}(t)$ = sample‑quantile $(R_t^{G,(b)})_{b=1}^B.$
>
> 4. **Thresholding:**
>    If
>    $$R_t > \hat q_{1-\alpha}(t) + c\,\sqrt n$$
>    then set $\hat T_0 = t$, $\hat s_0 = s_{\hat T_0}$ and **stop**,
>    otherwise $t\leftarrow t+1$.
>
> - Output:
> $$\hat T_0 I_{\{\hat T_0 < n\}}\; s_{\hat T_0}.$$
> A detailed explanation on the theoretical validity of this algorithm can be found in the rebuttal to **Question 5, reviewer SopJ**. For simulations, we consider the `Frand-eff` model introduced in **Section F of the manuscript**. Recall that at iteration $t$, client $k$ observes data $(y_{tk}, x_{tk}) \in \mathbb{R} \times \mathbb{R}^d$ generated by the linear model
> $y_{tk} \sim N(x_{tk}^\top \beta_k, \sigma_k^2).$
> We simulate an attack at $t_0 = T/2$, affecting $K_0 = K/2$ clients, shifting parameters to $\beta_k' = \beta_k + \mu$. Set $T=500$, $K=10$, $\tau=20$. The empirical power of the proposed detection algorithm is reported below for different values of $\mu$.
>
> | $\mu$| Probability of detection| Attack instance (mean, 95% CI)| Stopping time $\hat{T}_0$ (mean, 95% CI)|
> |-|-|-|-|
> | 0 (No attack)| 0.046 (False positive)| - | - |
> | 1 | 0.966 | 265.203, (144.05, 309)| 412.49, (345.05, 482.)|
> | 2 | 1 | 249.672, (118.85, 286.575)| 356.32, (311.475, 397.525)|
> | 3| 1| 249.572, (94.275, 276)| 334.57, (282.95, 367)|
>
> Clearly, the more severe an attack (large $\mu$), the quicker it gets detected, and the attack instance is also estimated with higher precision. Finally, note the algorithm can be modified to perform sequential tests only at synchronization steps, rather than at every iteration $t$.
>
> For the real-data experiment, we work on a federated learning (FL) setup with $K = 5$ clients collaboratively training a linear classifier on *MNIST* data. Let each image $x_i \in \mathbb{R}^{784}$. To handle high-dimensionality, a PCA transform $P: \mathbb{R}^{784} \rightarrow \mathbb{R}^d$ with $d = 3$ is fitted on the full training set $z_i = P(x_i) \in \mathbb{R}^3$. At time point $t$, each client $k \in [K]$ sequentially receives $(y_t^k, z_t^k)$ where $y_t^k$ are corresponding digit labels. Following the notation of the paper, the loss function of client $k$ corresponds to the *cross-entropy loss* for the logistic model $y_t^k \sim \operatorname{Multinomial} (1, W z_t^k + b_k)$, where $W \in \mathbb{R}^{10 \times 3}$ and $b \in \mathbb{R}^{10}$, so the parameter vector $\theta = \operatorname{Vec}([W:b]) \in \mathbb{R}^{d}$ with $d=40$. The aggregated objective uses weights $w_k=K^{-1}$ for $k \in [K]$, with step-size $\eta_t= 0.3 t^{-0.75}$ consistent with our empirical analysis. The synchronization parameter is $\tau=5$, and we reuse the connection matrix from simulation: $\mathbf{C}_{ij}=1/3 I\{|j-i|\leq 1\}$. Finally, we run the local SGD iterates for $n=100$ iterations.
>
> For a randomly selected subset of $K_0=3$ clients, a *label-flipping attack* (labels of digits $1,2,4$ swapped with $7,5,8$, respectively, and vice versa) is introduced at $t=50$. The Gaussian bootstrap-based algorithm is applied for detection, and the results are summarized below.
>
> || Probability of detection|Attack instance $\hat{s}_0$  (mean, 95% CI)|Stopping time $\hat{T}_0$ (mean, 95% CI)|
> |-|-|-|-|
> | No attack| 0.06 (false positive) |-|-|
> |Label flipping attack| 0.90|58.49, (31.7, 96.25)| 95.67, (89.1, 98.9)|
>
> The method maintains empirical validity at level $0.05$ and has high detection power under label-flipping attacks. Earlier detection is achievable by tuning the threshold constant $c$ (e.g., via cross-validation), demonstrating the practical applicability of Gaussian-bootstrap algorithms.

---

> > ### Comment · Reviewer_PXEE · 2025-08-06
> >
> > I would like to thank the authors for their response. My concerns are addressed. I will keep my score.

---

> > > ### Author Response · Authors · 2025-08-08
> > >
> > > Dear Reviewer,
> > > Thank you for your positive feedback.

---

### Note · Authors · 2025-08-13

We thank all the reviewers for their exhaustive reviews and detailed, constructive feedback on our paper. We found the reviewers to be in consensus about originality and non-triviality of our theoretical and numerical results. In particular, they appreciated that (i) our paper was the *first* paper going beyond convergence and central limit theory for *federated learning* to develop refined Gaussian approximation results; (ii) and the potential impact of our results in facilitating statistical inference and attack detection. Our theoretical results were deemed to be comprehensive and rigorous with detailed proofs, and with particular commendation for quantifying the roles various parameters of the local SGD algorithms played in the federated learning set-up. Our numerical exercises were recognized for not only validating the theory, but also the effectiveness of our proposed *time-uniform* results, which are also completely new contributions in this literature.

The reviewers also raised some interesting questions on potential applications as well as some further generalization of the theory. Most common questions include:  **(i)** potential relaxation from strong convexity, **(ii)** explicit quantification of effect of heterogeneity in our results, and **(iii)** justification or use case of adversarial attack detection using our time-uniform Gaussian approximation algorithms. We addressed each of the reviewers' queries in the rebuttal in appropriate places. Specifically, the theoretical concerns were addressed by producing generalizations of our theorems and clarifying technical assumptions; the attack detection question was resolved by proposing a novel bootstrap algorithm- complete with justifications based on our theoretical results, and pseudo-code- which were then validated via simulations and MNIST dataset. Reviewers broadly agree that our revisions comprehensively address their concerns.

For the camera ready version, we will revise our manuscript as indicated above.  We hope that our work will be evaluated favorably in light of its originality, technical novelty, potential impact and the revisions made in response to the reviews. Finally, we would like to express our most sincere gratitude to all reviewers, the AC, and the SAC again for their time, expertise, and thoughtful engagement with our work. The detailed feedback not only helped us address potential ambiguities but also inspired meaningful improvements of the quality of the final paper.

---

### Decision · Program_Chairs · 2025-09-17

**Decision:**

Accept (spotlight)

**Comment:**

This paper proposes two generalized Gaussian approximation methods for local SGD in decentralized federated learning. The authors derive Berry-Esseen bounds for the final local SGD iterates, enabling valid multiplier bootstrap procedures. They also provide two sharp time-uniform Gaussian approximations over the local SGD trajectory. The proposed asymptotic statistical guarantees for local SGD in decentralized federated learning are potentially impactful, the elaboration of the theoretical results is sufficient and intuitive, and the effectiveness of the proposed approximations are well justified. Although the analytical tools are originated from [Shao and Zhang, 2022], the application in decentralized federated learning is novel and the contributions are sufficient. For these reasons, I recommend Accept.

The authors are encouraged to incorporate the reviewers’ comments and polish the presentation during preparing the final version.